# Distilling Knowledge from Caption-Guided Replay for VLM-based Continual Learning

## Abstract

Continual learning with vision-language models is challenged by catastrophic forgetting, where the acquisition of new knowledge compromises previously learned information. Generative replay synthesizes past samples to mitigate forgetting, while avoiding the data-privacy risks and heavy storage overhead of directly replaying historical data. However, existing methods often rely on simple class-level prompts, such as class-name with templates, resulting in synthetic images that poorly capture the semantics of original images. To address this, we propose a *caption-guided replay paradigm* that stores instance-level captions generated by a Multi-modal LLM as memory and reconstructs past images using a LoRA-adapted text-to-image model. This approach enables high-fidelity and instance-aware synthetic replay while remaining efficient in storage. In addition to improving replay fidelity, we observe the phenomenon of *feature drift* in continual learning, which refers to pervasive shifts in intermediate representations during sequential training and is only partially addressed by logit distillation. To address this, we introduce a distribution-based distillation method that aligns feature distributions at multiple intermediate layers, effectively suppressing feature drift and enhancing model stability. Extensive experiments under various settings demonstrate that our proposed method consistently outperforms state-of-the-art approaches.

## 1 Introduction

Vision-language models (VLMs) like CLIP (Radford et al., 2021) exhibit strong zero-shot generalization due to large-scale pre-training, yet extending their knowledge often requires expanding the pre-training corpus and retraining the model, which is computationally expensive (Wang et al., 2024; Dohare et al., 2024; Kudithipudi et al., 2022). Continual learning (CL) provides an efficient alternative by incrementally learning new tasks. However, sequential fine-tuning is vulnerable to catastrophic forgetting, which degrades knowledge from previous tasks and pre-training data (McCloskey & Cohen, 1989; Goodfellow et al., 2013; Zheng et al., 2023). Replay-based strategies are widely used to mitigate forgetting (Rebuffi et al., 2017; Lopez-Paz & Ranzato, 2017), but directly replaying historical data raises privacy and storage concerns and is often infeasible when pre-training data are unavailable (Shin et al., 2017; Van de Ven et al., 2020; Zheng et al., 2023). As an alternative, generative replay addresses these constraints by synthesizing past sample, typically by storing a image generator to generate images for replay (Wu et al., 2025; Meng et al., 2024).

However, existing generative replay methods are limited by their reliance on simple prompts constructed from the class name and CLIP-style templates, which fail to capture instance-aware semantics and often result a fidelity gap relative to original task data. To overcome these limitations, we propose a novel *caption-guided replay paradigm* that stores instance-level captions as its primary memory and subsequently reconstructs past images from them. Specifically, we leverage a Multi-modal Large Language Model (Multi-modal LLM) (Bai et al., 2025) to generate instance-level captions that capture the semantic content of past samples. We then fine-tune Text-to-Image (T2I) model Esser et al. (2024) with task-specific LoRA (Hu et al., 2022) on past image-caption pairs to preserve the corresponding visual characteristics. Finally, we use instance-level captions to guide the LoRA-adapted T2I model to reconstruct the past images.

The effectiveness of our approach is demonstrated in Figure 1a, which compares real images with synthetic images generated by three strategies: class name with template (Class name & template),

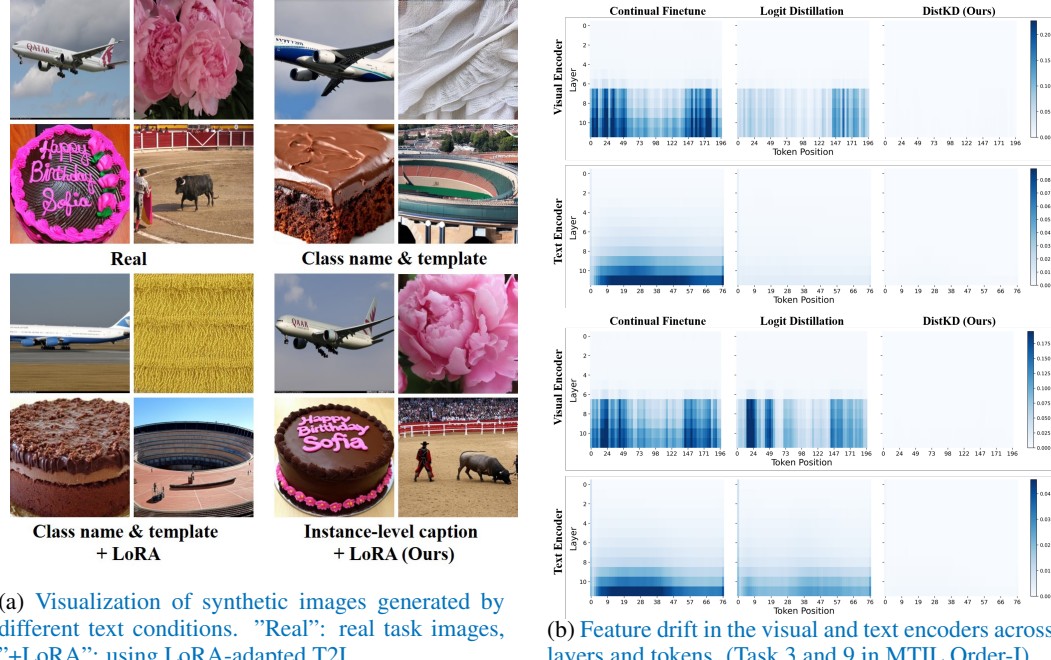

(a) Visualization of synthetic images generated by different text conditions. "Real": real task images, "+LoRA": using LoRA-adapted T2I.

(b) Feature drift in the visual and text encoders across layers and tokens. (Task 3 and 9 in MTIL Order-I).

Figure 1: Key challenges and improvements in replay-based continual learning. (a) Visual comparisons show that our caption-guided replay produces high-fidelity, instance-aware synthetic images, narrowing the gap to real data. (b) Continual finetuning and Logit distillation exhibit substantial multi-layer drift, whereas our DistKD suppresses these shifts, motivating multi-layer distribution distillation as a key ingredient for VLM-based continual learning.

class name combined with LoRA-adapted T2I (Class name & template+LoRA), and our caption-guided method (Ours). For example, on the left side, the lower right image for each method corresponds to the "bullring" category of SUN397 (Xiao et al., 2010). Synthetic images generated from simple class name prompts ("Class name & template" and "Class name & template+LoRA") appear as generic stadiums lacking key details such as the arena layout, red fencing, and ochre-colored ground that are found in the real image, while our method faithfully reconstructs the spatial structure, and entities of the scene, closely matching the real image.

With access to high-fidelity synthetic data, we investigate how to best leverage these samples for continual learning. Prior methods typically focus solely on aligning the output logits between the current and reference models (Rebuffi et al., 2017; Wu et al., 2025), but this strategy provides only limited protection against forgetting. To gain deeper insight, we visualize activation changes at each layer and token in Figure 1b. Specifically, we compute the mean squared Euclidean (MSE) distance between activations before and after learning the task, averaged over all replay samples. The visualization results reveal that continual fine-tuning (left) induces substantial shifts in multi-layer intermediate features, a phenomenon we term *feature drift*. However, logit distillation (middle) only partially mitigates this issue. The feature drift, which has received limited attention in prior work, contributes to instability and catastrophic forgetting, even when high-quality replay is available. Motivated by this observation, we introduce DistKD (right), a distribution-based distillation method that explicitly aligns feature distributions at multiple intermediate layers.

To this end, we propose CaRD, which integrates a Caption-guided Replay paradigm with multi-layer distribution-based Distillation. Our main contributions are summarized as follows:

- We propose a novel *caption-guided replay* paradigm for VLM-based continual learning, which stores instance-level captions generated by a Multi-modal LLM as memory and reconstructs past images using LoRA-adapted T2I model. This enables high-fidelity and instance-aware synthetic replay while preserving privacy and storage efficiency.

- We analyze forgetting during continual learning by visualizing activation changes at each layer and token, revealing that *feature drift* exists throughout the entire network. To mitigate this, we

introduce a multi-layer distribution-based distillation strategy that aligns feature distributions, effectively mitigating feature drift than conventional logit distillation.

- Extensive experiments on the standard continual learning benchmark MTIL demonstrate that our method CaRD consistently outperforms its replay- and distillation-based counterparts as well as previous state-of-the-art approaches across various scenarios.

## 2 RELATED WORK

**Continual Learning for VLMs.** Continual learning (CL) aims to enable models to learn from a sequence of tasks without suffering from catastrophic forgetting. In the context of vision-language models (VLMs), a key additional challenge is to preserve the generalization capabilities maintained from pre-training (Zheng et al., 2023). In prior work, regularization-based methods (Kirkpatrick et al., 2017) mitigate forgetting by penalizing changes to parameters important for previous tasks, typically via the Fisher Information Matrix. Architecture-based methods (Yu et al., 2024a; Xu et al., 2024) introduce lightweight, task-specific modules, such as adapters or learnable prompts (Wang et al., 2022). Recent work such as LADA (Luo et al., 2025) further advances this direction by introducing label-specific adapters that attach per-class memory units to the frozen CLIP backbone, improving plasticity without relying on task identifiers while leveraging distillation to stabilize representations..In contrast, replay-based methods (Rebuffi et al., 2017; Buzzega et al., 2020) revisit historical data to consolidate past knowledge. However, storing real images imposes heavy storage and privacy costs, especially in VLMs where pre-training data are often inaccessible. Several prototype augmentation-based methods (Zhu et al., 2021; Asadi et al., 2023) address this limitation by preserving a compact set of class prototypes and augmenting them through feature perturbation or relational constraints to approximate the distribution of historical tasks. Another prominent direction is generative replay (Wu et al., 2025; Meng et al., 2024), which synthesizes historical data using generative models instead of retaining raw samples. Furthermore, we propose a novel caption-guided replay paradigm that stores instance-level captions and subsequently reconstructs past images from them, more faithfully bridge the gap between synthetic and real historical data.

**Synthetic Data for CL.** Generative replay synthesizes synthetic historical data to mitigate forgetting. Early methods involved inverting the trained model (Yin et al., 2020; Choi et al., 2021) or training generative models like GANs (Shin et al., 2017; Wu et al., 2018) or Diffusion Models (Ho et al., 2020; Gao & Liu, 2023) from scratch. More recent work has leveraged powerful, large-scale pre-trained text-to-image (T2I) models to synthesize high-fidelity data (Rombach et al., 2022; Esser et al., 2024; Wu et al., 2025), sometimes fine-tuning them on task-specific data (Jodelet et al., 2023; Meng et al., 2024; Kim et al., 2024). A persistent limitation, however, is the reliance on simple class-level text prompts, which fail to capture the semantic content of the original data. While some efforts have used large language models (LLMs) with pre-defined rules to enrich prompts (Yang et al., 2025; Tian et al., 2024), these still lack the instance-level specificity needed for faithful data reconstruction. Beyond this, we employ a Multi-modal LLM to generate instance-level captions for historical task images and subsequently reconstructs these image from captions.

**Knowledge Distillation for CL.** Knowledge distillation (KD) is a cornerstone technique in CL for consolidating old knowledge, typically by constraining a student model (training on current task) to mimic the outputs of a teacher model (trained on last task) to resist forgetting. Prior works (Li & Hoiem, 2017; Zhou et al., 2023; Zhang et al., 2024) relied solely on current-task data for distillation, while subsequent methods (Rebuffi et al., 2017; Buzzega et al., 2020) improved performance by retaining a small set of past samples as more relevant inputs for distillation. In the VLM setting, ZSCL (Zheng et al., 2023) and Yu et al. (2024b) utilize ImageNet as a proxy dataset to preserve zero-shot capabilities from pre-training, while GIFT (Wu et al., 2025) employs a pre-trained T2I model to generate synthetic data for distillation. Regardless of the data source, these methods restrict the distillation objective to aligning logits at the final network layer. However, this strategy overlooks the "feature drift" phenomenon that arises across multi-layer intermediate features. Motivated by recent advances in distribution-based knowledge distillation Lv et al. (2024); Ahn et al. (2019), we address this limitation by combining a caption-guided replay paradigm with multi-layer distribution distillation, jointly improving the fidelity of replay data and mitigating "feature drift" that underlies forgetting in VLM-based continual learning. Although PODNet (Douillard et al., 2020) also adopts multi-layer distillation via spatial pooling, our approach is explicitly motivated by the empirical

162
163
164
165

diagnosis of feature drift in VLMs, necessitating a distribution-based alignment of token statistics to directly counteract these shifts.

166 ## 3 METHOD
167
168 As illustrated in Figure 2, our proposed CL method CaRD consists of two main components:
169 caption-guided replay and distribution-based distillation. Caption-guided replay leverages a Multi-
170 modal LLM and LoRA-adapted T2I model to preserve and reconstruct past images from previous
171 tasks, providing high-fidelity replay data for distillation. Distribution-based distillation aligns fea-
172 ture distributions at multiple layers, effectively mitigating feature drift that arises during CL.

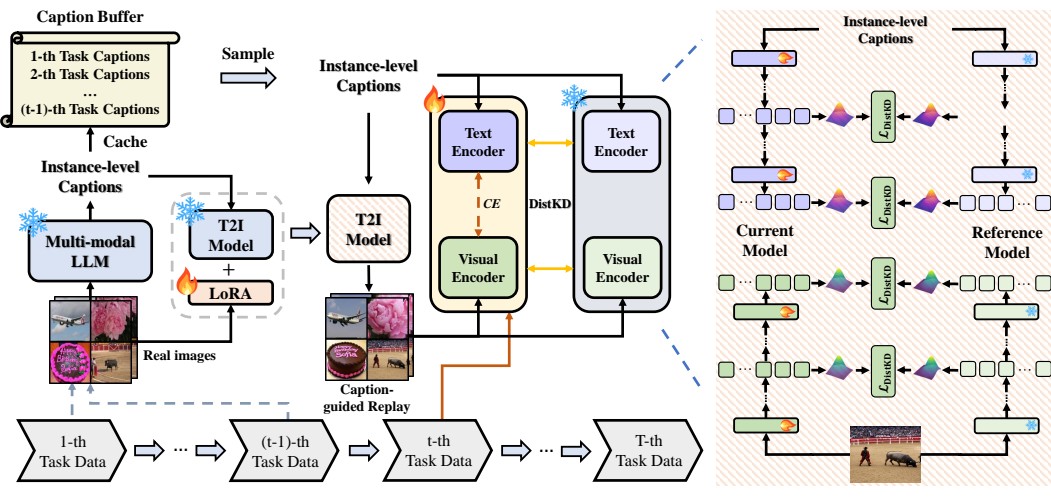

(a) Caption-guided Replay  (b) Distribution Distillation.

Figure 2: Overview of CaRD. (a) For each past task, Multi-modal LLM generates instance-level
captions from real images to capture their semantic content, and these captions are cached in a
*Caption Buffer*. These real images and their corresponding synthetic captions are then used to fine-
tune a T2I model with task-specific LoRA to preserve the visual information of each past task.
During CL on current task, captions sampled from the buffer guide the LoRA-adapted T2I model to
reconstruct past images. (b) The resulting synthetic image-caption pairs are used as replay data in
the DistKD (distribution distillation) stage, which aligns intermediate feature distributions of both
visual and text encoders between the current model and a frozen reference model.

### 3.1 CAPTION-GUIDED REPLAY

Our caption-guided replay paradigm includes three stages. First, instance-level captions for each
task are generated and stored in a caption buffer. Second, the T2I model is adapted to each task
by fine-tuning on real images and their corresponding captions using LoRA. Third, synthetic replay
data are constructed by combining sampled captions with task-specific LoRA weights during CL.

**Instance-level Captions Generation.** As shown in Figure 2a, upon completion of CL training on
task $i$,, we employ a frozen Multi-modal LLM to generate detailed, instance-level captions $c_j^i$ for
each image $x_j^i$ in the training set $\mathcal{X}_i = \{x_j^i\}_{j=1}^{N_i}$, where $N_i$ denotes the number of training samples
in task $i$ and $j$ indexes each image. The resulting synthetic captions for task $i$ are incrementally
cached to a *Caption Buffer*, which accumulates textual descriptions for all previously seen by the
VLM. Thus, at current task $t$, the buffer is defined as $\mathcal{B}_{t-1} = \mathcal{B}_{t-2} \cup \{c_j^{t-1}\}_{j=1}^{N_{t-1}}$.

**Task-specific adaptation of T2I Model via LoRA.** In our caption-guided replay paradigm,
instance-level captions generated by the multi-modal LLM provide a compact semantic represen-
tation for each training sample, while LoRA-based adaptation enables the T2I model to retain the
visual characteristics of historical data. For each completed task, we fine-tune the T2I model us-
ing the paired real images and their synthetic captions, updating only the LoRA weights and and
all other parameters of T2I remain frozen. LoRA adapters are applied to all attention layers of the

diffusion module, such as SD3-Medium's MMDIT (Esser et al., 2024). The resulting task-specific LoRA weights are cached for replay, allowing the model to reconstruct images with high fidelity that reflect both the semantic content and visual distribution of each seen task.

**Generative replay for continual learning.** At the start of training for current task $t$, we construct replay data by sampling instance-level captions from the Caption Buffer $\mathcal{B}_{t-1}$. For each sampled caption, the corresponding task-specific LoRA weights are loaded into the T2I model to reconstruct synthetic images. To further preserve CLIP's zero-shot generalization capabilities, we approximate its pre-training data by leveraging the method from (Yang et al., 2025) to generate LLM-augmented class-level descriptions for ImageNet class names. The resulting synthetic image-caption pairs allow the model to learn from both current and historical knowledge while respecting data privacy. Since synthetic images are generated on demand and discarded after each task, all synthetic images are discarded after each training task. We only incrementally retain the synthetic captions and lightweight LoRA adapters, substantially reducing storage costs compared to storing real data.

## 3.2 DISTRIBUTION DISTILLATION

To effectively address the pronounced feature drift that arises during CL and is insufficiently mitigated by conventional logit distillation, we introduce a distribution-based distillation method DistKD. It explicitly aligns multi-layer intermediate feature distributions between the current and reference models by combining global alignment of feature statistics with fine-grained local alignment at the token level.

**Global Distillation via Wasserstein Distance.** To regularize the global structure of the representation space, we align the mean and covariance of intermediate activations between the current model and the frozen reference model from last task. For distribution alignment, we consider all transformer layers indexed by $\ell \in \mathcal{S}$, where $\mathcal{S}$ is the set of selected layers. For each such layer and each modality $m \in \{v, t\}$ (where $v$ and $t$ denote the visual and text encoders, respectively), we extract the token representations $H_m^\ell \in \mathbb{R}^{P \times d}$ from the current model and $\hat{H}_m^\ell$ from the reference model, where $P$ is the number of tokens and $d$ is the feature dimension. We then compute the empirical mean and covariance for each modality and layer as follows: $\mu_m^\ell = \frac{1}{P} \sum_{j=1}^P H_{m,j}^\ell$ and $\Sigma_m^\ell = \frac{1}{P} \sum_{j=1}^P (H_{m,j}^\ell - \mu_m^\ell)(H_{m,j}^\ell - \mu_m^\ell)^\top$, where $\hat{\mu}_m^\ell$ and $\hat{\Sigma}_m^\ell$ are computed similarly for the reference model. The Wasserstein distance loss is then used to minimize the discrepancy between these statistics:

$$\mathcal{L}_{\text{Wass}}^{(\ell,m)} = \left\| \mu_m^\ell - \hat{\mu}_m^\ell \right\|_2^2 + \text{Tr}\left( \Sigma_m^\ell + \hat{\Sigma}_m^\ell - 2\big((\hat{\Sigma}_m^\ell)^{1/2} \Sigma_m^\ell (\hat{\Sigma}_m^\ell)^{1/2}\big)^{1/2} \right). \tag{1}$$

This geometry-aware objective matches both first- and second-order statistics, thereby preserving the overall distributional structure of representations even when feature differ substantially.

**Local Distillation via Token-Level Hint Alignment.** To complement the global alignment of feature statistics, we further enforce local consistency by introducing a token-level hint loss that directly aligns token representations between the current model and the reference model:

$$\mathcal{L}_{\text{Hint}}^{(\ell,m)} = \frac{1}{L} \sum_{j=1}^L \left\| H_{m,j}^\ell - \hat{H}_{m,j}^\ell \right\|_2^2. \tag{2}$$

This token-level criterion ensures fine-grained local alignment, helping to preserve detailed semantic information and maintain the relational structure among individual tokens.

**Overall Objective.** We aggregate the global and local alignment objectives into a unified distribution distillation loss by summing the Wasserstein and hint terms across all selected layers and both visual and text modalities:

$$\mathcal{L}_{\text{DistKD}} = \sum_{\ell \in \mathcal{S}} \sum_{m \in \{v,t\}} \left( \mathcal{L}_{\text{Wass}}^{(\ell,m)} + \mathcal{L}_{\text{Hint}}^{(\ell,m)} \right). \tag{3}$$

The overall training objective for CaRD is formulated as a weighted sum of the standard cross entropy loss $\mathcal{L}_{\text{ce}}$ and our distribution distillation loss $\mathcal{L}_{\text{DistKD}}$:

$$\mathcal{L} = \mathcal{L}_{\text{ce}} + \lambda \mathcal{L}_{\text{DistKD}}. \tag{4}$$

where $\lambda$ is the weight of distillation loss.

## 4 EXPERIMENTS

### 4.1 DATASETS AND METRICS.

Our experiments are conducted on two publicly established VLM-based CL benchmarks: MTIL Zheng et al. (2023) and X-TAIL (Xu et al., 2024). MTIL is a **Multi-domain Task-Incremental Learning** benchmark designed specifically for VLMs. It consists of 11 heterogeneous image classification datasets, each corresponding to an independent task. The model continually adapts to these tasks in a sequential manner, and evaluation jointly measures the retention of incrementally acquired task-specific knowledge and the preservation of pre-trained knowledge. Notably, **MTIL provides the task identity during inference**, making it a task-incremental setting. **X-TAIL extends MTIL by integrating the class-incremental evaluation protocol**. In this scenario, the model must continually learn new classes across multiple domains and distinguish among them **without access to task identity at inference time**. We adhere to the established protocols with two task orders for both full-shot and few-shot scenarios. Detailed results for few-shot scenarios are provided in the Appendix C.

For each benchmark, we report three key metrics (Zheng et al., 2023): Transfer (Trans.), Average (Avg.), and Last. To offer a more granular analysis of our method's effectiveness in mitigating catastrophic forgetting, we also include the Backward Transfer (BWT) (Lopez-Paz & Ranzato, 2017) and Average Forgetting (AF) (Yu et al., 2024b; Qian et al., 2023) metrics in our ablation studies. A comprehensive description of these evaluation metrics can be found in the Appendix A.3.

### 4.2 IMPLEMENTATION DETAILS

Following Zheng et al. (2023), we build our framework upon the CLIP ViT-B/16 architecture (Radford et al., 2021). For our caption-guided replay paradigm, we employ QWen2.5-VL (Bai et al., 2025) as Multi-modal LLM to generate instance-level captions with a temperature of 1.0, and use the SD3-Medium (Esser et al., 2024), configured with a classifier-free guidance of 5.0 and denoising steps of 28, to generate 2,000 synthetic images per task. To adapt the T2I model to each task, we fine-tune it using LoRA (Hu et al., 2022) with a rank of 4 for 1k iterations. During the continual learning on MTIL and X-TAIL, we train the CLIP using the AdamW optimizer (Loshchilov & Hutter, 2019) for 1k iterations with batch size of 64. All experiments are conducted using PyTorch (Paszke et al., 2019) with GeForce RTX 4090 GPUs. Further details are provided in the Appendix A.2.

### 4.3 COMPARISON WITH STATE-OF-THE-ART METHODS

We compare with the following state-of-the-art methods on full-shot settings of MTIL and X-TAIL in Table 1 and Table 2, respectively. For fair comparison, we group all methods into three categories: those that do not use any replay data (No replay), those that rely on real ImageNet samples for replay (Real (IN)), and those that employ generative replay (Gen.).

*For MTIL*, "CLIP Zero-shot" denotes the zero-shot performance of the initial CLIP model. "Continual Finetune" refers to continual fine-tuning without any protection. "CLIP Full Finetune" indicates independently fine-tuning the pre-trained model on each dataset, which can be seen as an upper-bound where no forgetting happens. CaRD outperforms the strongest prior methods, improving Average accuracy by 1.6% on Order-I and 2.2% on Order-II, and the Last performance by 2.3% on Order-I and 2.7% on Order-II. Specifically, CaRD achieves Last accuracies of 88.3% and 88.2% on Order-I and Order-II, respectively, approaching the upper bound "CLIP Full Finetune".

*For X-TAIL*, CaRD consistently surpasses all prior methods across both task orders. On Order-I, it achieves an *Average* accuracy of 75.8%, representing an 18.1% improvement over the CLIP zero-shot baseline, and a *Last* accuracy of 88.0%, which is 30.3% higher. Its *Transfer* score reaches 62.8%, exceeding the baseline by 5.1%. On Order-II, CaRD delivers similarly strong gains, with improvements of 15.8%, 30.0%, and 2.3% in *Average*, *Last*, and *Transfer*, respectively. Compared to the SOTA method LADA, CaRD maintains clear advantages across all metrics, particularly in *Average* and *Last* accuracy.

Table 1: Comparison of SOTA methods on MTIL under the full-shot setting for Order-I and Order-II. Δ represents the improvement over CLIP Zero-shot (%).

| Replay | Method | Order-I | | | | | | Order-II | | | | | |
|--------|--------|-------|-------|------|------|------|------|-------|-------|------|------|------|------|
| | | Trans | Δ | Avg | Δ | Last | Δ | Trans | Δ | Avg | Δ | Last | Δ |
| | CLIP Zero-shot | 69.4 | 0.0 | 65.3 | 0.0 | 65.3 | 0.0 | 65.4 | 0.0 | 65.3 | 0.0 | 65.3 | 0.0 |
| | Continual Finetune | 44.6 | -24.8 | 55.9 | -9.4 | 77.3 | +12.0 | 46.6 | -18.8 | 56.2 | -9.1 | 67.4 | +2.1 |
| | CLIP Full Finetune | – | – | – | – | 89.2 | +23.9 | – | – | – | – | 89.2 | +23.9 |
| No replay | LwF Li & Hoiem (2017) | 56.9 | -12.5 | 64.7 | -0.6 | 74.6 | +9.3 | 53.2 | -12.2 | 62.2 | -3.1 | 71.9 | +6.6 |
| | LwF-VR Zhou et al. (2023) | 57.2 | -12.2 | 65.1 | -0.2 | 76.6 | +11.3 | 53.1 | -12.3 | 60.6 | -4.7 | 68.3 | +3.0 |
| | MoE-Adapter Yu et al. (2024a) | 68.9 | -0.5 | 76.7 | +11.4 | 85.0 | +19.7 | 64.3 | -1.1 | 74.7 | +9.4 | 84.1 | +18.8 |
| | MulKI Zhang et al. (2024) | 70.1 | +0.7 | 77.3 | +12.0 | 85.1 | +19.8 | 65.6 | +0.2 | 75.0 | +9.7 | 84.2 | +18.9 |
| Real(IN) | ZSCL Zheng et al. (2023) | 68.1 | -1.3 | 75.4 | +10.1 | 83.6 | +17.4 | 64.2 | -1.2 | 74.5 | +9.2 | 83.4 | +18.1 |
| Gen. | GIFT Wu et al. (2025) | 69.3 | -0.1 | 77.3 | +12.0 | 86.0 | +20.7 | 65.9 | +0.5 | 75.7 | +10.4 | 85.3 | +20.0 |
| | LoRA-Loop Wang et al. (2025a) | 69.8 | +0.4 | 77.6 | +12.3 | 86.0 | +20.7 | 66.3 | +0.9 | 75.9 | +10.6 | 85.5 | +20.2 |
| | **CaRD (Ours)** | **71.2** | **+1.8** | **79.2** | **+13.9** | **88.3** | **+23.0** | **67.5** | **+2.1** | **78.1** | **+12.8** | **88.2** | **+22.9** |

Table 2: Comparison of SOTA methods on X-TAIL under the full-shot setting for Order-I and Order-II. Δ represents the improvement over CLIP Zero-shot (%).

| Replay | Method | Order-I | | | | | | Order-II | | | | | |
|--------|--------|-------|-------|------|------|------|------|-------|-------|------|------|------|------|
| | | Trans | Δ | Avg | Δ | Last | Δ | Trans | Δ | Avg | Δ | Last | Δ |
| | CLIP Zero-shot | 57.7 | 0.0 | 57.7 | 0.0 | 57.7 | 0.0 | 57.7 | 0.0 | 57.7 | 0.0 | 57.7 | 0.0 |
| No replay | LwF Li & Hoiem (2017) | 45.3 | -12.4 | 54.2 | -3.5 | 63.9 | +6.2 | 46.6 | -11.1 | 52.9 | -4.8 | 59.6 | +1.9 |
| | WiSE-FT Wortsman et al. (2022) | 40.9 | -16.8 | 49.2 | -8.5 | 58.0 | +0.3 | 45.2 | -12.5 | 52.7 | -5.0 | 54.2 | -3.5 |
| | MoE-Adapters Yu et al. (2024a) | 56.4 | -1.3 | 67.2 | +9.5 | 77.3 | +19.6 | 52.7 | -5.0 | 65.1 | +7.4 | 73.5 | +15.8 |
| | Primal-RAIL Xu et al. (2024) | – | – | 72.8 | +15.1 | 84.1 | +26.4 | – | – | 68.4 | +10.7 | 84.1 | +26.4 |
| | LADA Luo et al. (2025) | 61.9 | +4.2 | 75.2 | +17.5 | 86.9 | +29.2 | 55.4 | -2.3 | 69.2 | +11.5 | 86.9 | +29.2 |
| Real(IN) | ZSCL Zheng et al. (2023) | 59.0 | +1.3 | 64.5 | +6.8 | 72.1 | +14.4 | 54.2 | -3.5 | 65.1 | +7.4 | 70.1 | +12.4 |
| Gen. | **CaRD (Ours)** | **62.8** | **+5.1** | **75.8** | **+18.1** | **88.0** | **+30.3** | **60.0** | **+2.3** | **73.5** | **+15.8** | **87.7** | **+30.0** |

## 4.4 ABLATION

In this section, we conduct ablation studies on the MTIL benchmark under the full-shot setting of Order-I, isolating the contribution of each component by keeping all other parts fixed at their default settings. We first examine the impact of different textual conditions and the application of LoRA-based fine-tuning to the T2I model. Next, we compare various knowledge distillation strategies. Finally, we provide analysis of the computational cost. More ablation and visualizations are provided in the Appendix D and F.

### 4.4.1 ABLATION ON CAPTION-GUIDED SYNTHETIC REPLAY

**Textual condition and T2I adaptation.** We investigate the impact of textual condition granularity and generator adaptation on these metrics and overall synthetic replay performance in Table 3. When the T2I model is fixed (rows 3-5), using more informative textual prompts ranging from simple class names to detailed instance-level captions leads to incremental improvements in all metrics. For instance, using instance-level captions (row 5) increases Avg. to 79.1% compared to 78.8% from class name templates (row 3). The most significant gains are observed when LoRA-based adaptation is combined with instance-level captions (rows 6-8). This adaptation dramatically improves synthetic image quality, evidenced by the large reduction in FID from over 137 to ∼81, and consequently boosts learning performance. The optimal configuration, which combines LoRA with rich instance-level captions (row 8), maximizes all metrics. This result not only substantially outperforms the "No replay" (Li & Hoiem, 2017) but also closely matches the performance of using "Real replay" (row 2), demonstrating the high fidelity of our caption-guided synthesis.

**Image quality metrics.** In addition to CL metrics, we report image quality metrics to directly assess the fidelity between synthetic and real images. As shown in Table 3, we use FID (Heusel et al., 2017) to measure distributional-level similarity (lower is better) and CLIP-I/DINO-I (Ruiz et al., 2023) to measure image-level similarity (higher is better). The details about image quality metrics are provided in the Appendix A.4.

Table 3: Ablation on textual conditions and SD adaptation strategies for synthetic replay in CaRD. "FT SD" denotes the use of LoRA-adapted Stable Diffusion. "Class name & template" refers to class name with the CLIP-style template , while "LLM-aug. class desc." corresponds to descriptions generated using the method proposed by  Yang et al. (2025). The "No replay and "Real replay" represents training with only current-task data and replaying stored past images, respectively.

| | Textual Condition | FT SD | Continual Learning Metrics | | | | | Image Quality Metrics | | |
|---|---|---|---|---|---|---|---|---|---|---|
| | | | Trans.↑ | Avg.↑ | Last↑ | BWT↑ | AF↓ | FID↓ | CLIP-I↑ | DINO-I↑ |
| 1 | No replay | – | 69.6 | 77.0 | 85.9 | -1.99 | 2.02 | – | – | – |
| 2 | Real replay | – | 70.5 | 78.9 | 88.3 | **-0.55** | **0.60** | – | – | – |
| 3 | Class name & template | | 70.1 | 78.0 | 87.1 | -0.95 | 0.98 | 137.02 | 0.52 | 0.48 |
| 4 | LLM-aug. class desc. | | 70.8 | 78.4 | 87.4 | -0.98 | 1.10 | 137.76 | 0.58 | 0.50 |
| 5 | Instance-level Caption | | 71.1 | 79.1 | 88.0 | -0.75 | 0.80 | 124.94 | 0.66 | 0.58 |
| 6 | Class name & template | ✓ | 70.2 | 78.2 | 87.5 | -0.88 | 0.91 | 81.45 | 0.60 | 0.54 |
| 7 | LLM-aug. class desc. | ✓ | 71.0 | 78.8 | 87.7 | -0.77 | 0.79 | 81.03 | 0.62 | 0.59 |
| 8 | Instance-level Caption | ✓ | **71.2** | **79.2** | **88.3** | **-0.61** | **0.64** | **74.63** | **0.69** | **0.64** |

Table 4: Ablation of different models for synthetic data generation.

(a) Multi-Modal LLMs for generating captions

| Multi-Modal LLM | Params | Trans.↑ | Avg.↑ | Last↑ | BWT↑ | AF↓ |
|---|---|---|---|---|---|---|
| LLaVA Next | 7B | 70.8 | 78.9 | 87.8 | -0.86 | 0.88 |
| LLaMA-3.2 | 11B | 71.1 | 79.1 | 88.2 | -0.70 | 0.74 |
| QWen2.5-VL | 3B | 71.1 | 79.2 | 88.2 | -0.65 | 0.71 |
| QWen2.5-VL | 7B | **71.2** | **79.2** | **88.3** | **-0.61** | **0.64** |

(b) T2I models for synthesizing images.

| T2I Model | FT SD | Trans.↑ | Avg.↑ | Last↑ | BWT↑ | AF↓ |
|---|---|---|---|---|---|---|
| SD1.5 | | 70.8 | 79.0 | 87.6 | -0.89 | 0.97 |
| SD1.5 | ✓ | 71.0 | 79.1 | 88.0 | -0.80 | 0.83 |
| SD3-Medium | | 71.1 | 79.0 | 88.0 | -0.75 | 0.80 |
| SD3-Medium | ✓ | **71.2** | **79.2** | **88.3** | **-0.61** | **0.64** |

**Synthetic data generation from different models.**    We systematically evaluate the impact of the different models for synthetic data generation by ablating both the multi-modal LLM for captioning and the T2I model for image synthesis. As shown in Table 4a, our method is robust to the choice of MLLM, as its impact on final continual learning performance is minor, indicating a lack of dependency on any specific model. Among the evaluated models, QWen2.5-VL (7B) achieves the highest scores across all metrics. This robustness extends to the T2I model, as analyzed in Table 4b. Consistent with other results in Table 3, LoRA-based fine-tuning improves performance for both SD1.5 and SD3 architectures. Using a more powerful backbone such as SD3-Medium provides a slight additional increase, and the combination of a stronger T2I model with LoRA adaptation yields the best overall results.

### 4.4.2    DISTILLATION STRATEGIES

**Distillation method and position.**    Table 5 compares various distillation methods and the effect of aligning features at different network depths. Relying solely on logit-based knowledge distillation (KD, row 1) results in limited performance, with 69.4% Average accuracy and severe forgetting (BWT: -8.08). Applying the loss at the final encoder block (row 2) moderately improves Average to 73.9%, but still suffers from considerable forgetting. In contrast, restricting feature alignment to shallow layers (0–5, row 3) further degrades performance, underscoring the importance of deeper semantic alignment. Aligning all encoder blocks (0–11, row 5) significantly improves both accuracy and stability, reaching 79.0% Average and -0.83 BWT. Using only distribution-level Wasserstein alignment (Wass, row 6) also yields strong results, but our proposed DistKD (row 8), which jointly applies distribution-level and token-level alignment across all layers, achieves the best overall performance with 79.2% Average, 88.3% Last, and the lowest forgetting. Notably, while PODNet (row 7) also employs multi-layer feature alignment for CL, our reproduced PODNet under the CLIP-ViT yields substantially lower performance, indicating that pooled-feature alignment without explicit distribution-level modeling is insufficient to mitigate the multi-layer feature drift in VLM-based CL.

**Reference model selection.**    In Table 6, we ablate the choice of the reference model used for distribution alignment. "Initial" refers to the original pre-trained model. "Last" is the model from

Table 5: Ablation of distribution alignment strategies in CaRD. We compare different strategies, including traditional logit-based KD, token-wise MSE matching (Hint), distribution-level Wasserstein distance (Wass), PODNet (Douillard et al., 2020) and our proposed DistKD. Layers specifies which encoder blocks are targeted for feature alignment.

| | Distillation Strategies | Layers $\mathcal{S}$ | Trans.↑ | Avg.↑ | Last↑ | BWT↑ | AF↓ |
|---|---|---|---|---|---|---|---|
| 1 | KD | – | 57.4 | 69.4 | 80.9 | -8.08 | 9.77 |
| 2 | Hint | 11 | 64.3 | 73.9 | 83.1 | -6.91 | 7.82 |
| 3 | Hint | 0–5 | 55.0 | 66.8 | 82.4 | -7.45 | 10.75 |
| 4 | Hint | 6–11 | 71.2 | 78.9 | 87.8 | -0.82 | 0.90 |
| 5 | Hint | 0–11 | 71.0 | 79.0 | 87.8 | -0.83 | 0.92 |
| 6 | Wass | 0–11 | 70.6 | 78.4 | 87.5 | -0.72 | 0.82 |
| 7 | PODNet | 0–11 | 69.5 | 77.1 | 86.0 | -4.22 | 4.82 |
| 8 | DistKD | 0–11 | **71.2** | **79.2** | **88.3** | **-0.61** | **0.64** |

Table 6: Ablation on different reference model for distillation.

| Reference Model | Trans.↑ | Avg.↑ | Last↑ | BWT↑ | AF↓ |
|---|---|---|---|---|---|
| Initial | 71.1 | 69.2 | 79.3 | -10.56 | 10.61 |
| Last | **71.2** | **79.2** | **88.3** | **-0.61** | **0.64** |
| WISE(0.5) | 70.5 | 75.9 | 82.0 | -6.95 | 7.02 |
| WISE(0.8) | 70.4 | 77.4 | 85.2 | -3.43 | 3.45 |

Table 7: Ablation on different fine-tuning strategies of VLM.

| Tuning Strategies | Trans.↑ | Avg.↑ | Last↑ | BWT↑ | AF↓ |
|---|---|---|---|---|---|
| Visual-only | 69.4 | 76.1 | 84.4 | -1.20 | 1.35 |
| Text-only | 70.6 | 78.4 | 87.2 | -1.04 | 1.10 |
| Full | **71.2** | **79.2** | **88.3** | **-0.61** | **0.64** |

the last task. And "WISE($\alpha$)" is a weighted ensemble of the two (Wortsman et al., 2022), with $\alpha$ being the interpolation coefficient. Using the Initial model throughout training leads to suboptimal results and substantial forgetting. In contrast, using the model from last task as the reference proves to be the optimal strategy, achieving Last accuracy of 88.3%, and nearly eliminates forgetting. While the WISE strategies attempt to balance stability and plasticity by interpolating between the initial and previous models, they fail to match the effectiveness of using the last model alone.

**Fine-tuning strategies of VLM.** In Table 7, we compare different fine-tuning strategies of VLMs, including updating only the visual encoder (Visual-only), only the text encoder (Text-only), or both encoders jointly (Full). The results reveal that fine-tuning the text encoder alone yields a substantial improvement over updating only the visual encoder. Joint fine-tuning of both encoders (Full) achieves the highest scores across all metrics and most effectively mitigates catastrophic forgetting, as indicated by the best BWT and AF values.

### 4.4.3 Ablation on Cost Analysis of Replay Data

Table 8 presents a comprehensive analysis of replay strategies, comparing their computational overhead, storage requirements, and continual learning performance. For generative replay methods, we specify the employed Multi-modal LLM and T2I models. "Replay Total Time" aggregates the average per-task offline costs for caption generation, LoRA fine-tuning, and synthetic image generation, while "VLM Training Cost" measures the CLIP training duration. "Total Time" denotes the combined per-task duration of offline replay construction and CLIP training, measured on 2xRTX 4090 GPUs; parenthetical multipliers indicate costs relative to the GIFT. "Storage Cost" quantifies the incremental memory per task for LoRA parameters and caption buffers or real images.

In the full-shot setting, caption generation with QWen2.5-VL (7B) (13.6 min) incurs a cost comparable to image synthesis (11.6–12.4 min). In contrast, LoRA fine-tuning of T2I remains lightweight (5.1–8.2 min). This efficiency is further pronounced in the 5-shot scenario, where captioning time becomes negligible (0.5 min) relative to fine-tuning (8.2 min) and generation (12.4 min). Moreover, using two smaller Multi-modal LLMs (SmolVLM (0.5B) (Marafioti et al., 2025) and InternVL3.5 (1B) (Wang et al., 2025b)), paired with SD1.5, can further reduce replay cost while maintaining robust performance. Overall, the total cost of CaRD remains comparable to other methods.

### 4.4.4 Comparison under matched generation model

Table 9 provides a fair comparison under a matched generation model setting for the full MTIL benchmark. Under this controlled setup, CaRD consistently outperforms all competing approaches. When all methods employ the same generator combination, i.e., QWen2.5-VL (7B) + SD3-M (2B), CaRD achieves the strongest performance across all metrics. Moreover, even when the generator capacity used by CaRD is reduced, such as replacing QWen2.5-VL (7B) with 3B, or substituting SD3-M (2B) with SD1.5 (0.9B), CaRD still surpasses other methods that retain the larger generators.

Table 8: Comprehensive comparison of **OFFLINE** replay cost, VLM training cost, and storage cost across different replay strategies on the MTIL benchmark. All results are averaged over the full sequence of 11 tasks. For all generative replay methods shown in the table (including both Class name & Template (C&T) and Instance-level Caption), the replay samples are constructed once during an **OFFLINE** pre-processing stage before CLIP training begins, and therefore do not affect CLIP training or inference cost. This process does not store any real data and therefore introduces no privacy or storage risks. All synthesized images are discarded after CLIP training, and therefore incur no additional long-term storage cost. SD3-M: SD3-Medium.

| Method | OFFLINE Replay Cost | | | | | | | VLM Training Cost | | | Total Time (min) | Storage Cost (MB) | Trans. / Avg. / Last |
|---|---|---|---|---|---|---|---|---|---|---|---|---|---|
| | Multi-Modal LLM (#Params) | Caption Gen. (min) | T2I Model (#Params) | FT T2I | FT (min) | Image Gen. (min) | Replay Total Time (min) | CLIP Model (#Params) | Distill. Method | Training (min) | | | |
| *full-shot setting* | | | | | | | | | | | | | |
| GIFT (C&T) | – | – | SD1.5 (0.9B) | – | – | 11.6 | 11.6 | ViT-B/16 (0.15B) | KD | 6.2 | 17.8 (×1.0) | 0.2 | 69.3 / 77.3 / 86.0 |
| LoRA-Loop (C&T) | – | – | (0.9B) | ✓ | 5.1 | 11.6 | 16.7 | | KD | 6.2 | 22.9 (×1.3) | 30.2 | 69.8 / 77.6 / 86.0 |
| LoRA-Loop (C&T) | – | – | SD3-M (2B) | ✓ | 8.2 | 12.4 | 20.6 | | KD | 6.2 | 26.8 (×1.5) | 47.2 | 70.0 / 77.7 / 86.1 |
| LoRA-Loop (C&T) | – | – | | ✓ | 8.2 | 12.4 | 20.6 | | DistKD | 6.7 | 27.3 (×1.5) | 47.2 | 70.2 / 78.2 / 87.5 |
| CaRD (Instance-level Caption) | SmolVLM (0.5B) | 1.8 | SD1.5 (0.9B) | ✓ | 5.1 | 11.6 | 18.5 | ViT-B/16 (0.15B) | DistKD | 6.7 | 25.2 (×1.4) | 34.1 | 70.6 / 78.6 / 87.5 |
| | InternVL3.5 (1B) | 3.9 | SD1.5 (0.9B) | ✓ | 5.1 | 11.6 | 20.6 | | | 6.7 | 27.3 (×1.5) | 34.2 | 70.8 / 78.9 / 87.9 |
| | QWen2.5-VL (3B) | 8.4 | SD3-M (2B) | ✓ | 8.2 | 12.4 | 29.0 | | | 6.7 | 35.7 (×2.0) | 49.2 | 71.1 / 79.2 / 88.2 |
| | QWen2.5-VL (7B) | 13.6 | SD1.5 (0.9B) | ✓ | 5.1 | 11.6 | 30.3 | | | 6.7 | 37.0 (×2.1) | 34.2 | 71.0 / 79.1 / 88.0 |
| | QWen2.5-VL (7B) | 13.6 | SD3-M (2B) | ✓ | 8.2 | 12.4 | 34.2 | | | 6.7 | 40.9 (×2.3) | 49.2 | 71.2 / 79.2 / 88.3 |
| No replay | – | – | – | – | – | – | – | ViT-B/16 (0.15B) | DistKD | 6.3 | 6.3 | – | 69.6 / 77.0 / 85.9 |
| Real replay | – | – | – | – | – | – | – | | | 6.7 | 6.7 | 587.1 | 70.5 / 78.9 / 88.3 |
| *5-shot setting* | | | | | | | | | | | | | |
| GIFT (C&T) | – | – | SD3-M (2B) | – | – | 12.4 | 12.4 | ViT-B/16 (0.15B) | KD | 6.2 | 18.6 (×1.0) | 0.2 | 69.0 / 72.5 / 77.8 |
| LoRA-Loop (C&T) | – | – | | ✓ | 8.2 | 12.4 | 20.6 | | KD | 6.2 | 26.8 (×1.4) | 47.2 | 69.2 / 72.9 / 78.0 |
| CaRD (Instance-level Caption) | QWen2.5-VL (7B) | 0.5 | SD3-M (2B) | ✓ | 8.2 | 12.4 | 21.1 | ViT-B/16 (0.15B) | DistKD | 6.7 | 27.8 (×1.5) | 47.6 | 70.4 / 73.2 / 79.2 |

Table 9: Fair comparison under matched generation model on the full setting of MTIL. Within each panel, all methods use identical generation models, ensuring that any observed performance differences stem from the replay mechanism, rather than from generator capacity. *: Our reproduction.

| Method | Replay data | Generation Model | FT T2I | Trans.↑ | Avg.↑ | Last↑ | BWT↑ | AF↓ |
|---|---|---|---|---|---|---|---|---|
| GIFT | Class name & template | SD1.5 (0.9B) | | 69.3 | 77.3 | 86.0 | – | – |
| LoRA-Loop | | | ✓ | 69.8 | 77.6 | 86.0 | – | – |
| GIFT* | Class name & template | SD3-M (2B) | | 69.7 | 77.2 | 85.7 | -2.01 | 2.53 |
| LoRA-Loop* | | | ✓ | 70.0 | 77.7 | 86.1 | -1.92 | 2.31 |
| GIFT* + Caption | Instance-level Caption | QWen2.5-VL (7B) + SD3-M (2B) | ✓ | 69.7 | 77.6 | 86.3 | -1.59 | 1.67 |
| LoRA-Loop* + Caption | | | ✓ | 70.2 | 78.3 | 86.9 | -1.47 | 1.39 |
| **CaRD** | | | ✓ | **71.2** | **79.2** | **88.3** | **-0.61** | **0.64** |
| **CaRD** | Instance-level Caption | QWen2.5-VL (3B) + SD3-M (2B) | ✓ | 71.1 | 79.2 | 88.2 | -0.65 | 0.71 |
| **CaRD** | Instance-level Caption | QWen2.5-VL (7B) + SD3-M (0.9B) | ✓ | 71.0 | 79.1 | 88.0 | -0.80 | 0.83 |

This further confirms that CaRD's effectiveness arises from its caption-guided replay and DistKD design rather than from reliance on large auxiliary models.

## 5 CONCLUSION

Conventional generative replay for continual learning with VLMs remains limited by its reliance on simple class-level prompts, which results in a pronounced fidelity gap between synthetic and real data. We propose a caption-guided paradigm that stores instance-level captions generated by Multi-modal LLM and reconstructs past images using LoRA-adapted T2I model, enabling high-fidelity and instance-aware synthetic replay with low storage overhead. We further analyze the feature drift phenomenon, where continual fine-tuning induces pronounced changes in intermediate features that logit distillation only partially alleviates. To address this, we introduce DistKD, a distribution-based distillation method that explicitly aligns feature distributions at multiple intermediate layers. By integrating these two components, CaRD achieves state-of-the-art performance on standard benchmarks and consistently outperforms previous replay and distillation approaches. We refer to Appendix H for detailed discussion on limitations and future research.

## REPRODUCIBILITY STATEMENT

To ensure reproducibility, we provide detailed descriptions of the datasets and implementation in Section 4.1 and Section 4.2, including the exact task order for the 11 MTIL datasets, the calculation of all evaluation metrics, and the hyperparameters used in our method. All experiments in this paper are conducted on a local workstation equipped with two NVIDIA GeForce RTX 4090 GPUs (24GB memory each), an Intel Core i9-13900K CPU, and 64GB of DDR5 RAM, running Ubuntu 22.04 LTS. All implementations are developed using the PyTorch framework, with CUDA 12.1 and cuDNN 8.9 for acceleration. We will release source code and checkpoints upon acceptance.

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

APPENDIX

# A DETAILS OF EXPERIMENTAL SETTINGS

## A.1 DATASETS

We mainly evaluate CaRD on the MTIL and X-TAIL. MTIL (Zheng et al., 2023) is multi-domain task-incremental learning benchmark, which is specifically designed for VLMs and comprises 11 heterogeneous image classification datasets from diverse domains, including Aircraft Maji et al. (2013), Caltech101 Fei-Fei et al. (2004), CIFAR100 Krizhevsky (2009), DTD Cimpoi et al. (2014), EuroSAT Helber et al. (2019), Flowers Nilsback & Zisserman (2008), Food Bossard et al. (2014), MNIST LeCun et al. (1998), OxfordPet Parkhi et al. (2012), StanfordCars Krause et al. (2013) and SUN397 Xiao et al. (2010). We adhere to the established protocols, which include both full-shot and 5-shot settings, the latter involving training on only five labeled examples per class (Yu et al., 2024a). The evaluation is performed across two standard task sequences; the first one is alphabet order (Order-I): Aircraft, Caltech101, CIFAR100, DTD, EuroSAT, Flowers, Food, MNIST, OxfordPet, StanfordCars, SUN397. And the second one is a random order (Order-II): StanfordCars, Food, MNIST, OxfordPet, Flowers, SUN397, Aircraft, Caltech101, DTD, EuroSAT, CIFAR100.

The X-TAIL (Xu et al., 2024) benchmark extends MTIL (Zheng et al., 2023) for cross-domain task-agnostic incremental learning, where the model is evaluated on all tasks without being provided the task identity at inference time. In its construction, X-TAIL builds upon MTIL but removes the CIFAR100 (Krizhevsky, 2009) dataset to avoid significant class overlap with other datasets. The benchmark provides two evaluation settings, full-shot and 16-shot, and follows two standard task orders. The first one is alphabet order (Order-I): Aircraft, Caltech101, DTD, EuroSAT, Flowers, Food, MNIST, OxfordPet, StanfordCars, SUN397. And the second one is a random order (Order-II): StanfordCars, Aircraft, OxfordPet, Food, SUN397, MNIST, Flowers, DTD, Caltech101, EuroSAT. Consistent with MTIL, performance is reported using the Transfer, Average, and Last metrics.

## A.2 IMPLEMENTATION DETAILS

**Instance-level Captions.** To generate instance-level captions, we construct a prompt for the Multi-modal LLM QWen2.5-VL as follow:

```
The main object in this image is called {classname}, which is
{dataset_specific_attribute}.  Create detailed and engaging
descriptions of the {classname} in one sentence, while must
include the {classname} in the description, aiming to keep the
content concise and limited to 70 words in English.
```

The prompt use a combination of the class name and dataset-specific attribute. These attributes, detailed in Table 10, provide domain-specific priors that help the Multi-modal LLM better contextualize the image content. To ensure compatibility with the 77-token input limit of CLIP's text encoder, we also constrain the caption length. During generation, we set the temperature to 0.7 and the nucleus sampling (top-p) threshold to 0.9. In a post-processing step, we use regular expressions to filter the generated captions, discarding any that do not explicitly contain the target class name or include non-English text.

**Synthetic Replay Data.** Following the generation of captions for each task, we store them in a caption buffer. Concurrently, these generated captions are paired with their original corresponding images to fine-tune the SD3-Medium (Esser et al., 2024) using Low-Rank Adaptation (LoRA) (Hu et al., 2022), thereby creating a set of task-specific LoRA weights. Our implementation follows the standard practice[1] of integrating LoRA modules into the query, key, value, and output projection layers of each attention block within the SD3-Medium's MMDiT architecture, with the rank set to 4 and a scaling factor of 4. The fine-tuning process runs for 1,000 iterations using the AdamW Loshchilov & Hutter (2019) optimizer with a weight decay of 1e-4 and a batch size of 8. We employ a base learning rate of 1e-4, which includes a 100-iteration linear warmup to 4e-4, followed by a cosine decay schedule. During training, all images are resized to a 512x512 resolution.

---

[1]https://github.com/huggingface/diffusers/blob/main/examples/dreambooth/train_dreambooth_lora_sd3.py

Table 10: Dataset-specific attribute for different datasets.

| Dataset | Attribute |
|---|---|
| Aircraft | a type of an aircraft |
| Caltech101 | a type of a common object or human face |
| CIFAR100 | a type of object |
| DTD | a type of texture image |
| EuroSAT | a type of satellite image |
| Flowers | a type of flower |
| Food | a type of food |
| MNIST | a type of handwritten digit |
| OxfordPet | a type of pet (specific either a dog or a cat) |
| StanfordCars | a type of a car |
| SUN397 | a certain scene |
| ImageNet | a type of object |

During the image generation phase, captions are randomly sampled from the caption buffer. For each caption corresponding to a past task, we load its associated task-specific LoRA weights into the SD3-Medium model to guide the synthesis. A special case is made for the initial task to preserve the model's pre-trained zero-shot capabilities. Following the methodology of Yang et al. (2025), we use LLMs to expand the 1,000 ImageNet class names into the class-level descriptions. These serve as prompts for the pre-trained SD3-Medium model to generate a synthetic dataset that approximates the original CLIP pre-training data. For all synthesis, we use a classifier-free guidance scale of 5.0 and 28 denoising steps. By default, we generate 2,000 images to serve as the replay data for each continual learning task.

**Continual Learning for CLIP.** During the continual learning phase for the CLIP model, we use a batch size of 64, training for 1,000 iterations in the full-shot setting and 500 iterations in the 5-shot setting. Following the ZSCL protocol, we employ the AdamW optimizer ($\beta_1$=0.9, $\beta_2$=0.999) and apply label smoothing with a factor of 0.2. The learning rate undergoes a linear warmup to 1e-4 over the initial 100 iterations, followed by a cosine decay schedule for the remainder of training. The primary objective is a cross-entropy loss computed on the current task's data, based on the cosine similarity between the visual encoder's [CLS] token and the text encoder's [EOS] token. Concurrently, the synthetic replay data is utilized in the DistKD process. The weight for the DistKD loss is set to 100 by default.

**Implementation details for X-TAIL benchmark.** The implementation details and procedures for replay data generation are identical to those described for MTIL. During continual learning on CLIP, all training hyperparameters are kept consistent with the MTIL setting. For the full-shot and 16-shot scenarios, we train for 1000 and 500 iterations, respectively. During inference, since the task ID is not available, we first use the model trained on the current task to make predictions over the entire label space. Because the task information is available during training, the label space can be partitioned into seen and unseen categories, followed by Xu et al. (2024). If the predicted label falls into the unseen category space, we directly use this result. If the prediction belongs to the seen categories, we perform task-level classification for the test image. Specifically, during continual training, we use the pre-trained CLIP model to store a prototype for each seen category, defined as the mean of the [CLS] tokens of all training images for that class. For each test image, we compute the cosine similarity between its feature representation and each category prototype, and assign the task ID corresponding to the category with the highest similarity. Finally, under the model trained for the current task, we re-compute the predicted label for the test image, restricting the label space to that of the identified task ID.

## A.3 EVALUATION METRICS AND NOTATIONS

**Performance matrix.** Following standard continual-learning protocols, suppose there are $N$ tasks in total. Upon completion of training on task $T_i$, we evaluate the model on the test set of each task $T_j$ ($j = 0, 1, \ldots, N-1$) and record the results in a performance matrix $A \in \mathbb{R}^{N \times N}$, where $A_{i,j}$ denotes the accuracy on task $T_j$ after training on $T_i$. In conventional continual learning scenarios, only the lower triangular part of the matrix, denoted as $\mathrm{tril}(A) = \{A_{i,j} \mid i \geq j\}$, is considered meaningful, since models cannot be evaluated on future, unseen tasks ($i < j$). However, for continual learning with VLMs, particularly on benchmarks like MTIL, the upper triangular part $\mathrm{triu}(A) = \{A_{i,j} \mid$

$i < j\}$ becomes highly informative. It directly measures the degradation of the model's zero-shot capabilities on subsequent tasks, thus serving as a crucial indicator of how much pre-trained knowledge is forgotten.

On the MTIL (Zheng et al., 2023) benchmark, the performance of various methods is evaluated using three key metrics: Transfer, Average, and Last, which are calculated as follows:

**Transfer (Trans).** Transfer measures zero-shot transfer, i.e., how well knowledge acquired from earlier tasks generalizes to unseen ones. It is defined as the average accuracy on each task *before* it is learned, i.e., the mean of the strictly upper triangular part of $A$:

$$\text{Transfer} = \frac{1}{|\operatorname{triu}(A)|} \sum_{i=1}^{N} \sum_{j=i+1}^{N} A_{i,j}, \tag{5}$$

where $\operatorname{triu}(A) = \{A_{i,j} \mid i < j\}$ and $|\operatorname{triu}(A)| = \frac{N(N-1)}{2}$ is the number of elements above the main diagonal.

**Average accuracy (Avg).** Average accuracy summarizes performance across all tasks and all training stages. It corresponds to the mean of the entire matrix $A$:

$$\text{Avg} = \frac{1}{N^2} \sum_{i=1}^{N} \sum_{j=1}^{N} A_{i,j}. \tag{6}$$

**Last.** Last denotes the average retained performance after completing the final task, corresponding to the mean of the last row of $A$:

$$\text{Last} = \frac{1}{N} \sum_{j=1}^{N} A_{N,j}. \tag{7}$$

Additionally, to better assess how effectively each method mitigates forgetting, we introduce two metrics that explicitly measure this phenomenon: Backward Transfer (BWT) (Lopez-Paz & Ranzato, 2017) and Average Forgetting (AF) (Yu et al., 2024b; Qian et al., 2023), calculated as follows:

**Backward transfer (BWT).** Backward transfer quantifies the average influence of learning later tasks on earlier ones by comparing diagonal elements and the final row:

$$\text{BWT} = \frac{1}{N-1} \sum_{j=1}^{N-1} \left( A_{N,j} - A_{j,j} \right). \tag{8}$$

As models typically exhibit some degree of forgetting, BWT values are generally negative. Consequently, a higher BWT score (closer to zero) signifies a lower level of forgetting.

**Average forgetting (AF).** Similar to BWT, AF is a direct measure of forgetting. For each task $j < N$, define the forgetting after all tasks as the drop from its best historical accuracy to its final accuracy:

$$f_N^j = \max_{1 \le i < N} A_{i,j} - A_{N,j}. \tag{9}$$

The average forgetting is then given by

$$\text{AF} = \frac{1}{N-1} \sum_{j=1}^{N-1} f_N^j, \tag{10}$$

By construction, the per-task forgetting metric $f_N^j$ is generally non-negative, hence $\text{AF} \ge 0$. A lower AF score (closer to zero) therefore signifies a lower level of forgetting.

### A.4 IMAGE QUALITY METRIC

To quantitatively assess the fidelity between synthetic and real images, we employ three distinct metrics, each capturing a different aspect of image quality: FID for distributional similarity, and

CLIP-I and DINO-I for image-level similarity. For each dataset, we calculate these metrics by randomly sampling five real images and their five corresponding synthetic counterparts per class.

**FID** (Fréchet Inception Distance) (Heusel et al., 2017) is a standard metric for evaluating the quality of generated images. It measures the distance between the feature distributions of a set of real images and a set of synthetic images. Features are extracted from a Inception-v3 network, which pre-trained on ImageNet-1K. The metric models the features of both sets as multivariate Gaussian distributions and calculates the Fréchet distance (also known as Wasserstein distance) between them. A lower FID score indicates a smaller distance between the two distributions, suggesting higher fidelity in the generated images.

**CLIP-I** (CLIP Image Similarity) (Ruiz et al., 2023) quantifies the semantic similarity between pairs of images by leveraging the powerful image encoder of a pre-trained CLIP model. In our experiments, we utilize CLIP-ViT-H-14 pretrained on DFN5B [2] as the feature extractor. Specifically, the metric is calculated as the cosine similarity between the feature representations of a real image and its synthetic counterpart. A higher CLIP-I score indicates stronger semantic alignment, suggesting that the synthetic image effectively preserves the high-level content and concepts of the real image.

**DINO-I** (DINO Image Similarity) (Ruiz et al., 2023) provides a complementary metric to CLIP-I by utilizing features extracted from DINO, a vision transformer model pre-trained via self-supervised training. In our evaluation, we adopt DINOv2-giant [3] as the feature extractor, and compute the average pairwise cosine similarity between the embeddings of synthetic and real images. Unlike supervised networks, DINOv2 is not explicitly trained to ignore differences between subjects of the same class. Instead, its self-supervised objective encourages sensitivity to unique features of each image. Consequently, DINO-I is well-suited for assessing the structural fidelity and form consistency of generated images. Higher DINO-I scores indicate that synthetic images more faithfully preserve the structure and mid-level visual patterns of their real counterparts.

## B  ANALYSIS OF MULTI-LAYER FEATURE DRIFT AND THE DESIGN OF DISTILLATION

Our intention is not to claim that the use of Wasserstein statistics is itself novel—indeed, prior work such as Lv et al. (2024) has indeed explored distribution-based distillation for classic image classification. Rather, the contribution of DistKD lies in two aspects:

**(1) A systematic observation and characterization of a previously unreported failure mode specific to VLM-based continual learning.**

Our analysis (Figure 1b, Appendix Figure 9) reveals a multi-layer, multi-token drift phenomenon that emerges when CLIP is finetuned sequentially. **To our knowledge, this drift behavior has not been documented before in the distillation or continual learning literature.** Once this phenomenon was identified, it was natural and well-motivated to explore distribution-alignment tools to counteract the drift.

**(2) A complementary, multi-level distillation formulation tailored to the structure of VLMs.**

Wasserstein distance-based alignment is an effective and established choice for capturing global geometric of feature distributions. However, applying it directly is insufficient in the VLM setting: CLIP representations are tokenized, multi-layer, and multimodal; feature drift occurs inconsistently across layers and across modalities, and global alignment alone fails to preserve fine-grained token semantics. Therefore, DistKD is not presented as a new distillation primitive, but as a systematic, multi-level extension tailored to this drift phenomenon. It combines (i) global distribution alignment via Wasserstein distance-based alignment, and (ii) local, token-level semantic alignment via MSE, applied jointly to both the visual and text encoders across multiple layers.

This formulation directly follows from the characteristics of the drift. As shown in Table 5, applying only Wasserstein distance-based alignment or only Hint alignment fails to fully correct the drift, whereas the combined formulation yields significantly higher stability across tasks. Additional visualizations with Wass (Lv et al., 2024) and PODNet (Douillard et al., 2020) in Figure 3 indicate that

---

[2]https://huggingface.co/apple/DFN5B-CLIP-ViT-H-14/tree/main
[3]https://huggingface.co/facebook/dinov2-giant

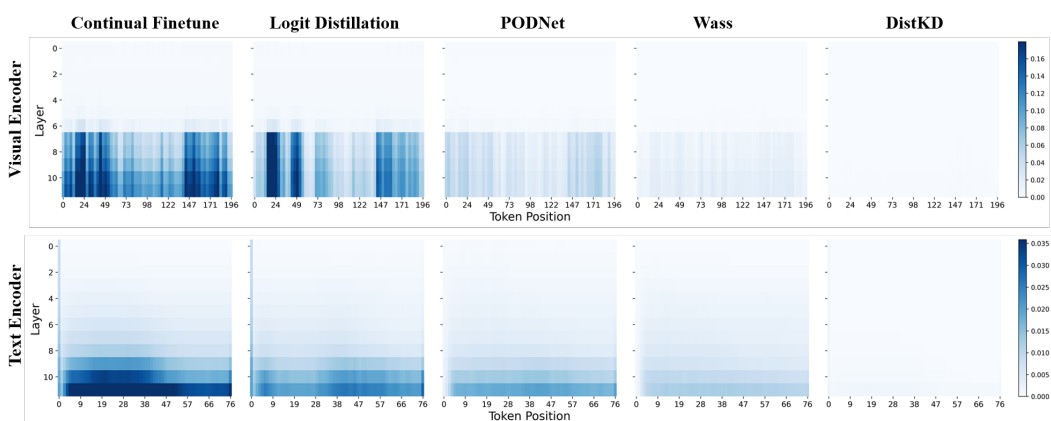

Figure 3: Visualization of multi-layer feature drift under different training strategies.

Wasserstein alignment alone cannot suppress the observed drift, highlighting that DistKD targets a continual-learning-specific challenge rather than reusing Wasserstein distance-based distillation (Wass).

## C EXPERIMENTAL RESULTS FOR FEW-SHOT SCENARIOS

We compare our approach against a broad range of state-of-the-art continual learning methods under both 5-shot settings on MTIL and 16-shot settings on X-TAIL, as reported in Table 11 and Table 12, respectively. Each table presents results for two task orderings (Order-I and Order-II) and evaluates Transfer (Trans.), Average (Avg.), and Last accuracy, along with their corresponding improvements (Δ) over the CLIP Zero-shot.

*For 5-shot setting on MTIL,* demonstrates clear superiority, yielding improvements of 0.8% and 1.9% in Average, and 2.0% and 2.3% in Last on Order-I and Order-II, respectively. These results validate that our caption-guided replay, combined with distribution-level knowledge distillation, effectively retains past knowledge while preserving the crucial zero-shot capabilities of VLMs. Additional detailed results are provided in the Appendix J.

*For 16-shot setting on X-TAIL,* CaRD again achieves the best overall performance. On Order-I, it improves *Transfer* by 4.3%, *Average* by 15.7%, and *Last* by 27.4% compared to CLIP zero-shot. For Order-II, the gains remain substantial, with increases of 1.8%, 13.7%, and 27.2% in *Transfer*, *Average*, and *Last*, respectively.

However, we have identified a factor that may limit our method's performance under the current evaluation protocol. The task-agnostic inference is performed over the combined class space of all datasets, *creating ambiguity when class names from different tasks overlap semantically.* For example, every class in the Aircraft dataset can be considered a sub-category of the "airplane" class in Caltech101. Consequently, when an image from Aircraft is classified as "airplane," it is penalized as an error, even though the prediction is semantically correct at a higher level of abstraction. This suggests that the development of a more sophisticated task-level classification mechanism, capable of resolving these hierarchical ambiguities, could further improve our method's performance.

## D MORE ABLATION STUDY

### D.1 DIFFERENT TEXTUAL CONDITIONS DURING FINE-TUNING AND GENERATION

In Table 3 of the main text, we maintained consistent textual conditions during both fine-tuning SD3-Medium and generating images. Here, we further analyze the impact of using different textual conditions in Table 13. Compared to the baseline approach where both fine-tuning and generation stages utilize "class name & template," employing Instance-level Caption solely during generation yields a significant improvement. This suggests that Instance-level Captions contain rich historical data information. However, the absence of instance-level textual information during the fine-tuning

Table 11: Comparison of SOTA methods on MTIL under the 5-shot setting for Order-I and Order-II. $\Delta$ represents the improvement over CLIP Zero-shot (%).

| Method | Order-I | | | | | | Order-II | | | | | |
|---|---|---|---|---|---|---|---|---|---|---|---|---|
| | Trans | $\Delta$ | Avg | $\Delta$ | Last | $\Delta$ | Trans | $\Delta$ | Avg | $\Delta$ | Last | $\Delta$ |
| CLIP Zero-shot | 69.4 | 0.0 | 65.3 | 0.0 | 65.3 | 0.0 | 65.4 | 0.0 | 65.3 | 0.0 | 65.3 | 0.0 |
| Continual Finetune | 51.2 | -18.2 | 58.5 | -6.8 | 67.1 | +1.8 | 49.2 | -16.2 | 49.0 | -16.3 | 42.8 | -22.5 |
| CLIP Full Finetune | – | – | – | – | 77.5 | +12.2 | – | – | – | – | 77.5 | +12.2 |
| LwF Li & Hoiem (2017) | 50.0 | -19.4 | 49.2 | -16.1 | 46.5 | -18.8 | 48.6 | -16.8 | 54.7 | -10.6 | 56.8 | -8.5 |
| LwF-VR Zhou et al. (2023) | 60.1 | -9.3 | 62.5 | -2.8 | 66.9 | +1.6 | 54.4 | -11.0 | 60.1 | -5.2 | 63.5 | -1.8 |
| ZSCL Zheng et al. (2023) | 65.3 | -4.1 | 64.4 | -0.9 | 67.4 | +2.1 | 62.8 | -2.6 | 67.6 | +2.3 | 71.8 | +6.5 |
| MoE-Adapter Yu et al. (2024a) | 68.9 | -0.5 | 71.4 | +6.1 | 76.1 | +10.8 | 64.7 | -0.7 | 69.5 | +4.2 | 75.7 | +10.4 |
| MulKI Zhang et al. (2024) | 69.5 | +0.1 | 72.4 | +7.1 | 77.2 | +11.9 | 65.2 | -0.2 | 71.4 | +6.1 | 76.7 | +11.4 |
| **CaRD (Ours)** | **70.4** | **+1.0** | **73.2** | **+7.9** | **79.2** | **+13.9** | **67.4** | **+2.0** | **73.3** | **+8.0** | **79.0** | **+13.7** |

Table 12: Comparison of SOTA methods on X-TAIL under the 16-shot setting for Order-I and Order-II. $\Delta$ represents the improvement over CLIP Zero-shot (%).

| Method | Order-I | | | | | | Order-II | | | | | |
|---|---|---|---|---|---|---|---|---|---|---|---|---|
| | Trans | $\Delta$ | Avg | $\Delta$ | Last | $\Delta$ | Trans | $\Delta$ | Avg | $\Delta$ | Last | $\Delta$ |
| CLIP Zero-shot | 57.7 | 0.0 | 57.7 | 0.0 | 57.7 | 0.0 | 57.7 | 0.0 | 57.7 | 0.0 | 57.7 | 0.0 |
| LwF Li & Hoiem (2017) | 47.7 | -10.0 | 53.2 | -4.5 | 62.8 | +5.1 | 46.9 | -10.8 | 52.0 | -5.7 | 55.4 | -2.3 |
| WiSE-FT Wortsman et al. (2022) | 52.3 | -5.4 | 54.2 | -3.5 | 58.0 | +0.3 | 51.1 | -6.6 | 56.9 | -0.8 | 61.5 | +3.8 |
| ZSCL Zheng et al. (2023) | 59.0 | +1.3 | 60.0 | +2.3 | 63.4 | +5.7 | 56.9 | -0.8 | 63.6 | +5.9 | 69.4 | +11.7 |
| MoE-Adapters Yu et al. (2024a) | 56.0 | -1.7 | 63.0 | +5.3 | 70.5 | +12.8 | 50.2 | -7.5 | 61.8 | +4.1 | 71.7 | +14.0 |
| Primal-RAIL Xu et al. (2024) | **62.4** | **+4.7** | 70.7 | +13.0 | 79.1 | +21.4 | 57.7 | 0.0 | 66.2 | +8.5 | 79.8 | +22.1 |
| Dual-RAIL Xu et al. (2024) | **62.4** | **+4.7** | 71.9 | +14.2 | 82.4 | +24.7 | 57.7 | 0.0 | 68.1 | +10.4 | 82.5 | +24.8 |
| LADA Luo et al. (2025) | 61.5 | +3.8 | 72.7 | +15.0 | 83.1 | +25.4 | 56.7 | -1.0 | 68.9 | +11.2 | 83.3 | +25.6 |
| **CaRD (Ours)** | 62.0 | +4.3 | **73.4** | **+15.7** | **85.1** | **+27.4** | 59.5 | +1.8 | **71.4** | **+13.7** | **84.9** | **+27.2** |

phase may prevent the SD model from explicitly distinguishing intra-class images, thereby limiting the quality of replay data. If Instance-level Captions are used only during fine-tuning while "class name & template" is retained for generation, the improvement over the baseline is marginal, indicating that introducing instance-level information during the generation phase is more critical. Ultimately, our proposed approach, which employs Instance-level Caption for both fine-tuning and generation, achieves the best performance across all metrics.

## D.2 HYPER-PARAMETERS OF DISTILLATION

**Number of Synthetic Images.** Figure 4 presents an ablation study on the number of synthetic images per task used for distillation. The results demonstrate that increasing the number of synthetic images is beneficial for distillation, with performance gradually rising as the number increases from 500 to 2000. Beyond 2000 images, further increases yield minimal gains. Therefore, we adopt 2000 synthetic images as the default setting.

**Distillation Loss Weight.** Figure 5 shows the effect of the distillation loss weight $\lambda$. The results indicate that the method's performance is stable across a wide range of weights from 20 to 200, which demonstrates our method's robustness to this hyperparameter. To balance the results across the three metrics, we select a weight of 100 as the default choice.

Table 13: Ablation study on different textual conditions for SD Training and SD Sampling.

| Textual Condition | | Trans.↑ | Avg.↑ | Last↑ | BWT↑ | AF↓ |
|---|---|---|---|---|---|---|
| SD Training | SD Sampling | | | | | |
| Class name & template | Class name & template | 70.1 | 78.2 | 87.5 | -0.90 | 0.97 |
| | Instance-level Caption | 70.7 | 78.6 | 87.9 | -0.77 | 0.76 |
| Instance-level Caption | Class name & template | 70.2 | 78.3 | 87.7 | -0.81 | 0.82 |
| | Instance-level Caption | **71.2** | **79.2** | **88.3** | **-0.61** | **0.64** |

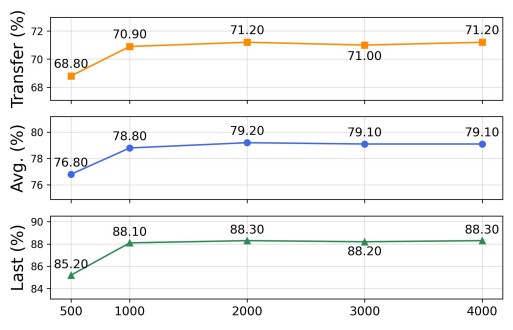 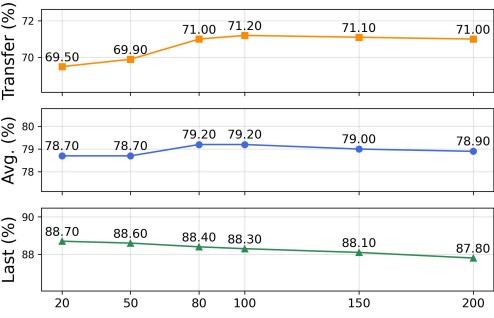

Figure 4: Different numbers of synthetic images     Figure 5: Different weight $\lambda$ of distillation loss

# E    INTEGRATING OTHER METHODS WITH OUR PROPOSED MODULES

**Integrating Caption-guided Replay and DistKD.**   In this section, we integrate our method with the classic method LwF (Li & Hoiem, 2017) and the state-of-the-art method GIFT (Wu et al., 2025), to demonstrate its generalizability. The standard LwF uses data from the current task for distillation and suffers from severe catastrophic forgetting. However, applying our proposed DistKD framework to it (LwF w/ DistKD) results in substantial improvements across all metrics, showcasing DistKD's effectiveness in mitigating forgetting.

We then apply our components to GIFT. The GIFT method uses "class name & template" to guide a pre-trained Stable Diffusion model to generate replay data, and it employs a contrastive distillation loss and Adaptive Weight Consolidation (AWC) to mitigate model forgetting. Specifically, we replace the data used for distillation in GIFT with our proposed instance-level captions, denoted as *GIFT w/ Instance-level caption*. Concurrently, we utilize GIFT's data but substitute its distillation and AWC mechanisms with our proposed DistKD, denoted as *GIFT w/ DistKD*. The results are presented in Table 14. As shown, compared to the data generated by GIFT, our instance-level data not only effectively improves the Last result but also significantly mitigates model forgetting. Furthermore, integrating DistKD with GIFT leads to substantial improvements across all metrics, effectively alleviating catastrophic forgetting.

Table 14: Integrate our proposed components with other methods. *: Reproduced by us.

| Method | Trans.↑ | Avg.↑ | Last↑ | BWT↑ | AF↓ |
|---|---|---|---|---|---|
| **LwF** | 56.0 | 64.4 | 75.5 | -11.73 | 26.43 |
| *LwF w/ DistKD* | 69.6 | 77.0 | 85.9 | -1.99 | 2.02 |
| **GIFT*** | 69.7 | 77.1 | 85.5 | -2.13 | 2.60 |
| *GIFT w/ Instance-level Caption* | 69.7 | 77.6 | 86.3 | -1.59 | 1.67 |
| *GIFT w/ DistKD* | 70.8 | 79.0 | 87.6 | -0.89 | 0.97 |

**Comparison of different textual augmentations.**   Data from the current task is typically used to compute the cross-entropy (CE) loss, allowing the model to learn new knowledge. However, previous continual learning methods have generally used only "class name & CLIP-style template" as input for the CLIP text encoder. Several existing works (Pratt et al., 2023; Menon & Vondrick, 2023; Yang et al., 2025) have demonstrated that using a Large Language Model (LLM) to expand each category into a richer class description is beneficial for CLIP-based zero-shot and few-shot tasks. Here, we adopt this concept and apply LLM-based textual augmentation to the continual learning setting.

For a fair comparison, we apply the same augmentation techniques to both our method, CaRD, and the state-of-the-art method, GIFT. The results are detailed in Table 15. For our proposed CaRD, using LLM-augmented class descriptions as textual input proves beneficial. Furthermore, integrating the original CLIP-style templates with LLM-augmented descriptions, as proposed by Yang et al. (2025), achieves the best performance. However, we observe that applying LLM-augmented descriptions to GIFT does not yield additional improvements. We hypothesize that this is because

GIFT uses simple template-based distillation data, while task data are supervised with cross-entropy loss on richer text, which may introduce a mismatch in the textual domain.

Table 15: Comparison of different textual augmentations on current task data. *: Reproduced by us.

| Method | Textual Input | Trans.↑ | Avg.↑ | Last↑ | BWT↑ | AF↓ |
|--------|---------------|---------|-------|-------|------|-----|
| **GIFT** | CLIP* | 69.7 | 77.1 | 85.5 | -2.13 | 2.60 |
| | *CuPL* (Pratt et al., 2023) | 69.5 | 77.1 | 85.6 | -1.76 | 1.88 |
| | *ImagineFSL* (Yang et al., 2025) | 69.6 | 77.2 | 85.6 | -1.92 | 1.98 |
| **CaRD** | CLIP | 69.6 | 78.2 | 87.6 | -0.96 | 0.99 |
| | *CuPL* (Pratt et al., 2023) | 70.8 | 78.9 | 88.0 | **-0.59** | **0.62** |
| | *ImagineFSL* (Yang et al., 2025) | **71.2** | **79.2** | **88.3** | -0.61 | 0.64 |

## F   MORE VISUALIZATION OF FORGETTING

### F.1   VISUALIZATION OF CAPTION-GUIDED REPLAY IMAGES

While Figure 2a in the main text provides a partial visualization of synthetic images, we offer a more detailed exhibition in Figures 6 and Figure 7, showcasing data from all 11 datasets in the MTIL benchmark. Due to space constraints, Figure 6 displays results from Aircraft to Eurosat, and Figure 7 presents results from Food to SUN397. Each row of images includes three components: the real image, a synthetic image generated by the pre-trained SD3-Medium guided by a "class name & template" prompt (along with that text), and a synthetic image generated by our proposed method using instance-level caption generated by Multi-modal LLM.

We observe that images generated with our instance-level caption approach more closely resemble the real images and capture more instance-specific characteristics. For example, in Caltech101, our generated "motorbike" maintains the color and structural properties consistent with the real image. Similarly, the "barrel" image includes text, color, and background information that matches the original. Furthermore, we observe that for datasets such as CIFAR100 and MNIST, our generated images better approximate the style of the real images, the relative blurriness of CIFAR100 due to its low resolution and the distinct handwritten style of MNIST. This is a benefit of fine-tuning the T2I model on the real data using LoRA, which allows it to preserve the distribution characteristics of the source data. By leveraging this high-quality synthetic replay data, our method is able to effectively mitigate catastrophic forgetting.

### F.2   VISUALIZATION OF FEATURE DRIFT

In Figure 2b of the main text, we exclusively showed the results for Task 9. Here, we present a comprehensive visualization of feature drift across all subsequent tasks in Figure 9. The first task is omitted, as our forgetting metrics only evaluate performance on previously learned tasks.

To generate these heatmaps, we use our proposed method to create replay data. We then calculate the Mean Squared Euclidean (MSE) distance for each token at every layer by comparing the model's state after learning the current task against its state after the previous task. The average MSE across all synthetic replay data is then visualized as a heatmap, where darker colors indicate a larger difference in token responses and thus more severe feature drift.

Each sub-figure for a given task contains a 2x3 grid of heatmaps. The two rows correspond to the visual and text backbones, respectively. The three columns compare three methods: continuous fine-tuning, logit distillation, and our proposed DistKD. Across all tasks, the results show that while logit distillation alleviates feature drift to some extent compared to continuous fine-tuning, it does not fully resolve the issue. In contrast, our DistKD method significantly mitigates feature drift, which is crucial for its effectiveness in preventing catastrophic forgetting.

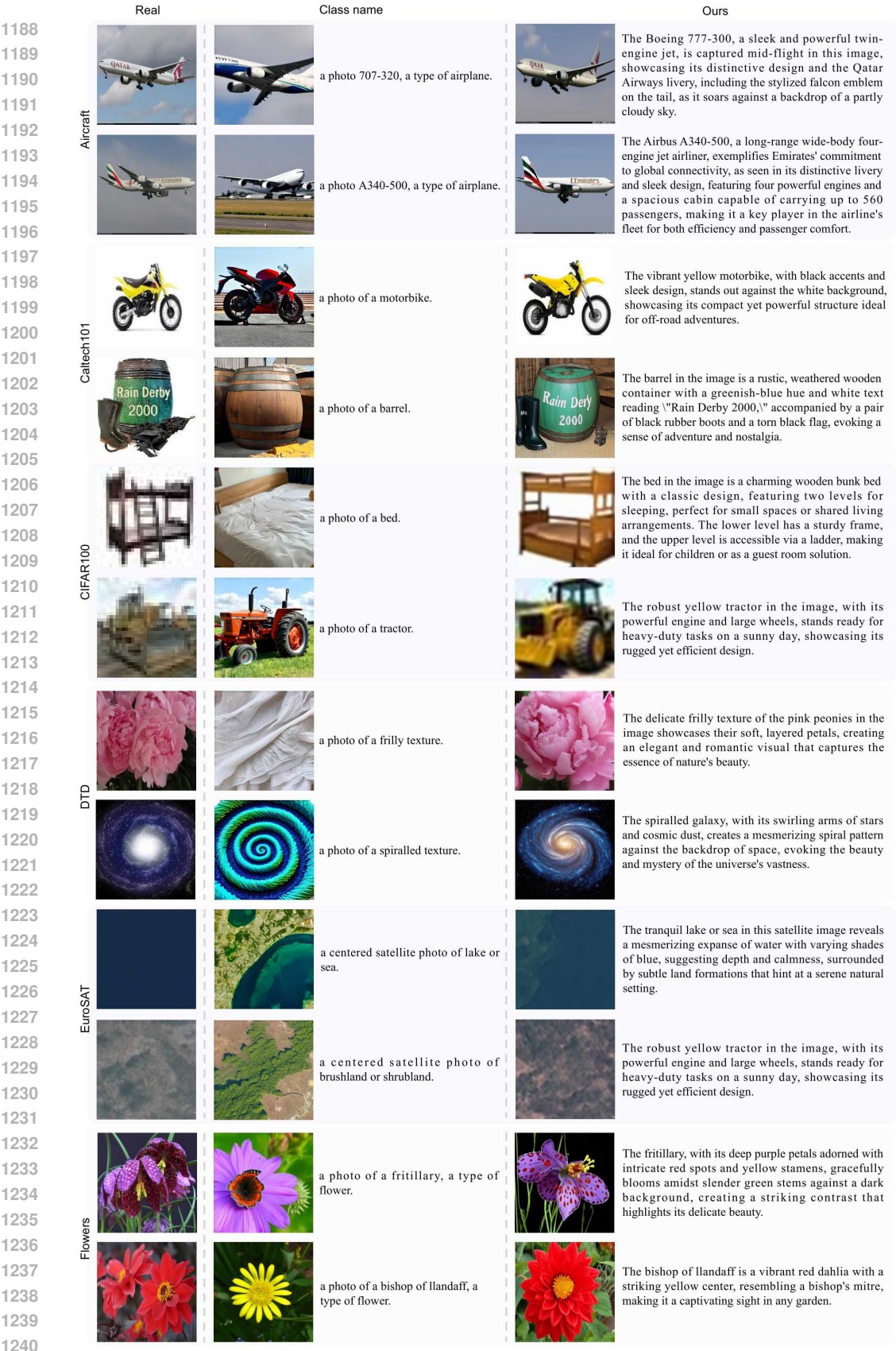

Figure 6: Visualization of synthetic data for different downstream datasets(Aircraft-EuroSAT).

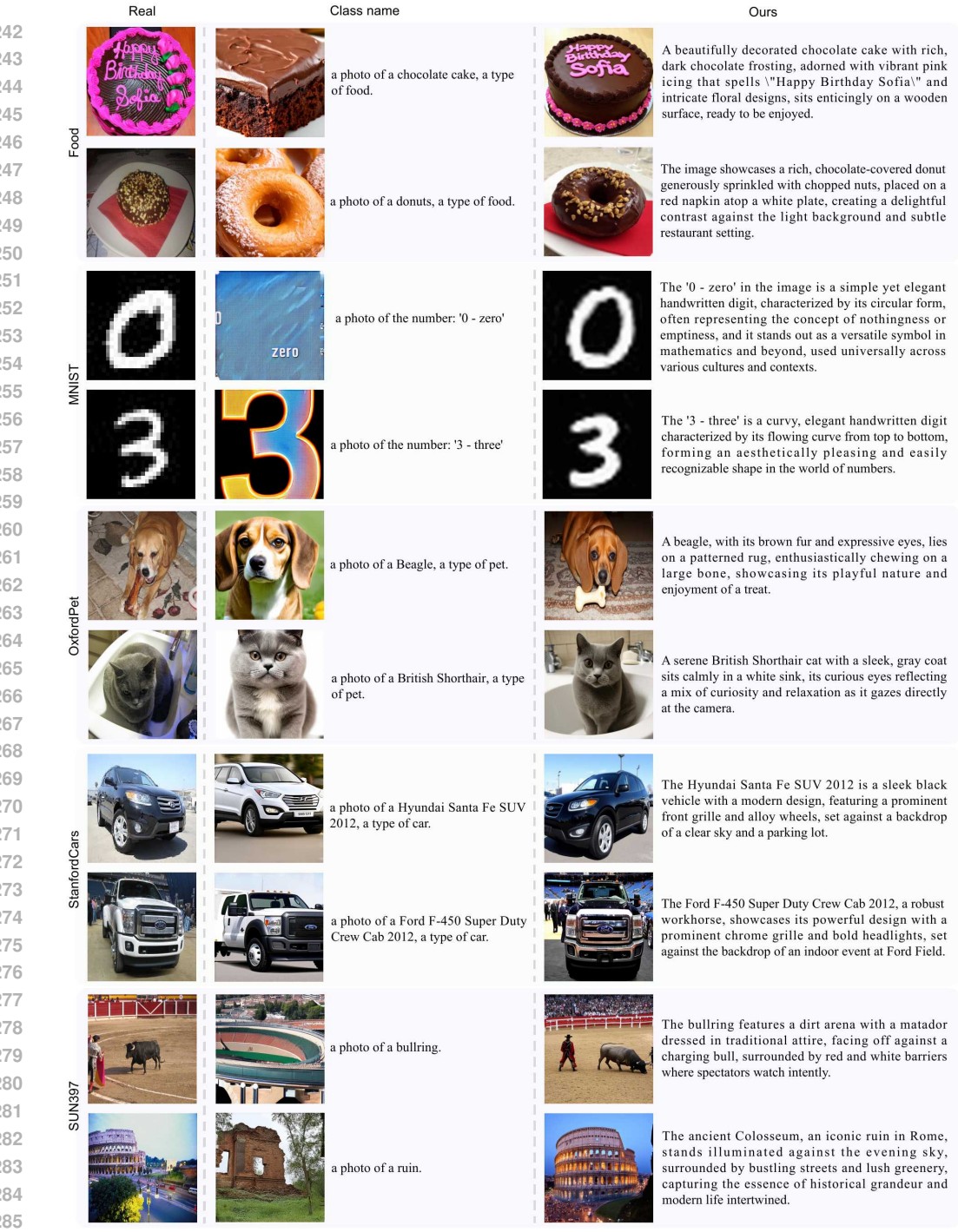

Figure 7: Visualization of synthetic data for different downstream datasets(Food-SUN397).

## G Extra experiment on Class Incremental Learning

In this section, we have extended our experiments and applied CaRD directly to the standard class-incremental learning (CIL) setup on fine-grained dataset CUB.

Following the protocol used in PROOF (Zhou et al., 2025b), we divide the classes within the same dataset into sequential tasks. The model receives new classes task by task and is evaluated on all previously seen classes after each stage. "B m" denotes the number of base classes, and "Inc n" indicates the number of incremental classes per task (m = 0 means that all classes are evenly split

Table 16: Average and last performance comparison of different methods on CUB under class-incremental learning setting.

| Replay | Method | CUB | | | |
|---|---|---|---|---|---|
| | | B0 Inc20 | | B100 Inc20 | |
| | | $\bar{\mathcal{A}}$ | $\mathcal{A}_B$ | $\bar{\mathcal{A}}$ | $\mathcal{A}_B$ |
| No Replay | Finetune | 2.06 | 0.64 | 0.56 | 0.47 |
| | SimpleCIL (Zhou et al., 2025a) | 83.81 | 77.52 | 79.75 | 77.52 |
| | RAPF (Huang et al., 2024) | 83.04 | 76.34 | – | – |
| | ZS-CLIP (Radford et al., 2021) | 74.38 | 63.06 | 67.96 | 63.06 |
| Real Replay | iCaRL (Rebuffi et al., 2017) | 82.04 | 74.74 | 78.57 | 75.07 |
| | MEMO (Zhou et al., 2022) | 77.32 | 65.69 | 72.88 | 66.41 |
| | PROOF (Zhou et al., 2025b) | 84.93 | 79.43 | 81.67 | 79.18 |
| | CaRD (Ours) | 85.41 | 79.82 | 82.28 | 79.52 |
| Generative Replay | CaRD (Ours) | 83.44 | 77.55 | 79.81 | 77.20 |
| Generative Replay + Real Replay | CaRD (Ours) | **86.22** | **80.64** | **83.13** | **80.01** |

across tasks). All method within real replay use 20 real samples per class as replay data, and CaRD uses 40 synthetic images for generative replay in all settings.

We compare CaRD against other CIL methods specifically designed for this setting, and Table 16 reports the full results, including both the average accuracy $\bar{\mathcal{A}}$ and the final-stage accuracy $\mathcal{A}_B$. The results show that CaRD equipped only with our caption-guided generative replay achieves performance on par with real-replay methods such as PROOF under two settings. Moreover, when combined with real replay, CaRD further surpasses the performance of the real-replay method alone.

Finally, regarding the concern about whether the T2I model can retain preserve the subtle, class-specific attributes, we provide additional qualitative visualizations in Figure 8. These results indicate that the images reconstructed via our caption-guided generation strategy maintain consistency with the real samples.

## H  LIMITATIONS AND FUTURE RESEARCH

Compared to approaches that generate replay data from pre-trained T2I models using only class names and templates, our caption-guided replay paradigm introduces slightly higher storage cost, mainly due to the LoRA fine-tuning of the T2I model. However, as shown in Table 8, this overhead remains moderate and may be further reduced in the future by leveraging more advanced parameter-efficient adaptation techniques. In addition, the quality of caption-guided replay is expected to further improve as the capabilities of multi-modal LLMs and T2I models continue to grow. Our approach also has the potential to be extended beyond image domains, such as to continual learning in video understanding.

We also observe the phenomenon of feature drift during continual learning, and show that multi-layer distribution-based distillation mitigates this effect more effectively than conventional logit distillation. Nonetheless, our current understanding of feature drift remains empirical, and a more thorough theoretical analysis is needed to uncover its underlying causes. Furthermore, in DistKD we model feature distributions with Gaussians, following the design of Lv et al. (2024). However, Gaussian distributions may not always provide the optimal fit for the features of CLIP-based models. Exploring alternative distributional assumptions and associated distillation objectives represents an interesting direction for future research.

## I  THE USE OF LARGE LANGUAGE MODELS (LLMs)

We used large language models (LLMs) solely for writing assistance, including polishing grammar, wording, and presentation of already–written text. LLMs were not used to generate ideas, design experiments, analyze data, select citations, or write technical content. All scientific claims, algorithms,

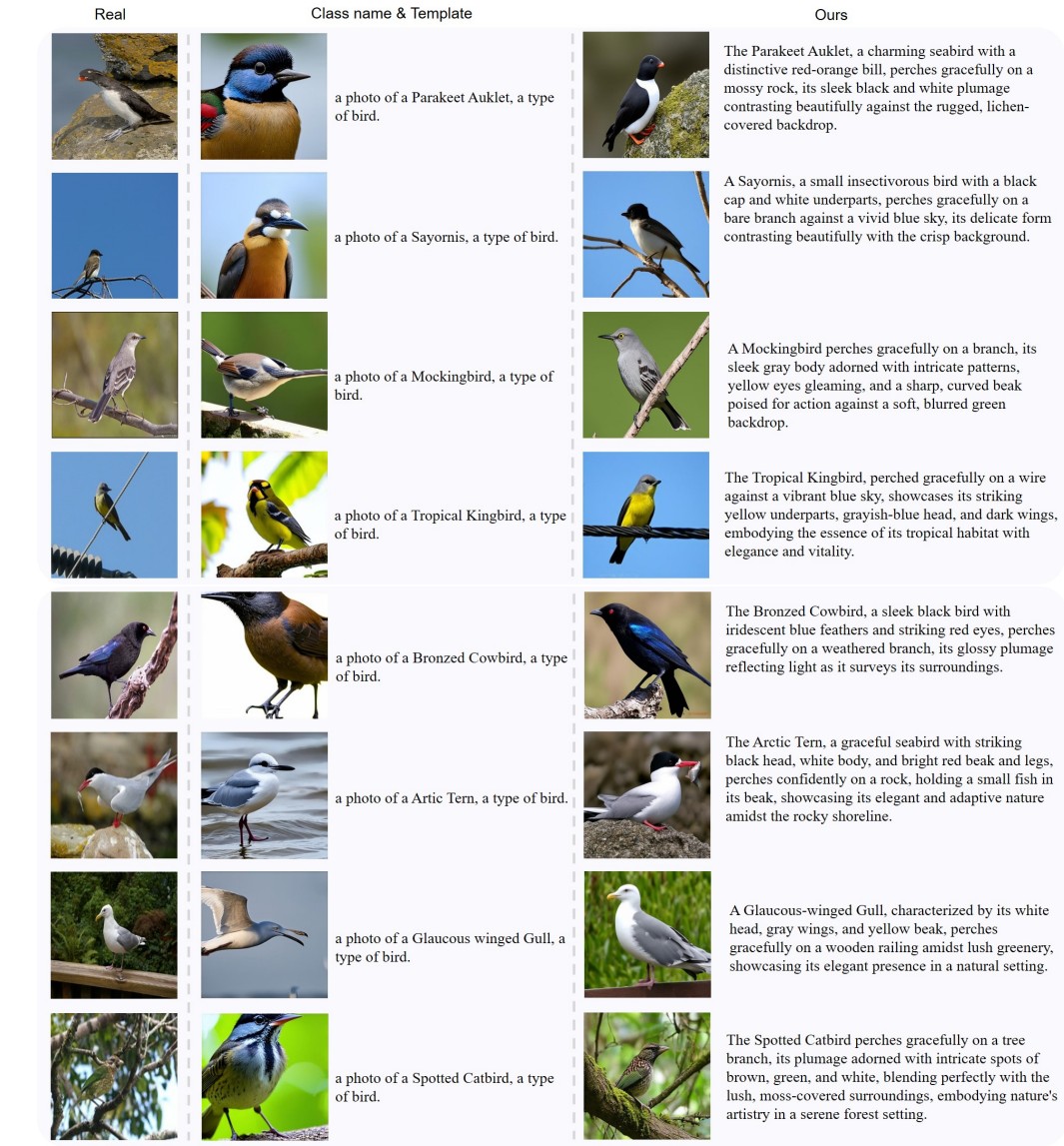

Figure 8: Visualization of synthetic data for CUB dataset.

and results are authored and verified by the authors; any LLM-suggested phrasing was reviewed and, when necessary, rewritten to ensure accuracy and avoid fabricated citations or biased language. No confidential, proprietary, or personally identifiable data were provided to the models.

## J    DETAILED RESULTS ON MTIL BENCHMARK

For a comprehensive and granular analysis, Table 17 and Table 19 present the detailed, per-dataset results for the Transfer, Avg, and Last metrics on the MTIL benchmark under Order I and Order II, respectively. To contextualize the performance of our method, we establish several key reference points. "Zero-shot" denotes the performance of the initial CLIP model, while "CLIP Fine-tune" represents the accuracy achieved by training the model on each dataset in isolation. These two serve as practical upper bounds, illustrating performance in the absence of catastrophic forgetting. At the other extreme, "CLIP Continual Finetuning" fine-tunes the model on each new task without any protective mechanisms, thereby suffering from maximal forgetting.

Across both orders, our method (CaRD) attains consistently strong performance and maintains stable accuracy as tasks accumulate. In settings with substantial domain shifts (e.g., MNIST, EuroSAT), baselines based on sequential fine-tuning or replay with class-name templates exhibit marked degra-

dation, reflecting the *catastrophic forgetting* discussed in the Introduction. In contrast, CaRD remains stable across the sequence and attains high *Last* accuracy on most datasets, indicating effective retention after learning all tasks.

Table 17: Detailed *Transfer*, *Avg.*, and *Last* scores (%) of different CL methods on the MTIL *full-shot* under **Order I**. The highest single score of each metric in each column is highlighted in **bold**.

| Method | Aircraft | Caltech101 | CIFAR100 | DTD | EuroSAT | Flowers | Food | MNIST | OxfordPet | Cars | SUN397 | Average |
|---|---|---|---|---|---|---|---|---|---|---|---|---|
| Zero-shot | 24.3 | 88.4 | 68.2 | 44.6 | 54.9 | 71.0 | 88.5 | 59.4 | 89.0 | 64.7 | 65.2 | 65.3 |
| Fine-tune | 62.0 | 95.1 | 89.6 | 79.5 | 98.9 | 97.5 | 92.7 | 99.6 | 94.7 | 89.6 | 81.8 | 89.2 |
| *Transfer* | | | | | | | | | | | | |
| Continual Finetune | | 67.1 | 46.0 | 32.1 | 35.6 | 35.0 | 57.7 | 44.1 | 60.8 | 20.5 | 46.6 | 44.6 |
| LwF (Li & Hoiem, 2017) | | 74.5 | 56.9 | 39.1 | 51.1 | 52.6 | 72.8 | 60.6 | 75.1 | 30.3 | 55.9 | 56.9 |
| iCaRL (Rebuffi et al., 2017) | | 56.6 | 44.6 | 32.7 | 39.3 | 46.6 | 68.0 | 46.0 | 77.4 | 31.9 | 60.5 | 50.4 |
| LwF-VR (Zhou et al., 2023) | | 77.1 | 61.0 | 40.5 | 45.3 | 54.4 | 74.6 | 47.9 | 76.7 | 36.3 | 58.6 | 57.2 |
| WiSE-FT (Wortsman et al., 2022) | | 73.5 | 55.6 | 35.6 | 41.5 | 47.0 | 68.3 | 53.9 | 69.3 | 26.8 | 51.9 | 52.3 |
| ZSCL (Zheng et al., 2023) | | 86.0 | 67.4 | 45.4 | 50.4 | 69.1 | 87.6 | 61.8 | 86.8 | 60.1 | 66.8 | 68.1 |
| MoE-Adapter (Yu et al., 2024a) | | 87.9 | 68.2 | 44.4 | 49.9 | 70.7 | 88.7 | 59.7 | 89.1 | 64.5 | 65.5 | 68.9 |
| GIFT (Wu et al., 2025) | | 88.5 | **69.8** | 46.0 | 49.4 | **68.5** | 87.1 | **69.9** | **88.9** | 57.7 | **67.7** | 69.3 |
| **CaRD** (Ours) | | **90.5** | 69.6 | **54.1** | **59.9** | 68.3 | **88.1** | 65.0 | 88.3 | **61.3** | 66.8 | **71.2** |
| *Avg.* | | | | | | | | | | | | |
| Continual Finetune | 25.5 | 81.5 | 59.1 | 53.2 | 64.7 | 51.8 | 63.2 | 64.3 | 69.7 | 31.8 | 49.7 | 55.9 |
| LwF (Li & Hoiem, 2017) | 36.3 | 86.9 | 72.0 | 59.0 | 73.7 | 60.0 | 73.6 | 74.8 | 80.0 | 37.3 | 58.1 | 64.7 |
| iCaRL (Rebuffi et al., 2017) | 35.5 | 89.2 | 72.2 | 60.6 | 68.8 | 70.0 | 78.2 | 62.3 | 81.8 | 41.2 | 62.5 | 65.7 |
| LwF-VR (Zhou et al., 2023) | 29.6 | 87.7 | 74.4 | 59.5 | 72.4 | 63.6 | 77.0 | 66.7 | 81.2 | 43.7 | 60.7 | 65.1 |
| WiSE-FT (Wortsman et al., 2022) | 26.7 | 86.5 | 64.3 | 57.1 | 65.7 | 71.1 | 70.5 | 75.8 | 36.9 | 54.6 | 60.7 | 64.0 |
| ZSCL (Zheng et al., 2023) | 45.1 | 92.0 | 80.1 | 64.3 | 79.5 | 81.6 | 89.6 | 75.2 | 88.9 | 64.7 | 68.0 | 75.4 |
| MoE-Adapter (Yu et al., 2024a) | 50.2 | 91.9 | **83.1** | 69.4 | 78.9 | **84.0** | 89.1 | 73.7 | 89.3 | **67.7** | 66.9 | 76.7 |
| GIFT (Wu et al., 2025) | 51.9 | 93.9 | 81.4 | 67.7 | 80.3 | 82.8 | 89.3 | **80.6** | 90.3 | 63.1 | **68.9** | 77.3 |
| **CaRD** (Ours) | **59.2** | **95.8** | 82.6 | **72.6** | **84.2** | 81.5 | **90.0** | 77.5 | 90.3 | 66.3 | 68.2 | **79.2** |
| *Last* | | | | | | | | | | | | |
| Continual Finetune | 31.0 | 89.3 | 65.8 | 67.3 | 88.9 | 71.1 | 85.6 | **99.6** | 92.9 | 77.3 | 81.1 | 77.3 |
| LwF (Li & Hoiem, 2017) | 26.3 | 87.5 | 71.9 | 66.6 | 79.9 | 66.9 | 83.8 | 99.6 | 92.1 | 66.1 | 80.4 | 74.6 |
| iCaRL (Rebuffi et al., 2017) | 35.8 | 90.3 | 77.0 | 72.0 | 83.3 | 88.5 | 90.4 | 86.7 | 93.2 | 81.2 | 81.9 | 80.1 |
| LwF-VR (Zhou et al., 2023) | 20.5 | 89.8 | 72.3 | 67.6 | 85.5 | 73.8 | 85.7 | 99.6 | 93.1 | 73.3 | 80.9 | 76.6 |
| WiSE-FT (Wortsman et al., 2022) | 27.2 | 90.8 | 68.0 | 68.9 | 86.9 | 74.0 | 87.6 | 99.6 | 92.6 | 77.8 | 81.3 | 77.7 |
| ZSCL (Zheng et al., 2023) | 40.6 | 92.2 | 81.3 | 70.5 | 94.8 | 90.5 | 91.9 | 98.7 | 93.9 | 85.3 | 80.2 | 83.6 |
| MoE-Adapter (Yu et al., 2024a) | 49.8 | 92.2 | **86.1** | 78.1 | 95.7 | 94.3 | 89.5 | 98.1 | 89.9 | 81.6 | 80.0 | 85.0 |
| GIFT (Wu et al., 2025) | 47.9 | 95.6 | 82.8 | 75.1 | 97.3 | 94.2 | 91.7 | 99.2 | 94.2 | 87.0 | 80.9 | 86.0 |
| **CaRD** (Ours) | **57.8** | **95.7** | 85.3 | **79.3** | **98.2** | **96.6** | **92.6** | 99.4 | **95.3** | **89.0** | **81.6** | **88.2** |

Table 18: Accuracy on MTIL **full-shot** and **5-shot** under **Order I**. Each row reports the performance with *Transfer*, *Average*, and *Last* on every dataset of the model trained after the corresponding task.

(a) Full-shot in Order I

| | Aircraft | Caltech101 | CIFAR100 | DTD | EuroSAT | Flowers | Food | MNIST | OxfordPet | Cars | SUN397 | |
|---|---|---|---|---|---|---|---|---|---|---|---|---|
| **Transfer** | | 90.5 | 69.6 | 54.1 | 59.9 | 68.3 | 88.1 | 65.0 | 88.3 | 61.3 | 66.8 | 71.2 |
| Aircraft | 60.3 | 90.5 | 69.7 | 55.4 | 56.1 | 71.2 | 89.2 | 58.5 | 90.4 | 65.2 | 68.0 | |
| Caltech101 | 59.9 | 96.6 | 69.5 | 53.0 | 57.4 | 67.6 | 88.6 | 59.5 | 88.6 | 61.5 | 66.7 | |
| CIFAR100 | 59.5 | 96.7 | 86.2 | 53.8 | 62.7 | 68.4 | 88.4 | 61.2 | 88.7 | 61.5 | 67.8 | |
| DTD | 59.4 | 96.1 | 85.8 | 79.6 | 63.3 | 67.3 | 87.5 | 67.7 | 87.6 | 60.7 | 66.3 | |
| EuroSAT | 59.6 | 96.4 | 85.4 | 79.4 | 98.2 | 66.9 | 87.3 | 69.4 | 87.8 | 60.8 | 66.3 | |
| Flowers | 59.4 | 96.2 | 85.4 | 79.3 | 98.2 | 97.7 | 87.5 | 68.6 | 87.8 | 60.4 | 66.2 | |
| Food | 59.7 | 96.1 | 85.3 | 79.4 | 98.2 | 97.3 | 92.9 | 69.8 | 87.8 | 61.0 | 66.9 | |
| MNIST | 58.9 | 96.7 | 85.3 | 80.0 | 98.2 | 97.3 | 92.9 | 99.3 | 87.9 | 60.3 | 66.7 | |
| OxfordPet | 58.7 | 96.3 | 85.3 | 79.4 | 98.1 | 96.9 | 92.8 | 99.3 | 95.3 | 60.1 | 66.7 | |
| Cars | 57.8 | 96.3 | 85.4 | 79.8 | 98.1 | 97.1 | 92.8 | 99.4 | 95.1 | 89.1 | 66.6 | |
| SUN397 | 57.8 | 95.7 | 85.3 | 79.3 | 98.2 | 96.6 | 92.6 | 99.4 | 95.3 | 89.0 | 81.6 | 88.2 |
| **Average** | 59.2 | 95.8 | 82.6 | 72.6 | 84.2 | 81.5 | 90.0 | 77.5 | 90.3 | 66.3 | 68.2 | 79.2 |

(b) 5-shot in Order I

| | Aircraft | Caltech101 | CIFAR100 | DTD | EuroSAT | Flowers | Food | MNIST | OxfordPet | Cars | SUN397 | |
|---|---|---|---|---|---|---|---|---|---|---|---|---|
| **Transfer** | – | 90.4 | 69.4 | 53.1 | 57.1 | 69.3 | 88.3 | 57.0 | 88.4 | 62.0 | 66.9 | 70.2 |
| Aircraft | 35.9 | 90.4 | 69.8 | 55.1 | 54.7 | 71.1 | 89.2 | 58.8 | 90.2 | 65.3 | 68.2 | |
| Caltech101 | 35.8 | 92.7 | 68.9 | 52.4 | 57.6 | 68.4 | 88.5 | 57.4 | 88.3 | 62.1 | 67.0 | |
| CIFAR100 | 35.6 | 92.3 | 75.1 | 51.7 | 59.0 | 69.8 | 78.5 | 55.3 | 88.7 | 62.3 | 67.6 | |
| DTD | 35.4 | 92.9 | 74.5 | 68.4 | 57.0 | 68.5 | 87.3 | 57.3 | 88.1 | 61.4 | 66.6 | |
| EuroSAT | 35.2 | 92.9 | 74.5 | 68.5 | 86.0 | 68.4 | 87.8 | 56.5 | 88.1 | 61.5 | 66.6 | |
| Flowers | 34.9 | 92.7 | 74.0 | 68.4 | 86.0 | 95.8 | 87.9 | 57.0 | 97.6 | 61.6 | 65.4 | |
| Food | 34.6 | 92.6 | 74.2 | 68.2 | 85.9 | 95.2 | 87.5 | 56.9 | 88.0 | 61.5 | 66.7 | |
| MNIST | 34.1 | 92.9 | 74.8 | 68.2 | 86.4 | 95.5 | 87.6 | 92.6 | 88.1 | 61.4 | 66.8 | |
| OxfordPet | 33.9 | 92.7 | 74.4 | 67.8 | 86.2 | 94.6 | 87.2 | 92.8 | 89.8 | 60.8 | 66.6 | |
| Cars | 33.3 | 92.4 | 74.6 | 67.8 | 86.0 | 94.7 | 86.9 | 92.7 | 89.2 | 75.8 | 66.3 | |
| SUN397 | 33.3 | 92.3 | 74.1 | 67.1 | 85.6 | 94.6 | 87.0 | 92.8 | 90.0 | 75.2 | 75.1 | 78.8 |
| **Average** | 34.7 | 92.4 | 73.5 | 64.0 | 75.5 | 83.3 | 87.8 | 70.0 | 88.8 | 64.5 | 67.6 | 72.9 |

Table 19: Detailed *Transfer*, *Avg.*, and *Last* scores (%) of different CL methods on the MTIL *full-shot* under **Order II**. The highest single score of each metric in each column is highlighted in **bold**.

| Method | Cars | Food | MNIST | OxfordPet | Flowers | SUN397 | Aircraft | Caltech101 | DTD | EuroSAT | CIFAR100 | Average |
|---|---|---|---|---|---|---|---|---|---|---|---|---|
| Zero-shot | 64.7 | 88.5 | 59.4 | 89.0 | 71.0 | 65.2 | 24.3 | 88.4 | 44.6 | 54.9 | 68.2 | 65.3 |
| Fine-tune | 89.6 | 92.7 | 94.7 | 94.7 | 97.5 | 81.8 | 62.0 | 95.1 | 79.5 | 98.9 | 89.6 | 89.2 |
| *Transfer* | | | | | | | | | | | | |
| Continual Finetune | | 85.9 | 59.6 | 57.9 | 40.0 | 46.7 | 11.1 | 70.0 | 30.5 | 26.6 | 37.7 | 46.6 |
| LwF (Li & Hoiem, 2017) | | 87.8 | 58.5 | 71.9 | 46.6 | 57.3 | 12.8 | 81.4 | 34.5 | 34.5 | 46.8 | 53.2 |
| iCaRL (Rebuffi et al., 2017) | | 86.1 | 51.8 | 67.6 | 50.4 | 57.9 | 11.0 | 72.3 | 31.2 | 32.7 | 48.1 | 50.9 |
| LwF-VR (Zhou et al., 2023) | | 88.2 | 57.0 | 71.4 | 50.0 | 58.0 | 13.0 | 82.0 | 34.4 | 29.3 | 47.6 | 53.1 |
| WiSE-FT (Wortsman et al., 2022) | | 87.2 | 57.6 | 67.0 | 45.0 | 54.0 | 12.9 | 78.6 | 35.5 | 28.4 | 44.3 | 51.0 |
| ZSCL (Zheng et al., 2023) | | 88.3 | 57.5 | 84.7 | 68.1 | 64.8 | 21.1 | 88.2 | 45.3 | 55.2 | 68.2 | 64.2 |
| MoE-Adapter (Yu et al., 2024a) | | 88.8 | 59.5 | 89.1 | 69.9 | 64.4 | 18.1 | 86.9 | 43.7 | 54.6 | 68.2 | 64.3 |
| GIFT (Wu et al., 2025) | | 88.3 | **63.4** | 88.1 | **70.8** | 67.7 | 22.8 | 90.4 | 46.7 | 51.8 | 68.8 | 65.9 |
| **CaRD** | | **89.6** | 59.6 | **90.1** | 70.7 | **68.0** | **26.6** | **90.5** | **52.6** | **57.5** | **69.3** | **67.4** |
| *Avg.* | | | | | | | | | | | | |
| Continual Finetune | 42.1 | 70.5 | 92.2 | 80.1 | 54.5 | 59.1 | 19.8 | 78.3 | 41.0 | 38.1 | 42.3 | 56.2 |
| LwF (Zhou et al., 2023) | 49.0 | 77.0 | 92.1 | 85.9 | 66.5 | 67.2 | 20.9 | 84.7 | 44.6 | 45.5 | 50.5 | 62.2 |
| iCaRL (Rebuffi et al., 2017) | 52.0 | 75.9 | 77.4 | 74.6 | 58.4 | 59.3 | 11.7 | 79.6 | 42.1 | 43.2 | 51.7 | 56.9 |
| LwF-VR (Zhou et al., 2023) | 44.9 | 75.8 | 91.8 | 85.3 | 63.5 | 67.6 | 16.9 | 84.9 | 44.0 | 40.6 | 51.3 | 60.6 |
| WiSE-FT (Wortsman et al., 2022) | 52.6 | 79.3 | 91.9 | 89.3 | 63.4 | 65.2 | 23.3 | 83.7 | 45.4 | 40.0 | 48.3 | 61.5 |
| ZSCL (Zheng et al., 2023) | 81.7 | 91.3 | 91.1 | 91.0 | 82.9 | 72.5 | 33.6 | 89.7 | 53.3 | 62.8 | 69.9 | 74.5 |
| MoE-Adapter (Yu et al., 2024a) | 84.9 | 89.9 | 89.3 | 91.4 | 86.2 | 72.2 | 33.4 | 89.4 | 53.3 | 61.4 | 69.9 | 74.7 |
| **GIFT** (Wu et al., 2025) | 83.2 | 90.8 | **92.6** | 92.8 | 85.8 | 74.1 | 36.0 | **92.1** | 54.7 | 60.0 | 70.4 | 75.7 |
| **CaRD** | **86.9** | **92.3** | 92.2 | **93.9** | **87.0** | **75.1** | **43.0** | 92.7 | **60.0** | **64.9** | **70.8** | **78.1** |
| *Last* | | | | | | | | | | | | |
| Continual Finetune | 24.0 | 67.3 | 99.1 | 87.4 | 44.3 | 67.0 | 29.5 | 92.3 | 61.3 | 81.0 | **88.1** | 67.4 |
| LwF (Li & Hoiem, 2017) | 34.6 | 69.6 | 99.3 | 88.7 | 61.1 | 72.5 | 32.5 | 88.1 | 65.6 | 90.9 | 87.9 | 71.9 |
| iCaRL (Rebuffi et al., 2017) | 46.0 | 81.5 | 91.3 | 82.8 | 66.5 | 72.2 | 16.3 | 91.6 | 68.1 | 83.2 | 87.8 | 71.6 |
| LwF-VR (Zhou et al., 2023) | 27.4 | 61.2 | 99.4 | 86.3 | 60.6 | 70.7 | 23.4 | 88.0 | 61.3 | 84.3 | 88.1 | 68.3 |
| WiSE-FT (Wortsman et al., 2022) | 35.6 | 76.9 | **99.5** | 89.1 | 62.1 | 71.8 | 27.8 | 90.8 | 67.0 | 85.6 | 87.6 | 72.2 |
| ZSCL (Zheng et al., 2023) | 78.2 | 91.1 | 97.6 | 92.5 | 87.4 | 78.2 | 45.0 | 92.3 | 72.7 | 96.2 | 86.3 | 83.4 |
| MoE-Adapter (Yu et al., 2024a) | 84.1 | 88.5 | 94.0 | 91.8 | 94.1 | 77.8 | 50.4 | 93.3 | 77.1 | 87.7 | 86.6 | 84.1 |
| **GIFT** (Wu et al., 2025) | 81.0 | 90.2 | 98.6 | 94.0 | 91.5 | 78.6 | 51.7 | 94.6 | 75.6 | 95.4 | 86.0 | 85.3 |
| **CaRD** | **85.2** | **92.0** | 99.4 | **95.3** | **95.2** | **80.6** | **62.0** | **96.5** | **79.4** | **98.1** | 86.3 | **88.2** |

Table 20: Accuracy on MTIL **full-shot** and **5-shot** under **Order II**. Each row reports the performance with *Transfer*, *Average*, and *Last* on every dataset of the model trained after the corresponding task.

(a) Full-shot in Order II

| | Cars | Food | MNIST | OxfordPet | Flowers | SUN397 | Aircraft | Caltech101 | DTD | EuroSAT | CIFAR100 | |
|---|---|---|---|---|---|---|---|---|---|---|---|---|
| **Transfer** | – | 89.6 | 59.6 | 90.1 | 70.7 | 68.0 | 26.6 | 90.5 | 52.6 | 57.5 | 69.3 | 67.4 |
| Cars | 88.5 | 89.6 | 59.5 | 90.6 | 71.3 | 67.8 | 27.2 | 90.9 | 53.9 | 52.1 | 68.5 | |
| Food | 88.3 | 93.0 | 59.7 | 89.8 | 70.9 | 67.9 | 27.2 | 90.3 | 52.9 | 54.8 | 68.8 | |
| MNIST | 88.3 | 93.0 | 99.4 | 90.0 | 70.3 | 68.2 | 27.0 | 90.5 | 52.8 | 58.0 | 70.7 | |
| OxfordPet | 88.0 | 93.0 | 99.4 | 95.5 | 70.3 | 68.1 | 25.8 | 90.5 | 52.7 | 57.8 | 70.2 | |
| Flowers | 87.8 | 93.0 | 99.4 | 95.4 | 97.5 | 68.0 | 25.8 | 91.0 | 53.1 | 58.5 | 70.0 | |
| SUN397 | 87.6 | 92.7 | 99.4 | 95.4 | 97.1 | 81.7 | 26.8 | 90.1 | 50.0 | 60.3 | 69.6 | |
| Aircraft | 86.3 | 92.5 | 99.5 | 95.4 | 96.9 | 81.5 | 63.5 | 90.1 | 53.4 | 58.8 | 69.7 | |
| Caltech101 | 85.7 | 92.4 | 99.5 | 95.4 | 96.2 | 81.2 | 62.5 | 96.9 | 52.1 | 59.0 | 69.0 | |
| DTD | 85.2 | 92.1 | 99.4 | 95.3 | 95.6 | 80.8 | 62.3 | 96.1 | 79.7 | 57.8 | 68.3 | |
| EuroSAT | 85.4 | 92.1 | 99.4 | 95.4 | 95.5 | 80.8 | 62.5 | 96.7 | 79.9 | 98.2 | 67.8 | |
| CIFAR100 | 85.2 | 92.0 | 99.4 | 95.3 | 95.2 | 80.6 | 62.0 | 96.5 | 79.4 | 98.1 | 86.3 | 88.2 |
| **Average** | 86.9 | 92.3 | 92.2 | 93.9 | 87.0 | 75.1 | 43.0 | 92.7 | 60.0 | 64.9 | 70.8 | 78.1 |

(b) 5-shot in Order II

| | Cars | Food | MNIST | OxfordPet | Flowers | SUN397 | Aircraft | Caltech101 | DTD | EuroSAT | CIFAR100 | |
|---|---|---|---|---|---|---|---|---|---|---|---|---|
| **Transfer** | – | 89.7 | 60.7 | 90.2 | 71.2 | 67.5 | 26.5 | 91.1 | 53.0 | 55.2 | 68.6 | 67.4 |
| Cars | 75.8 | 89.7 | 60.6 | 90.5 | 71.6 | 67.7 | 26.9 | 90.7 | 54.2 | 52.8 | 68.4 | |
| Food | 75.6 | 88.5 | 60.8 | 89.7 | 71.0 | 67.3 | 27.2 | 90.8 | 53.3 | 54.0 | 68.4 | |
| MNIST | 75.5 | 88.6 | 91.5 | 90.3 | 70.9 | 67.5 | 26.8 | 91.4 | 53.5 | 55.4 | 70.1 | |
| OxfordPet | 74.8 | 88.3 | 91.6 | 90.4 | 71.1 | 67.4 | 26.0 | 91.2 | 54.0 | 56.1 | 69.8 | |
| Flowers | 74.5 | 88.1 | 91.5 | 90.1 | 95.6 | 67.3 | 26.2 | 91.4 | 53.5 | 56.8 | 69.4 | |
| SUN397 | 74.0 | 88.0 | 91.6 | 90.1 | 95.3 | 75.3 | 25.8 | 91.0 | 53.1 | 51.2 | 57.4 | |
| Aircraft | 74.0 | 87.6 | 91.6 | 90.4 | 94.9 | 75.2 | 38.0 | 91.2 | 52.8 | 56.3 | 69.2 | |
| Caltech101 | 72.3 | 87.6 | 91.9 | 90.1 | 90.8 | 74.9 | 37.3 | 92.8 | 52.3 | 55.3 | 68.1 | |
| DTD | 71.9 | 86.8 | 92.3 | 90.1 | 94.2 | 74.4 | 37.5 | 93.0 | 68.7 | 52.4 | 66.9 | |
| EuroSAT | 72.0 | 86.8 | 92.5 | 89.9 | 94.0 | 74.0 | 37.1 | 92.5 | 68.7 | 86.0 | 69.7 | |
| CIFAR100 | 72.3 | 86.9 | 92.7 | 90.5 | 93.8 | 74.7 | 37.0 | 92.7 | 68.7 | 83.8 | 75.4 | 79.0 |
| **Average** | 73.9 | 87.9 | 86.2 | 90.2 | 86.1 | 71.4 | 31.5 | 91.7 | 57.3 | 60.6 | 69.2 | 73.3 |

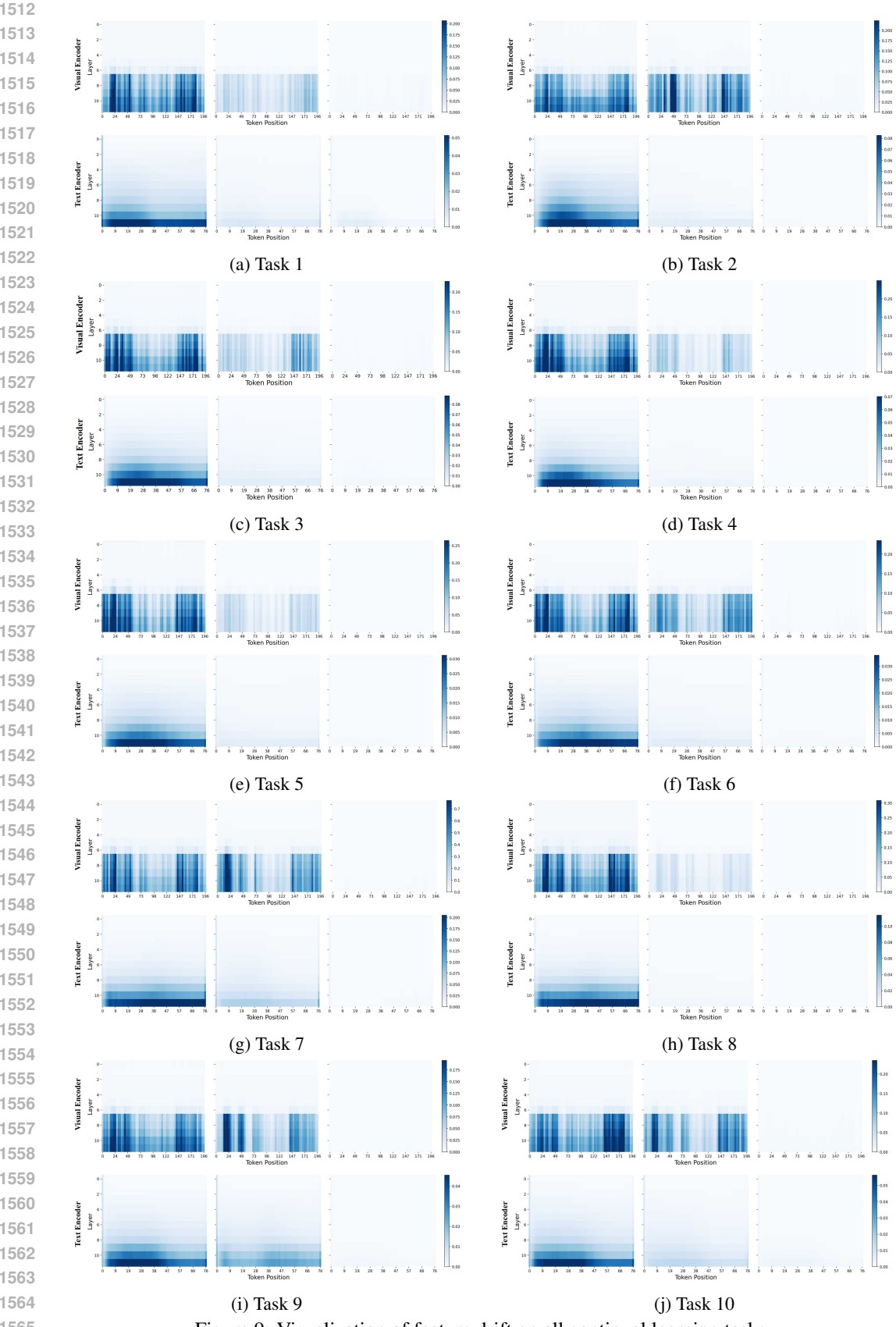

(a) Task 1

(b) Task 2

(c) Task 3

(d) Task 4

(e) Task 5

(f) Task 6

(g) Task 7

(h) Task 8

(i) Task 9

(j) Task 10

Figure 9: Visualization of feature drift on all continual learning tasks.

