# OpenReview forum: "Distilling Knowledge from Caption-Guided Replay for VLM-based Continual Learning"
_ICLR.cc/2026/Conference — Submitted to ICLR 2026_

### Official Review · Reviewer_DtAe · 2025-10-22

**Soundness:** 3
**Presentation:** 3
**Contribution:** 2
**Rating:** 4
**Confidence:** 4

**Summary:**

This paper proposes a caption-guided replay paradigm that stores instance-level captions generated by a multi-modal large language model as memory and reconstructs past images using a LoRA-adapted text-to-image model, achieving high-fidelity and instance-aware synthetic replay with efficient storage. This approach effectively addresses the data privacy concerns and heavy storage demands associated with replay-based methods that rely on real historical samples. Additionally, this paper identifies the issue of feature drift in continual learning—shifts in intermediate representations during sequential training that are only partially alleviated by conventional logit distillation. To tackle this, the proposed method introduces a distribution-based distillation strategy that aligns feature distributions across multiple intermediate layers, thereby suppressing feature drift and enhancing model stability.

**Strengths:**

1.	This paper introduces a novel caption-guided replay paradigm that not only addresses the storage burden and data privacy issues inherent in traditional replay-based methods relying on real historical data, but also overcomes the limitations of conventional generative replay approaches. Existing generative replay methods, which typically use simple prompts constructed from class names and CLIP-style templates, fail to capture instance-aware semantics and thus often produce synthetic images with a noticeable fidelity gap compared to original task data. In contrast, this method generates detailed, instance-level captions for each image, enabling the text-to-image (T2I) model to synthesize images that are semantically and visually closer to the originals.
2.	Task-specific adaptation is achieved by fine-tuning the text-to-image model with task-specific LoRA modules, ensuring that the visual characteristics of each task are well preserved during replay.
3.	The proposed caption-guided replay paradigm is compatible with existing replay-based frameworks and can be seamlessly integrated to enhance replay fidelity and efficiency.

**Weaknesses:**

1.	Although the proposed method leverages a Multi-modal Large Language Model (MLLM) to generate captions for each image, enabling the text-to-image (T2I) model to produce high-fidelity, instance-aware samples, the overall generation process is computationally expensive. It involves three stages: (1) the MLLM generates instance-level captions, (2) the T2I model undergoes LoRA fine-tuning using these captions, and (3) the T2I model synthesizes replay samples. All three steps are highly time-consuming. While Table 8 reports the training time of the method, the reported duration (Real images: 403 ms; Instance-level Caption: 404 ms) is almost identical to that of real-image replay, suggesting that the additional computational overhead of these three processes may not have been included in the reported training time.
2.	Using MLLMs to produce instance-level descriptions and T2I models to synthesize images can introduce domain-specific challenges. For fine-grained categories or cross-domain scenarios such as medical scenarios, these foundation models may lack the necessary domain knowledge, leading to noisy or invalid captions and images that fail to capture meaningful semantics.
3.	In terms of novelty, the proposed multi-layer distribution-based distillation strategy lacks originality, as a similar concept has already been introduced in PODNet [1], which also employs MSE loss for multi-layer distribution distillation in continual learning. The authors should clearly explain the differences between their approach and PODNet to justify their contribution.
4.	The paper also lacks a clear description of the preliminary. Continual learning encompasses various settings, including class-incremental, task-incremental, domain-incremental, and online learning. From the context, it appears that this work adopts a cross-dataset continual learning setup, but this should be explicitly stated in the methodology section.
5.	While the proposed approach performs well in cross-dataset continual learning, the domain gap between datasets is substantial. It would be valuable to investigate whether the method can be applied to intra-dataset class-incremental learning, particularly on fine-grained datasets such as CUB [2] or Cars [3], and how does it perform compared to existing baselines?  This is especially relevant since the method relies on a T2I model—whether such a model can generate fine-grained images that truly capture class-specific attributes remains an open question.
6.	The comparison with state-of-the-art methods also raises concerns. For instance, MoE-Adapter [4] is a non-replay-based method that does not use generated old samples, which makes the comparison potentially unfair. The authors should clearly indicate in the results table whether each compared method employs replay, and if so, whether it uses real or synthetic samples.
7.	The organization of the paper could be improved. The Introduction section should avoid including excessive experimental results and instead focus on motivation and preliminary observations directly related to the research question.
8.	The manuscript contains several typographical and formatting errors (e.g., ”***”), which should be carefully corrected throughout the paper.
[1] Douillard A, Cord M, Ollion C, et al. Podnet: Pooled outputs distillation for small-tasks incremental learning[C]. ECCV. 2020.
[2] Wah C, Branson S, Welinder P, et al. The caltech-ucsd birds-200-2011 dataset[J]. 2011.
[3] Krause J, Stark M, Deng J, et al. 3d object representations for fine-grained categorization[C]. ICCV. 2013.
[4] Yu J, Zhuge Y, Zhang L, et al. Boosting continual learning of vision-language models via mixture-of-experts adapters[C]. CVPR. 2024.

**Questions:**

1.	The proposed method involves three computationally intensive stages: MLLM caption generation, LoRA fine-tuning, and image synthesis. However, the training time reported in Table 8 is almost identical to that of real-image replay. Could you please clarify exactly which processes are included in this reported time? Specifically, does it encompass the full overhead of the MLLM inference and the T2I model fine-tuning steps? If not, please provide a more comprehensive breakdown of the computational cost.
2.	The method relies on general-purpose foundation models (MLLM and T2I). How does it handle domain-specific scenarios, such as medical imaging or fine-grained categories, where these models may lack the necessary knowledge? What evidence or experiments can you provide to show that the generated captions and images in such domains are semantically meaningful and not noisy?
3.	The multi-layer distribution-based distillation strategy bears a strong resemblance to the approach used in PODNet, which also employs MSE loss for multi-layer distribution distillation. Could you explicitly delineate the key differences between your distillation strategy and that of PODNet? Please justify the novelty of your contribution in this specific aspect.
4.	The paper does not explicitly define the continual learning setting it adopts (e.g., class-incremental, task-incremental, domain-incremental). Based on the experiments, it seems to be a form of domain-incremental or cross-dataset continual learning. Please clearly state and formally define the exact continual learning setting used in the methodology section.
5.	While the method is evaluated on cross-dataset scenarios with large domain gaps, its performance on standard class-incremental learning within a single fine-grained dataset (e.g., CUB, Cars) remains unclear. Can you demonstrate the method's effectiveness in such a setting? Given the reliance on a T2I model, can it reliably generate high-fidelity images that preserve the subtle, class-specific attributes necessary for fine-grained classification?
6.	The comparisons include methods like MoE-Adapter, which is a non-replay-based approach. To ensure a fair and transparent comparison, please clearly indicate in the results tables (e.g., with a footnote or a separate column) which compared methods use a replay strategy and, if so, whether they use real or synthetic samples for replay.
7.	The Introduction section currently includes extensive experimental results. It is recommended to streamline this section by focusing on the motivation, background, and the specific research gaps this work aims to address. Detailed experimental analysis should be reserved for the later sections to improve the paper's narrative flow.
8.	The manuscript contains several typographical and formatting errors (e.g., ”***”). Please perform a thorough proofreading of the entire document to correct these issues and improve the overall presentation quality.

---

> ### Author Response · Authors · 2025-11-21
> **Author Response 1**
>
> Dear Reviewer DtAe：
>
> We sincerely thank the reviewer for the thoughtful and detailed comments. We also appreciate the strengths highlighted in the review, including that “a novel caption-guided replay paradigm”, and that “This paper features a clear structure and method diagram, and is easy to understand”. All modifications in the revised submission are highlighted in blue, and each revision is explicitly referenced in our responses below. Our point-by-point replies are provided next.

---

> ### Author Response · Authors · 2025-11-21
> **Author Response 2**
>
> > ### Q1: Provide a more comprehensive breakdown of the computational cost.
>
> We thank the reviewer for the constructive feedback. Kindly noted that, CaRD is a generative replay–based method, and therefore should be compared primarily against methods of the same category when analyzing cost and performance. **Importantly, CaRD introduces no additional inference-time cost and that its overall computational complexity is comparable to existing generative replay approaches.** Like GIFT and LoRA-Loop, CaRD synthesizes replay samples **once during an offline pre-processing stage** before CLIP training begins. In generative replay, a generation model is used to synthesize images that approximate the data distribution of previous tasks, enabling replay without accessing original real data to avoid data privacy or storage concerns. This offline construction does not participate in CLIP’s training loop or inference, and all synthesized samples are discarded immediately after training, resulting in **no long-term storage overhead.**
>
> We then provide a detailed analysis of computation cost. In the original submission, Table 8 indeed reported only the per-step CLIP training time and did not clearly account for the costs of each replay stage. **In the revised manuscript, we have fully redesigned Table 8 (highlighted in blue) in section 4.4.3 at page 10 (and also reproduced below for convenience)**, to incorporate all components of CaRD’s computational overhead, including (1) Caption generation, (2) LoRA fine-tuning of the T2I model, (3) Synthetic image generation, and (4) CLIP training. All results are averaged over the full sequence of 11 MTIL tasks.
>
> In this revised table, Replay Total Time corresponds to the actual wall-clock time for stages (1)–(3), while *Total Time* reflects the sum of replay construction and CLIP training. **All timing measurements are obtained under a unified experimental setup**, for example, using vLLM for efficient Multi-modal LLM inference, TensorRT acceleration for T2I generation, and consistent LoRA hyperparameters across methods, to ensure a fair comparison.
>
> Overall, the revised Table provides a complete and transparent view of computational cost. CaRD’s replay construction cost remains comparable to other generative replay methods. Importantly. The captioning cost can be further reduced by using smaller MLLMs (13.6 min → 8.4 when QWen2.5-VL 7B → 3B), and in 5-shot settings it becomes almost negligible (0.5 min) due to the limited number of training images. **Furthermore, employing smaller captioning or T2I models, or adopting more advanced acceleration techniques, can further reduce replay construction cost.** These improvements are orthogonal to CaRD’s core representation and replay mechanisms and do not alter the method’s fundamental contributions.
>
>
> | Method | Multi-Modal LLM (#Params) | Caption Gen. (min) | T2I Model (#Params) | FT T2I (min) | Image Gen. (min) | Replay Total Time (min) | Training (min) | Total Time (min) | Storage Cost (MB) | Trans. / Avg. / Last |
> |:--------|:--------------------------:|:-------------------:|:---------------------:|:------------:|:-----------------:|:------------------------:|:---------------:|:------------------:|:--------------------:|:----------------------:|
> | *Full-shot* | | | | | | | | | | |
> | GIFT (C&T) | – | – | SD1.5 (0.9B) | –   | 11.6 | 11.6 | 6.2 | 17.8 (**×1.0**) | 0.2  | 69.3 / 77.3 / 86.0 |
> | LoRA-Loop (C&T) | – | – | SD1.5 (0.9B) | 5.1 | 11.6 | 16.7 | 6.2 | 22.9 (**×1.3**) | 30.2 | 69.8 / 77.6 / 86.0 |
> | LoRA-Loop (C&T) | – | – | SD3-M (2B)   | 8.2 | 12.4 | 20.6 | 6.2 | 26.8 (**×1.5**) | 47.2 | 70.0 / 77.7 / 86.1 |
> | CaRD | QWen2.5-VL (3B) | 8.4  | SD3-M (2B)   | 8.2 | 12.4 | 29.0 | 6.7 | 35.7 (**×2.0**) | 49.2 | 71.1 / 79.2 / 88.2 |
> | CaRD | QWen2.5-VL (7B) | 13.6 | SD1.5 (0.9B) | 5.1 | 11.6 | 30.3 | 6.7 | 37.0 (**×2.1**) | 34.2 | 71.0 / 79.1 / 88.0 |
> | CaRD | QWen2.5-VL (7B) | 13.6 | SD3-M (2B)   | 8.2 | 12.4 | 34.2 | 6.7 | 40.9 (**×2.3**) | 49.2 | 71.2 / 79.2 / 88.3 |
> |  |  |  |  |  |  |  |  |  |  |  |
> | *5-shot* | | | | | | | | | | |
> | GIFT (C&T) | – | – | SD3-M (2B)   | –   | 12.4 | 12.4 | 6.2 | 18.6 (**×1.0**) | 0.2  | 69.0 / 72.5 / 77.8 |
> | LoRA-Loop (C&T) | – | – | SD3-M (2B)   | 8.2 | 12.4 | 20.6 | 6.2 | 26.8 (**×1.4**) | 47.2 | 69.2 / 72.9 / 78.0 |
> | CaRD | QWen2.5-VL (7B) | 0.5 | SD3-M (2B)   | 8.2 | 12.4 | 21.1 | 6.7 | 27.8 (**×1.5**) | 47.6 | 70.4 / 73.2 / 79.2 |
>
>
> [vLLM] Kwon, Woosuk, et al. "Efficient memory management for large language model serving with pagedattention." Proceedings of the 29th symposium on operating systems principles. 2023. https://docs.vllm.ai/en/latest/
>
> [TensorRT] https://github.com/NVIDIA/TensorRT/tree/main/demo/Diffusion

---

> ### Author Response · Authors · 2025-11-21
> **Author Response 3**
>
> > ### Q2: How does it handle domain-specific scenarios, such as medical imaging or fine-grained categories, where these general-purpose foundation models may lack the necessary knowledge? What evidence or experiments can you provide to show that the generated captions and images in such domains are semantically meaningful and not noisy?
>
>
> We appreciate the reviewer’s attention to CaRD’s applicability in domain-specific scenarios. We fully understand the concern: in specialized domains such as medical imaging or fine-grained classification, general-purpose foundation models may lack adequate domain coverage, potentially affecting the semantic quality of both captions and synthesized images.
>
> First, **CaRD is fundamentally a model-agnostic and component-replaceable framework**, and does not rely on the domain generality of QWen2.5-VL or SD3-Medium. The core contribution lies in the caption-guided replay + multi-layer distribution distillation pipeline, which remains unchanged regardless of the specific models instantiated. For example, in medical imaging scenarios, the general-purpose Multi-model LLM can be replaced with a domain-specialized multimodal model (e.g., LLaVA-Med), and the generic diffusion model can be substituted with a medical-specific generator (e.g., Medfusion), allowing CaRD to produce replay images that remain aligned with the domain’s semantic structure.
>
> Second, although this paper does not include medical tasks, **the MTIL benchmark already presents significant domain shifts and fine-grained categories,** including Aircraft, Cars, OxfordPet, DTD (textures), and EuroSAT (remote sensing). Many of these domains deviate substantially from the T2I model’s training distribution. Despite such cross-domain discrepancies, the generated captions and synthetic images consistently preserve semantically coherent structures aligned with the corresponding real data, enabling effective replay for continual learning. This is supported by the image-quality metrics in Table 3 and the visualizations in Appendix Figures 6 and 7. These results serve as indirect yet compelling evidence that CaRD can still produce semantically reliable replay data in challenging fine-grained and cross-domain settings.
>
> In summary, CaRD does not depend on any particular general-purpose foundation model; rather, it provides a flexible and domain-adaptable framework that can incorporate specialized multimodal LLMs and diffusion models when applied to expert domains. Meanwhile, MTIL results already demonstrate that CaRD maintains sufficient semantic fidelity even under notable domain shift and fine-grained variation.
>
> [LLaVA-Med] Li, Chunyuan, et al. "Llava-med: Training a large language-and-vision assistant for biomedicine in one day." Advances in Neural Information Processing Systems 36 (2023): 28541-28564.
>
> [Medfusion] Gustav Müller-Franzes, et al. " Diffusion Probabilistic Models beat GANs on Medical Images." arXiv preprint arXiv:2212.07501 (2022).

---

> ### Author Response · Authors · 2025-11-21
> **Author Response 4**
>
> > ### Q3: Explicitly delineate the key differences between your distillation strategy and that of PODNet? Please justify the novelty of your contribution in this specific aspect.
>
> We thank the reviewer for the thoughtful assessment of DistKD’s novelty. Below we clarify the key conceptual and technical differences between DistKD and PODNet, and provide direct empirical evidence (revised Table 5) supporting the contribution of our method.
>
> **(1) Rather, the main contribution of DistKD lies: A systematic observation and characterization of a previously unreported failure mode specific to VLM-based continual learning.**
>
> Our analysis (Figure 1(b), Appendix Figure 9) reveals a multi-layer, multi-token drift phenomenon that emerges when CLIP is finetuned sequentially. **To our knowledge, this drift behavior has not been documented before in the distillation or continual learning literature.** Once this phenomenon was identified, it was natural and well-motivated to explore distribution-alignment tools to counteract the drift.
>
> **(2) Differences between DistKD and PODNet.**
>
> Regarding PODNet, we clarify that its formulation is fundamentally tied to CNN-based class-incremental learning. PODNet constrains the evolution of convolutional feature maps via spatial pooling and elementwise MSE. **It does not model the geometry of feature distributions**, nor does it handle multi-token transformer representations or cross-modal encoders.
>
> In contrast, DistKD differs from PODNet in several core aspects:
> (1) **Distillation targets:** PODNet operates on 2D convolutional feature maps, whereas DistKD targets transformer token sequences (vision patch tokens and text tokens), whose representational structures differ substantially.
> (2) **Distillation objectives:** PODNet applies MSE to pooled features. DistKD instead estimates mean and covariance of token distributions at each layer and aligns their global geometry via the closed-form 2-Wasserstein distance, while also applying token-level MSE to preserve fine-grained semantics.
> (3) ** Distillation Modalities:** PODNet is restricted to the visual branch. DistKD applies to both the vision and text encoders, explicitly addressing feature drift unique to VLMs in CL.
>
> To further support this distinction, **we implemented a PODNet on CLIP-ViT** (revised Table 5, Row 7), applying spatial pooling + MSE to visual tokens (reshaped into H×W grids) and pooled-token MSE to text features. Under the same MTIL full-shot protocol, this method achieves 69.5 / 77.1 / 86.0 / –4.22 / 4.82 (Trans./Avg./Last/BWT/AF), which is clearly below both our Wasserstein-only variant (70.6 / 78.4 / 87.5 / –0.72 / 0.82) and the full DistKD (71.2 / 79.2 / 86.3 / -0.61 / 0.64) formulation (Table 5, Row 8). This demonstrates that directly transplanting PODNet into the VLM setting does not replicate the effectiveness of DistKD. Moreover, in the revised manuscript we include a visualization comparison with PODNet in Figure 3 (Appendix B, page 19), which further demonstrates that DistKD is markedly more effective in mitigating feature drift.
>
>
> > ### Q4: Please clearly state and formally define the exact continual learning setting used in the methodology section.
>
>
> We thank the reviewer for highlighting the importance of clearly specifying the continual learning protocol. In the revised manuscript, we have further clarified the benchmark settings in Section 4.1: DATASETS AND METRICS.
>
> Our experiments are conducted on two publicly established VLM-based CL benchmarks: MTIL and X-TAIL. MTIL is a **Multi-domain Task-Incremental Learning** benchmark designed specifically for VLMs. It consists of 11 heterogeneous image classification datasets, each corresponding to an independent task. The model continually adapts to these tasks in a sequential manner, and evaluation jointly measures the retention of incrementally acquired task-specific knowledge and the preservation of pre-trained knowledge. Notably, **MTIL provides the task identity during inference**, making it a task-incremental setting.  **X-TAIL extends MTIL by integrating the class-incremental evaluation protocol**.  In this scenario, the model must continually learn new classes across multiple domains and distinguish among them **without access to task identity at inference time**. We adhere to the established protocols with two task orders for both full-shot and few-shot scenarios.

---

> ### Author Response · Authors · 2025-11-21
> **Author Response 5**
>
> > ### Q5: Demonstrate the method's effectiveness in class-incremental learning. Given the reliance on a T2I model, can it reliably generate high-fidelity images that preserve the subtle, class-specific attributes necessary for fine-grained classification?
>
> Thank you for raising this insightful question regarding the applicability of our approach to intra-dataset class-incremental learning, particularly on fine-grained datasets. To address this concern, we have extended our experiments and applied CaRD directly to the standard class-incremental learning (CIL) setup on CUB in **Appendix Table 16 (page 25, and reproduced below for convenience).**
>
> Following the protocol used in *PROOF*, we divide the classes within the same dataset into sequential tasks. “**B**m” denotes the number of base classes, and “**Inc**n” indicates the number of incremental classes per task (m = 0 means that all classes are evenly split across tasks). All method within real replay use 20 real samples per class as replay data.
>
> We compare CaRD against other CIL methods specifically designed for this setting, and Table 16 (page 25) reports the full results, including both the average accuracy Ā and the final-stage accuracy A$_B$. The results show that CaRD equipped only with our caption-guided generative replay achieves performance on par with real-replay methods such as PROOF under two settings. Moreover, when combined with real replay, CaRD further surpasses the performance of the real-replay method alone.
>
> Finally, regarding the concern about whether the T2I model can retain preserve the subtle, class-specific attributes, **we provide additional qualitative visualizations in Figure 8 (page 26)**. These results indicate that the images reconstructed via our caption-guided generation strategy maintain consistency with the real samples.
>
> [PROOF] Zhou, Da-Wei, et al. "Learning without forgetting for vision-language models." IEEE Transactions on Pattern Analysis and Machine Intelligence (2025).
>
> | Replay | Method |        CUB B0 Inc20        |        |        CUB B100 Inc20       |        |
> |--------|--------|:----------------------------:|:--------:|:------------------------------:|:--------:|
> |        |        | Ā      | A$_B$                | Ā      | A$_B$                          |
> | **No Replay** | Finetune | 2.06 | 0.64 | 0.56 | 0.47 |
> | | SimpleCIL  | 83.81 | 77.52 | 79.75 | 77.52 |
> | | RAPF  | 83.04 | 76.34 | – | – |
> | | ZS-CLIP  | 74.38 | 63.06 | 67.96 | 63.06 |
> | **Real Replay** | iCaRL | 82.04 | 74.74 | 78.57 | 75.07 |
> | | MEMO  | 77.32 | 65.69 | 72.88 | 66.41 |
> | | PROOF  | 84.93 | 79.47 | 81.67 | 79.18 |
> | **Generative Replay** | CaRD (Ours) | 83.44 | 77.55 | 79.81 | 77.20 |
> | **Generative Replay + Real Replay** | CaRD (Ours) | **86.22** | **80.64** | **83.13** | **80.01** |
>
>
> > ### Q6: Clearly indicate in the results tables (e.g., with a footnote or a separate column) which compared methods use a replay strategy and, if so, whether they use real or synthetic samples for replay.
>
>
> We thank the reviewer for this important suggestion. In response, we reorganized the main results tables and explicitly added a **“Replay” column** to Table 1 and Table 2 in the revised manuscript.
>
>
> > ### Q7: The Introduction section should avoid including excessive experimental results and instead focus on motivation and preliminary observations directly related to the research question.
>
> We thank the reviewer for the insightful suggestions regarding the overall structure of the paper, particularly the organization of the Introduction. In the revised manuscript, we have made the corresponding adjustments. Specifically, the original Figure 1(a) has been moved to the experimental section, and Figure 1(c) has been removed. **The accompanying narrative around Figure 1 has also been rewritten to retain only the preliminary observations and qualitative comparisons directly relevant to the motivation of our method.**
>
> These revisions make the Introduction more focused on the problem setting, key intuitions, and the motivation behind our approach, thereby improving the clarity and coherence of the overall presentation. We sincerely appreciate the reviewer’s valuable feedback.
>
>
> > ### Q8: The manuscript contains several typographical and formatting errors.
>
> We thank the reviewer for pointing out the notation and formatting issues in the manuscript. In the revised version, we have conducted a thorough manual proofreading and formatting audit across the entire paper, and we have corrected all identified inconsistencies and typographical errors. We appreciate the reviewer’s attention to these important presentation details.

---

> > ### Comment · Reviewer_DtAe · 2025-11-26
> >
> > From the perspective of the reported time cost, the total time required for generating replay images, including caption generation, fine-tuning the T2I model with task-specific LoRA, and synthetic image generation, remains within a reasonable and manageable range. In continual learning, ensuring that synthetic images closely match the distribution of real images is a crucial challenge, and employing task-specific LoRA fine-tuning for T2I models is indeed an innovative idea. I find this direction generally convincing. The authors also provide a clear explanation of how their multi-layer knowledge distillation approach differs from PODNet. In addition, the intra-dataset experiments on CUB200 further demonstrate the effectiveness of the proposed method on fine-grained datasets. However, it is worth noting that in the last row of the table, the authors did not specify the ratio or exact number of generated samples relative to real samples. It would be beneficial if the authors could also report the results of using only real samples in this experiment. If the authors can address this issue, I would be inclined to reconsider my evaluation and raise the score.

---

> > > ### Author Response · Authors · 2025-11-27
> > > **Author Response**
> > >
> > > We are pleased that our previous response addressed some your concerns, and we sincerely thank you for the thoughtful follow-up. To resolve the remaining issue, we have further enriched the CUB200 results by (i) adding results of CaRD using only real samples (**denoted *CaRD#***), and (ii) explicitly reporting the **exact number of Real vs. Synthetic samples together with the Total replay samples per class** used by each method. The updated table is reproduced below for convenience.
> > >
> > > * Real Replay. All methods in the *Real Replay* , including iCaRL, MEMO, PROOF, and our *CaRD#*, use exactly 20 real images per class as replay data. As shown, ***CaRD#* performs competitively** among real-replay methods under both B0 Inc20 and B100 Inc20.
> > >
> > > * Generative Replay. For CaRD with only generative replay, we vary the number of synthetic images per class from 20 → 40 → 80. We observe that increasing synthetic replay from 20 to 40 images yields clear gains, while further increasing to 80 produces only marginal improvements. **We therefore adopt 40 synthetic images per class for trade-off.**
> > >
> > > * Generative Replay + Real Replay. Finally, when combined with real replay, CaRD further surpasses the performance of the real-replay method alone **(last row, 20 real + 40 synthetic images)**. For a fair comparison under the same total budget as real replay, we also include a 5 real + 15 synthetic variant, which remains competitive.
> > >
> > > We hope these additional experiments fully address your concern.
> > >
> > >
> > >
> > > | Replay Type | Method | Replay |Samples  | (per class) |  CUB B0 Inc20  |      |   CUB B100 Inc20  |      |
> > > |-------------|--------|-------------------------------|-------------|------------------|:--------------:|:----:|:-----------------:|:----:|
> > > |      |   | Total | Real | Synthetic | Ā              | A_B  | Ā                | A_B  |
> > > | **No Replay** | Finetune | -- | -- | -- | 2.06 | 0.64 | 0.56 | 0.47 |
> > > |             | SimpleCIL | -- | -- | -- | 83.81 | 77.52 | 79.75 | 77.52 |
> > > |             | RAPF | -- | -- | -- | 83.04 | 76.34 | – | – |
> > > |             | ZS-CLIP | -- | -- | -- | 74.38 | 63.06 | 67.96 | 63.06 |
> > > | **Real Replay** | iCaRL | 20 | 20 | -- | 82.04 | 74.74 | 78.57 | 75.07 |
> > > |             | MEMO | 20 | 20 | -- | 77.32 | 65.69 | 72.88 | 66.41 |
> > > |             | PROOF | 20 | 20 | -- | 84.93 | 79.43 | 81.67 | 79.18 |
> > > |             | **CaRD# (Ours)** | **20** | **20** | -- | **85.41** | **79.82** | **82.28** | **79.52** |
> > > | **Generative Replay** | CaRD (Ours) | 20 | -- | 20 | 83.04 | 76.62 | 79.11 | 76.93 |
> > > |             | **CaRD (Ours)** | **40** | -- | **40** | **83.44** | **77.55** | **79.81** | **77.20** |
> > > |             | CaRD (Ours) | 80 | -- | 80 | 83.53 | 77.79 | 80.04 | 77.51 |
> > > | **Generative Replay + Real Replay** | CaRD (Ours) | 20 | 5 | 15 | 85.10 | 79.12 | 81.94 | 78.93 |
> > > |             | **CaRD (Ours)** | **60** | **20** | **40** | **86.22** | **80.64** | **83.13** | **80.01** |

---

### Official Review · Reviewer_KkVw · 2025-10-27

**Soundness:** 2
**Presentation:** 3
**Contribution:** 2
**Rating:** 4
**Confidence:** 4

**Summary:**

This work introduces a caption-guided generative replay framework for continual learning with vision-language models. Instead of relying on coarse class-level prompts, it stores instance-level captions generated by a multimodal LLM and reconstructs past samples using a LoRA-adapted text-to-image model, enabling high-fidelity, instance-aware replay with minimal storage. Furthermore, it identifies and addresses feature drift—representation shifts during sequential learning—through a distribution-based distillation strategy that aligns intermediate feature distributions. Together, these techniques enhance both replay quality and model stability in continual learning.

**Strengths:**

1. The experiments in this paper are extensive, analyzing the proposed method from different perspectives.
2. This paper features a clear structure and method diagram, and is easy to understand

**Weaknesses:**

1. Training the T2I model and ingesting a multimodal LLM will bring additional training costs. Is it necessary to ingest two large-scale models to train the CLIP model?
2. Authors emphasize in the abstract and introduction that the simple prompts constructed by using class name and CLIP templates in existing methods have limitations, but they still use class name in the comparison in Figure 1. This cannot well provide support for the claimed motivation.
3. The compared methods, such as GIFT and LoRA-Loopd, both use SD1.5 as the T2I model. Why does this paper adopt SD3-Medium? This may cause a discussion on comparison fairness. And from Table 4(b), SD3-Medium does not bring obvious performance boosts.
4. The ablation study provided by the authors do not showcase the actual baseline, which may confuse readers about the results. Row 3 of Table 3, when using class name and template, already outperforms all the comparison methods. Is this due to the use of a stronger baseline model?
5. What is the core innovation of DistKD? Distilling the hidden states at the middle layers of the model is a common technique.

**Questions:**

See weaknesses.

---

> ### Author Response · Authors · 2025-11-21
> **Author Response 1**
>
> Dear Reviewer KkVw：
>
> We sincerely thank the reviewer for the thoughtful and detailed comments. We also appreciate the strengths highlighted in the review, including that “The experiments in this paper are extensive”, and that “This paper features a clear structure and method diagram, and is easy to understand”. All modifications in the revised submission are highlighted in blue, and each revision is explicitly referenced in our responses below. Our point-by-point replies are provided next.
>
> > ### Q1: Training the T2I model and ingesting a multimodal LLM will bring additional training costs. Is it necessary to ingest two large-scale models to train the CLIP model?
>
>
> We thank the reviewer for the insightful comments. Our response addresses on the necessity of using a Multi-modal LLM and a T2I model in VLM-based continual learning.
>
> Replay-based strategies are widely used in continual learning because they consistently outperform non-replay methods and effectively mitigate forgetting. However, directly storing or replaying historical samples is often infeasible due to privacy constraints, storage limitations, or the unavailability of pre-training data. As an alternative, generative replay addresses these challenges by synthesizing past samples, typically through an auxiliary image generator that reconstructs the data distribution of previous tasks. Such generators enable replay without accessing original data and therefore avoid privacy or storage concerns.
>
> **Compared with non-replay methods, generative replay approaches, including GIFT, LoRA-Loop, and our CaRD, naturally introduce additional offline computation. We view this as an inherent trade-off between performance and cost. So incorporating a generation model is necessary to obtain the performance benefits associated with replay.**
>
> Building on this paradigm, we observe that existing generative replay methods are constrained by their dependence on class name and CLIP-style templates, which lack instance-specific semantics and often produce synthetic images that deviate substantially from real task data. To overcome this limitation, CaRD introduces instance-level, caption-guided replay. We use a Multi-modal LLM to produce captions that capture fine-grained semantic content of past samples. These captions, in turn, guide a Text-to-Image (T2I) model that is lightly adapted via task-specific LoRA to preserve the corresponding visual characteristics.
>
> **Thus, within the generative replay framework, introducing a Multi-modal LLM and a T2I model serves clear and necessary roles: the T2I model synthesizes replay samples that replace unavailable historical data, while the Multi-modal LLM enables the transition from class-level to instance-level replay, substantially improving fidelity and mitigating forgetting.**
>
> **Our experiments confirm this necessity.** As shown in Table 3, all generative replay methods (Rows 3–8) outperform the non-replay baselines (Rows 1), demonstrating the clear benefit of synthesizing past samples. Moreover, within generative replay, **instance-level captions (Rows 5 and 8) consistently outperform class name with template (Rows 3 and 6)** in both Avg. and Last. Image-quality metrics further show that instance-conditioned generation produces samples with higher fidelity.
>
>
> > ### Q2: Authors emphasize in the abstract and introduction that the simple prompts constructed by using class name and CLIP templates in existing methods have limitations, but they still use class name in the comparison in Figure 1.
>
> We thank the reviewer for highlighting this source of ambiguity. In Figure 1, **our comparison baseline in fact uses the standard class name + CLIP-style template prompts (denoted as “class name & template”)**, which is the default practice in existing generative replay methods.
>
> In the original submission, this setting was abbreviated as “Class name,” which can be misleading. We have revised Figure 1(a) and the corresponding descriptions in the main text to explicitly use “class name & template” (Lines 53, 81, and 84) to ensure precise and consistent terminology.

---

> ### Author Response · Authors · 2025-11-21
> **Author Response 2**
>
> > ### Q3: Why does this paper adopt SD3-Medium? This may cause a discussion on comparison fairness.
>
> We appreciate the reviewer’s attention to the choice of generation models and the fairness of our comparisons, and we have clarified these points comprehensively in the revised manuscript.
>
> **First, CaRD’s core conclusions do not depend on using SD3-Medium.** In the newly added Table 9 at page 10, we adopt a *matched generation model protocol*, where all methods are evaluated using the same generator (either SD1.5 or SD3-M). Under this strictly controlled setting, CaRD (Avg. 79.2, Last 88.3) still clearly outperforms GIFT (Avg. 77.2, Last 85.7) and LoRA-Loop (Avg. 77.8, Last 86.3). This demonstrates that CaRD’s improvements primarily stem from caption-guided replay and DistKD, rather than differences in generator capacity.
>
> **Second, our use of SD3-Medium in the main results is motivated by performance considerations, not by an intention to amplify performance gaps**. As shown in Table 4(b), SD3-Medium provides only moderate gains over SD1.5, though it yields slightly more stable improvements across datasets. For this reason, we adopt SD3-Medium in the primary experiments. Moreover, the revised Table 8 explicitly reports the additional replay cost introduced by SD3-Medium relative to SD1.5. We view this cost as a transparent and reasonable performance–efficiency trade-off, particularly because all auxiliary models in CaRD operate only during the offline replay-construction stage, without affecting inference-time efficiency or memory usage.
>
> **Finally, to further ensure fairness,** Table 9 (page 10) also includes GIFT* + Caption and LoRA-Loop* + Caption, which use the same QWen2.5-VL + SD3-M configuration as CaRD. Under strictly matched multimodal LLM and T2I conditions, CaRD continues to achieve the strongest performance, reinforcing that its advantages arise from the proposed methodology rather than from reliance on a particular generation model.
>
> > ### Q4: The ablation study provided by the authors do not showcase the actual baseline. When using class name and template, already outperforms all the comparison methods. Is this due to the use of a stronger baseline model?
>
> We thank the reviewer for highlighting the potential baseline confusion in Table 3. **In ablation studies, we isolate the contribution of each component by keeping all other parts fixed at their default settings.** So, Table 3 reports an internal ablation study conducted entirely within the unified CaRD training framework.
>
> Specifically, all rows in Table 3 share exactly the same training configuration—including the CLIP ViT-B/16 model, the DistKD distillation strategy, the same Multi-modal LLM and T2I generator, and identical training procedures and hyperparameters. The only factors that vary are the replay text conditions and whether LoRA adaptation is applied. Consequently, Row 3 (class name & template) should not be interpreted as an external baseline; rather, **it represents a simplified variant within a consistent CaRD framework.**
>
> To avoid ambiguity, we have strengthened the explanation in Section 4.4 of the revised manuscript.

---

> ### Author Response · Authors · 2025-11-21
> **Author Response 3**
>
> > ### Q5: What is the core innovation of DistKD? Distilling the hidden states at the middle layers of the model is a common technique.
>
> We thank the reviewer for the thoughtful comments regarding the core innovation of DistKD. We would like to clarify that DistKD is not a direct reuse of existing feature-based distillation techniques. Instead, it is motivated by our systematic observation of **multi-layer, multi-token feature drift** in VLM-based continual learning (see Fig. 1(b) and Appendix Figure 9). We find that, even with standard logit-level KD, the visual and textual encoders of CLIP exhibit substantial distributional shifts across multiple intermediate layers during sequential learning, and these shifts strongly correlate with performance degradation on previous tasks. **DistKD is designed specifically to address this failure mode—one that has not been analyzed or discussed in prior distillation or continual learning work.**
>
> Therefore, DistKD is not presented as a new distillation primitive, but as a systematic, multi-level extension tailored to this drift phenomenon. It combines (i) global distribution alignment via Wasserstein distance-based alignment, and (ii) local, token-level semantic alignment via MSE, applied jointly to both the visual and text encoders across multiple layers.
>
> This formulation follows directly from the characteristics of the drift, which DistKD is able to address whereas other middle-layer distillation methods struggle to fully mitigate. Accordingly, we further compare DistKD with other multi-layer feature distillation approaches, including Wasserstein-only and PODNet. As shown in Table 5 of the revised manuscript, both Wasserstein-only alignment (70.6 / 78.4 / 87.5 / –0.72 / 0.82 in Trans./Avg./Last/BWT/AF) and PODNet (69.5 / 77.1 / 86.0 / –4.22 / 4.82) perform noticeably worse than our DistKD (71.2 / 79.2 / 86.3 / –0.61 / 0.64). In addition, the visualizations in Figure 9 of Appendix Section B (page 19) further confirm that neither Wasserstein alignment nor PODNet effectively mitigates this drift, underscoring that DistKD specifically targets a continual-learning-related failure mode.
>
>
>
> [PODNet] Douillard A, Cord M, Ollion C, et al. Podnet: Pooled outputs distillation for small-tasks incremental learning[C]. ECCV. 2020.

---

> ### Comment · Reviewer_KkVw · 2025-11-25
> **Response to Rebuttal**
>
> Thank you for your response. Table 8 in the revised manuscript presents a detailed cost comparison. May I ask whether the full-shot setting also involves fine-tuning the generative model using only 5 images?

---

> > ### Author Response · Authors · 2025-11-26
> > **Author Response**
> >
> > Thank you for the follow-up question. We would like to clarify that in the full-shot setting, the T2I model is fine-tuned using **all training images** of each task, rather than only 5 images.
> >
> > The T2I fine-tuning times in Table 8 are similar between the full-shot and 5-shot settings because we fix the training schedule to 1000 iterations with batch size = 4 for all settings. This means that even if the number of available training images differs, the optimizer still runs for the same number of updates, leading to similar wall-clock durations.
> >
> > We did not perform separate hyperparameter tuning for different settings, and all timing results in Table 8 were measured under this unified configuration to ensure consistency and fairness in comparison.

---

> > > ### Comment · Reviewer_KkVw · 2025-11-26
> > > **Official Comment by Reviewer KkVw**
> > >
> > > The authors' rebuttal largely addressed my concerns, but the practical value of using two large-scale models to assist CLIP in continual learning remains debatable. I will consider raising my score.

---

> > > > ### Author Response · Authors · 2025-11-26
> > > > **Author Response**
> > > >
> > > > We are pleased that our response has addressed most of your concerns, and we sincerely appreciate the thoughtful follow-up. Regarding the question of practical value, we offer the following clarification. **In brief, although CaRD utilizes two auxiliary models, their computation is offline and does not affect CLIP’s training or inference efficiency. Moreover, this offline cost can be further reduced by using smaller models or more advanced acceleration techniques.**
> > > >
> > > > We would like to clarify that CaRD employs a Multi-modal LLM and a T2I model **only once during an offline pre-processing stage** before CLIP training begins. These auxiliary models are never involved during CLIP training or inference, and thus introduce no additional memory or latency requirements at deployment.
> > > >
> > > > **This offline paradigm is also common in prior work.** For example, generative replay approaches such as GIFT and LoRA-Loop rely on T2I models to synthesize replay data in CLIP-based continual learning, while methods like [R1] and [R2] use large language models (LLMs) to enrich prompts for CLIP-based zero-/few-shot recognition.
> > > >
> > > > **Importantly, the auxiliary models in CaRD are modular and plug-and-play.** They can be replaced with lighter MLLMs or T2I models, or accelerated using techniques such as model quantization or few-step image generators[R3,R4]. Because all computation occurs offline, CaRD can directly benefit from such improvements without modifying the caption-guided replay pipeline.
> > > >
> > > > Taken together, this design provides a reasonable balance between practicality and performance: auxiliary models improve replay fidelity, while their cost is incurred offline and can be further reduced with smaller or more efficient models.
> > > >
> > > >
> > > > [R1] Yang, Haoyuan, et al. "ImagineFSL: Self-Supervised Pretraining Matters on Imagined Base Set for VLM-based Few-shot Learning." Proceedings of the Computer Vision and Pattern Recognition Conference. 2025.
> > > >
> > > > [R2] Tian, Yonglong, et al. "Learning vision from models rivals learning vision from data." Proceedings of the IEEE/CVF conference on computer vision and pattern recognition. 2024.
> > > >
> > > > [R3] https://huggingface.co/stabilityai/sdxl-turbo
> > > >
> > > > [R4] https://huggingface.co/tensorart/stable-diffusion-3.5-medium-turbo

---

> > > > > ### Author Response · Authors · 2025-11-28
> > > > > **Additional Author Response**
> > > > >
> > > > > We sincerely thank you again for the thoughtful follow-up. To better address your concern regarding the practical value, **we have extended our analysis by adding two additional configurations that use smaller Multi-modal LLM and T2I model.** Specifically, we evaluate **SmolVLM (0.5B) and InternVL3.5 (1B) paired with the SD1.5 (0.9B).** The corresponding results are provided in the table below.
> > > > >
> > > > > These new experiments show that CaRD remains robust even when the captioner is replaced with substantially smaller models. Although using lighter captioners leads to a modest drop in performance, the results remain comparable to those obtained with larger captioners, while caption-generation time is reduced substantially (for example, QWen2.5-VL 7B to SmolVLM 0.5B decreases from 13.6 minutes to 1.8 minutes). Importantly, under the configuration of SmolVLM (0.5B) + SD1.5 (0.9B), CaRD operates at a similar generation model scale as GIFT (SD1.5, 0.9B) and LoRA-Loop (SD1.5, 0.9B), yet still maintains competitive performance. **These observations suggest that CaRD’s improvements primarily stem from its caption-guided replay rather than the scale of the auxiliary models, and also shows that CaRD’s offline cost can be further reduced by adopting smaller models when needed.**
> > > > >
> > > > > We hope this additional analysis helps address your concern.
> > > > >
> > > > >
> > > > > | Method | Multi-Modal LLM (#Params) | Caption Gen. (min) | T2I Model (#Params) | FT T2I (min) | Image Gen. (min) | Replay Total Time (min) | Training (min) | Total Time (min) | Storage Cost (MB) | Trans. / Avg. / Last |
> > > > > |:--------|:--------------------------:|:-------------------:|:---------------------:|:------------:|:-----------------:|:------------------------:|:---------------:|:------------------:|:--------------------:|:----------------------:|
> > > > > | *Full-shot* | | | | | | | | | | |
> > > > > | GIFT (C&T) | – | – | SD1.5 (0.9B) | –   | 11.6 | 11.6 | 6.2 | 17.8 (**×1.0**) | 0.2  | 69.3 / 77.3 / 86.0 |
> > > > > | LoRA-Loop (C&T) | – | – | SD1.5 (0.9B) | 5.1 | 11.6 | 16.7 | 6.2 | 22.9 (**×1.3**) | 30.2 | 69.8 / 77.6 / 86.0 |
> > > > > | LoRA-Loop (C&T) | – | – | SD3-M (2B)   | 8.2 | 12.4 | 20.6 | 6.2 | 26.8 (**×1.5**) | 47.2 | 70.0 / 77.7 / 86.1 |
> > > > > | CaRD | **SmolVLM (0.5B)** | 1.8 | SD1.5 (0.9B) | 5.1 | 11.6 | 18.5 | 6.7 | 25.2 (**×1.4**) | 34.1 | 70.6 / 78.6 / 87.5 |
> > > > > | CaRD | **InternVL3.5 (1B)** | 3.9 | SD1.5 (0.9B) | 5.1 | 11.6 | 20.6 | 6.7 | 27.3 (**×1.5**) | 34.2 | 70.8 / 78.9 / 87.9 |
> > > > > | CaRD | QWen2.5-VL (3B) | 8.4  | SD3-M (2B)   | 8.2 | 12.4 | 29.0 | 6.7 | 35.7 (**×2.0**) | 49.2 | 71.1 / 79.2 / 88.2 |
> > > > > | CaRD | QWen2.5-VL (7B) | 13.6 | SD1.5 (0.9B) | 5.1 | 11.6 | 30.3 | 6.7 | 37.0 (**×2.1**) | 34.2 | 71.0 / 79.1 / 88.0 |
> > > > > | CaRD | QWen2.5-VL (7B) | 13.6 | SD3-M (2B)   | 8.2 | 12.4 | 34.2 | 6.7 | 40.9 (**×2.3**) | 49.2 | **71.2** / **79.2** / **88.3** |
> > > > >
> > > > > [SmoVLM] Marafioti, Andrés, et al. "Smolvlm: Redefining small and efficient multimodal models." arXiv preprint arXiv:2504.05299 (2025). https://huggingface.co/HuggingFaceTB/SmolVLM-500M-Instruct
> > > > >
> > > > > [InternVL3.5] Wang, Weiyun, et al. "Internvl3. 5: Advancing open-source multimodal models in versatility, reasoning, and efficiency." arXiv preprint arXiv:2508.18265 (2025). https://huggingface.co/OpenGVLab/InternVL3_5-1B

---

### Official Review · Reviewer_LRVX · 2025-10-31

**Soundness:** 3
**Presentation:** 3
**Contribution:** 2
**Rating:** 4
**Confidence:** 4

**Summary:**

This paper proposes CaRD to address catastrophic forgetting in VLMs for continual learning. The method has two key components: a caption-guided replay paradigm that reconstructs high-fidelity past images using instance-level captions and a LoRA-adapted T2I model, and a multi-layer distribution distillation method to mitigate the feature drift. Experiments show the method outperforms existing approaches on the MTIL and X-TAIL benchmarks.

**Strengths:**

1. The paper is well-written and the proposed method is presented clearly.
2. The proposed caption-guided replay paradigm is a well-motivated and intuitive innovation and the visualization of feature drift in Fig. 1b is insightful.
3. The experimental evaluation is thorough including MTIL and X-TAIL with full and few-shot settings.

**Weaknesses:**

1. The method introduces significant computational overhead, which raises questions about its practical feasibility. For each new task, it requires running a large VLM to generate captions for all training samples and separately fine-tuning a T2I model to obtain task-specific LoRA weights.
2. The novelty of the distribution distillation component is limited. While successfully applied to VLM continual learning, the core idea of matching the mean and covariance of feature maps is a well-explored concept in knowledge distillation by Lv et al [1].
3. The related work section has some omissions. The discussion on replay could be strengthened by including methods based on a series of prototype augmentation methods [2] [3]. The method LADA, which is used as a key comparison baseline in Tab.10 and 11, is not introduced in the related work section.
4. The paper's presentation could be strengthened by reorganizing the main experimental results. The authors might consider moving the X-TAIL full-shot results into the main paper while retaining the MTIL full-shot results. The MTIL 5-shot setting, while valuable, could be moved to the appendix to make space for the more challenging task-agnostic X-TAIL benchmark.


Reference:

[1] Lv, Jiaming, Haoyuan Yang, and Peihua Li. "Wasserstein distance rivals kullback-leibler divergence for knowledge distillation." *Advances in Neural Information Processing Systems* 37 (2024).

[2] Zhu, Fei, et al. "Prototype augmentation and self-supervision for incremental learning." *Proceedings of the IEEE/CVF conference on computer vision and pattern recognition*. 2021.

[3] Asadi, Nader, et al. "Prototype-sample relation distillation: towards replay-free continual learning." *International conference on machine learning*. PMLR, 2023.

[4] Luo, Mao-Lin, et al. "LADA: Scalable Label-Specific CLIP Adapter for Continual Learning." *Forty-second International Conference on Machine Learning*. PMLR, 2025.

**Questions:**

Please refer to weakness.

---

> ### Author Response · Authors · 2025-11-21
> **Author Response 1**
>
> Dear Reviewer LRVX：
>
> We sincerely thank the reviewer for the thoughtful and detailed feedback. We also appreciate the strengths highlighted in the review, including that “The paper is well-written and the proposed method is presented clearly,” that “The proposed caption-guided replay paradigm is a well-motivated,” that “the visualization of feature drift is insightful,” and that “The experimental evaluation is thorough”. We have revised the manuscript accordingly, with all changes highlighted in blue. Our point-by-point responses are provided below.

---

> ### Author Response · Authors · 2025-11-21
> **Author Response 2**
>
> > ### Q1: The method introduces significant computational overhead, which raises questions about its practical feasibility.
>
> We thank the reviewer for the insightful comments regarding computational overhead and practical feasibility. We note that **CaRD introduces no additional inference-time cost and that its overall computational complexity is comparable to existing generative replay approaches.** Like GIFT and LoRA-Loop, CaRD synthesizes replay samples **once during an offline pre-processing stage** before CLIP training begins. In generative replay, a generation model is used to synthesize images that approximate the data distribution of previous tasks, enabling replay without accessing original real data to avoid data privacy or storage concerns. This offline construction does not participate in CLIP’s training loop or inference, and all synthesized samples are discarded immediately after training, resulting in **no long-term storage overhead.**
>
> We then provide a detailed analysis of computation cost. In the original submission, Table 8 indeed reported only the per-step CLIP training time and did not clearly account for the costs of each replay stage. **In the revised manuscript, we have fully redesigned Table 8 (highlighted in blue) in section 4.4.3 at page 10 (and also reproduced below for convenience)**, to incorporate all components of CaRD’s computational overhead, including (1) Caption generation, (2) LoRA fine-tuning of the T2I model, (3) Synthetic image generation, and (4) CLIP training. All results are averaged over the full sequence of 11 MTIL tasks.
>
> In this revised table, Replay Total Time corresponds to the actual wall-clock time for stages (1)–(3), while *Total Time* reflects the sum of replay construction and CLIP training. **All timing measurements are obtained under a unified experimental setup**, for example, using vLLM for efficient Multi-modal LLM inference, TensorRT acceleration for T2I generation, and consistent LoRA hyperparameters across methods, to ensure a fair comparison.
>
> Overall, the revised Table provides a complete and transparent view of computational cost. CaRD’s replay construction cost remains comparable to other generative replay methods. Importantly. The captioning cost can be further reduced by using smaller MLLMs (13.6 min → 8.4 when QWen2.5-VL 7B → 3B), and in 5-shot settings it becomes almost negligible (0.5 min) due to the limited number of training images. **Furthermore, employing smaller captioning or T2I models, or adopting more advanced acceleration techniques, can further reduce replay construction cost.** These improvements are orthogonal to CaRD’s core representation and replay mechanisms and do not alter the method’s fundamental contributions.
>
>
> | Method | Multi-Modal LLM (#Params) | Caption Gen. (min) | T2I Model (#Params) | FT T2I (min) | Image Gen. (min) | Replay Total Time (min) | Training (min) | Total Time (min) | Storage Cost (MB) | Trans. / Avg. / Last |
> |:--------|:--------------------------:|:-------------------:|:---------------------:|:------------:|:-----------------:|:------------------------:|:---------------:|:------------------:|:--------------------:|:----------------------:|
> | *Full-shot* | | | | | | | | | | |
> | GIFT (C&T) | – | – | SD1.5 (0.9B) | –   | 11.6 | 11.6 | 6.2 | 17.8 (**×1.0**) | 0.2  | 69.3 / 77.3 / 86.0 |
> | LoRA-Loop (C&T) | – | – | SD1.5 (0.9B) | 5.1 | 11.6 | 16.7 | 6.2 | 22.9 (**×1.3**) | 30.2 | 69.8 / 77.6 / 86.0 |
> | LoRA-Loop (C&T) | – | – | SD3-M (2B)   | 8.2 | 12.4 | 20.6 | 6.2 | 26.8 (**×1.5**) | 47.2 | 70.0 / 77.7 / 86.1 |
> | CaRD | QWen2.5-VL (3B) | 8.4  | SD3-M (2B)   | 8.2 | 12.4 | 29.0 | 6.7 | 35.7 (**×2.0**) | 49.2 | 71.1 / 79.2 / 88.2 |
> | CaRD | QWen2.5-VL (7B) | 13.6 | SD1.5 (0.9B) | 5.1 | 11.6 | 30.3 | 6.7 | 37.0 (**×2.1**) | 34.2 | 71.0 / 79.1 / 88.0 |
> | CaRD | QWen2.5-VL (7B) | 13.6 | SD3-M (2B)   | 8.2 | 12.4 | 34.2 | 6.7 | 40.9 (**×2.3**) | 49.2 | 71.2 / 79.2 / 88.3 |
> |  |  |  |  |  |  |  |  |  |  |  |
> | *5-shot* | | | | | | | | | | |
> | GIFT (C&T) | – | – | SD3-M (2B)   | –   | 12.4 | 12.4 | 6.2 | 18.6 (**×1.0**) | 0.2  | 69.0 / 72.5 / 77.8 |
> | LoRA-Loop (C&T) | – | – | SD3-M (2B)   | 8.2 | 12.4 | 20.6 | 6.2 | 26.8 (**×1.4**) | 47.2 | 69.2 / 72.9 / 78.0 |
> | CaRD | QWen2.5-VL (7B) | 0.5 | SD3-M (2B)   | 8.2 | 12.4 | 21.1 | 6.7 | 27.8 (**×1.5**) | 47.6 | 70.4 / 73.2 / 79.2 |
>
>
> [vLLM] Kwon, Woosuk, et al. "Efficient memory management for large language model serving with pagedattention." Proceedings of the 29th symposium on operating systems principles. 2023. https://docs.vllm.ai/en/latest/
>
> [TensorRT] https://github.com/NVIDIA/TensorRT/tree/main/demo/Diffusion

---

> ### Author Response · Authors · 2025-11-21
> **Author Response 3**
>
> > ### Q2: The novelty of the distribution distillation component is limited. While successfully applied to VLM continual learning, the core idea of matching the mean and covariance of feature maps is a well-explored concept in knowledge distillation by Lv et al [1].
>
> We thank the reviewer for raising the concern regarding the novelty of our distribution distillation component. Our intention is not to claim that the use of Wasserstein distance is novel—indeed, prior work such as Lv et al. has indeed explored distribution-based distillation for classic image classification. Rather, the contribution of DistKD lies in two aspects:
>
> **(1) A systematic observation and characterization of a previously unreported failure mode specific to VLM-based continual learning.**
>
> Our analysis (Figure 1(b), Appendix Figure 9) reveals a multi-layer, multi-token drift phenomenon that emerges when CLIP is finetuned sequentially. **To our knowledge, this drift behavior has not been documented before in the distillation or continual learning literature.** Once this phenomenon was identified, it was natural and well-motivated to explore distribution-alignment tools to counteract the drift.
>
> **(2) A complementary, multi-level distillation formulation tailored to the structure of VLMs.**
>
> Wasserstein distance-based alignment is an effective and established choice for capturing global geometric of feature distributions. However, applying it directly is insufficient in the VLM setting: CLIP representations are tokenized, multi-layer, and multimodal; feature drift occurs inconsistently across layers and across modalities, and global alignment alone fails to preserve fine-grained token semantics.
>
> **Therefore, DistKD is not presented as a new distillation primitive, but as a systematic, multi-level extension tailored to this drift phenomenon.** It combines (i) global distribution alignment via Wasserstein distance-based alignment, and (ii) local, token-level semantic alignment via MSE, applied jointly to both the visual and text encoders across multiple layers.
>
> This formulation directly follows from the characteristics of the drift. As shown in Table 5, applying only Wasserstein distance-based alignment or only Hint alignment fails to fully correct the drift, whereas the combined formulation yields significantly higher stability across tasks. Additional visualizations in Figure 3 of Appendix Section B (page 19) further indicate that Wasserstein alignment alone cannot suppress the observed drift, highlighting that DistKD targets a continual-learning-specific challenge rather than reusing Wasserstein distance-based distillation (Wass).
>
> > ### Q3: The discussion on replay could be strengthened by including methods based on a series of prototype augmentation methods [2] [3]. The method LADA, which is used as a key comparison baseline in Tab.10 and 11, is not introduced in the related work section.
>
> We thank the reviewer for pointing out the omissions in our related work section. Following the suggestion, we have updated Section 2 (Continual Learning for VLMs) on page 3, with all additions highlighted in blue in the revised manuscript.
>
> First, we have added a description of LADA, ensuring consistency with its role as a key comparison baseline in Tables 2 and 12. The revised text explains that LADA further advances Architecture-based CL methods by introducing label-specific adapters that attach per-class memory units to the frozen CLIP backbone, improving plasticity without relying on task identifiers while leveraging distillation to stabilize representations.
>
> Second, we expanded the discussion of replay-based methods by including representative prototype-augmentation approaches [2,3]. These methods preserve a compact set of class prototypes and augment them through feature perturbation or prototype–sample relational constraints, thereby approximating the feature distribution of past tasks.
>
>
> > ### Q4: Suggest reorganizing the main experimental results by moving the X-TAIL full-shot results into the main paper and moving the MTIL 5-shot setting to the appendix.
>
>
>
> Thank you for the helpful suggestion regarding the organization of the experimental results. We agree that the task-agnostic X-TAIL benchmark provides a more challenging and informative evaluation of continual learning performance. In the revised manuscript, we have reorganized the main results accordingly—specifically in Section 4.1, Section 4.3 on page 6, and Table 2 on page 7 (highlighted in blue): the X-TAIL full-shot results have been moved from the appendix into the main paper alongside the MTIL full-shot results, while the MTIL 5-shot results have been relocated to Appendix Section C to keep the main text focused on the most representative experimental settings. We believe this restructuring improves clarity and better highlights the key empirical findings.

---

> > ### Comment · Reviewer_LRVX · 2025-11-28
> > **Endorsing the Value of Rebuttal and Raising the Evaluation Score**
> >
> > First and foremost, I would like to express my gratitude to the authors for their thoughtful and comprehensive rebuttal. The supplementary experiments presented in their response are not only well-designed and targeted, but also sufficiently persuasive to effectively address and resolve all the questions I raised earlier.
> >
> > I fully concur that the phenomenon of multi-layer and multi-token drift emerging during the sequential fine-tuning of CLIP holds significant academic value. The in-depth analysis of this intrinsic issue, coupled with the proposed solution, is anticipated to generate far-reaching implications for the research field of CLIP-based continual learning.
> >
> > Considering the authors' rigorous response and the substantial improvements of the work, I have decided to raised my evaluation score to 6.

---

> > > ### Author Response · Authors · 2025-11-28
> > > **Acknowledgment of Reviewer Feedback**
> > >
> > > Thank you very much for the positive and encouraging feedback. We sincerely appreciate the time and care you devoted to reviewing our work. We are glad that the additional analyses and experiments were helpful in addressing your concerns. Your constructive feedback has greatly improved the clarity and quality of our paper, and we are truly grateful for your support.

---

### Official Review · Reviewer_J4Rf · 2025-10-31

**Soundness:** 3
**Presentation:** 3
**Contribution:** 2
**Rating:** 4
**Confidence:** 4

**Summary:**

The authors introduce CaRD, a methodology designed to mitigate catastrophic forgetting in Continual Learning using Vision-Language Models (VLMs). CaRD is characterized by the following components:

 - Generative Replay: the method leverages QWen2.5-VL to generate captions for each instance, which are stored in a replay buffer. A Text-to-Image (T2I) model -- specifically, Stable Diffusion 3 (medium) -- is then trained to synthesize high-quality images for the current task. For each task, the T2I model is fine-tuned with a distinct set of LoRA modules, which are preserved across tasks. During subsequent training, these task-specific LoRA adapters can be reloaded individually to generate accurate synthetic samples representing all previous tasks.

- Knowledge Distillation, which employs the model from the most recent task as the teacher. The distillation loss combines, in a layer-wise fashion, the Wasserstein distance between feature statistics (means and covariances) and the squared norm of the differences between raw feature representations.

Experiments are carried out on standard VLM continual learning benchmarks, such as MTIL, under both full-shot and 5-shot configurations, and evaluated across two different task orderings.

**Strengths:**

- Experiments have been conducted on a variety of scenarios, and additional metrics are provided in the Appendix, validating the proposed approach. CaRD outperforms all competitors in standard benchmarks for CL with VLMs.
- The paper is well written and easy to follow.
- The ablation studies justify many of the authors' choices for their pipeline.

**Weaknesses:**

1. Total training time: in Table 8, the authors report that CaRD does not introduce a substantial computational overhead. However, it appears that the reported training time only accounts for a single forward and backward pass, hence reflecting only the minor overhead caused by the additional per-layer loss terms. It is unlikely that the table includes the time required to fine-tune the diffusion model. A more comprehensive comparison of CaRD’s total training time against that of other methods would be valuable to fairly assess its overall efficiency and trade-offs.

2. Model sizes: CaRD relies on two large auxiliary models -- a Text-to-Image (T2I) model, Stable Diffusion 3-medium (≈2B parameters), and a Multimodal LLM (MLLM) captioner, QWen2.5-VL (≈7B parameters) -- totaling roughly 9B parameters external to the trained model. In contrast, the main VLM used for continual learning is CLIP with a ViT-B/16 vision backbone, which contains around 150M parameters. This means that the auxiliary "helper" models have roughly 60 times more parameters than the model being trained, giving CaRD an unfair advantage over competing methods that do not employ such massive external models.

**Questions:**

See "Weaknesses".

---

> ### Author Response · Authors · 2025-11-21
> **Author Response 1**
>
> Dear Reviewer J4Rf：
>
> We sincerely thank the reviewer for the thoughtful and detailed comments. We also appreciate the strengths highlighted in the review, including that “Experiments have been conducted on a variety of scenarios”, and that “The paper is well written and easy to follow”. The concerns regarding computational cost and auxiliary model fairness are highly valuable, and we have substantially revised the manuscript accordingly. All modifications in the revised submission are highlighted in blue, and each revision is explicitly referenced in our responses below. Our point-by-point replies are provided next.

---

> ### Author Response · Authors · 2025-11-21
> **Author Response 2**
>
> > ### Q1: A more comprehensive comparison of CaRD’s total training time against that of other methods would be valuable to fairly assess its overall efficiency and trade-offs.
>
> We thank the reviewer for the constructive feedback. Kindly noted that, CaRD is a generative replay–based method, and therefore should be compared primarily against methods of the same category when analyzing cost and performance. **Importantly, CaRD introduces no additional inference-time cost and that its overall computational complexity is comparable to existing generative replay approaches.** Like GIFT and LoRA-Loop, CaRD synthesizes replay samples **once during an offline pre-processing stage** before CLIP training begins. In generative replay, a generation model is used to synthesize images that approximate the data distribution of previous tasks, enabling replay without accessing original real data to avoid data privacy or storage concerns. This offline construction does not participate in CLIP’s training loop or inference, and all synthesized samples are discarded immediately after training, resulting in **no long-term storage overhead.**
>
> We then provide a detailed analysis of computation cost. In the original submission, Table 8 indeed reported only the per-step CLIP training time and did not clearly account for the costs of each replay stage. **In the revised manuscript, we have fully redesigned Table 8 (highlighted in blue) in section 4.4.3 at page 10 (and also reproduced below for convenience)**, to incorporate all components of CaRD’s computational overhead, including (1) Caption generation, (2) LoRA fine-tuning of the T2I model, (3) Synthetic image generation, and (4) CLIP training. All results are averaged over the full sequence of 11 MTIL tasks.
>
> In this revised table, Replay Total Time corresponds to the actual wall-clock time for stages (1)–(3), while *Total Time* reflects the sum of replay construction and CLIP training. **All timing measurements are obtained under a unified experimental setup**, for example, using vLLM for efficient Multi-modal LLM inference, TensorRT acceleration for T2I generation, and consistent LoRA hyperparameters across methods, to ensure a fair comparison.
>
> Overall, the revised Table provides a complete and transparent view of computational cost. CaRD’s replay construction cost remains comparable to other generative replay methods. Importantly. The captioning cost can be further reduced by using smaller MLLMs (13.6 min → 8.4 when QWen2.5-VL 7B → 3B), and in 5-shot settings it becomes almost negligible (0.5 min) due to the limited number of training images. **Furthermore, employing smaller captioning or T2I models, or adopting more advanced acceleration techniques, can further reduce replay construction cost.** These improvements are orthogonal to CaRD’s core representation and replay mechanisms and do not alter the method’s fundamental contributions.
>
>
> | Method | Multi-Modal LLM (#Params) | Caption Gen. (min) | T2I Model (#Params) | FT T2I (min) | Image Gen. (min) | Replay Total Time (min) | Training (min) | Total Time (min) | Storage Cost (MB) | Trans. / Avg. / Last |
> |:--------|:--------------------------:|:-------------------:|:---------------------:|:------------:|:-----------------:|:------------------------:|:---------------:|:------------------:|:--------------------:|:----------------------:|
> | *Full-shot* | | | | | | | | | | |
> | GIFT (C&T) | – | – | SD1.5 (0.9B) | –   | 11.6 | 11.6 | 6.2 | 17.8 (**×1.0**) | 0.2  | 69.3 / 77.3 / 86.0 |
> | LoRA-Loop (C&T) | – | – | SD1.5 (0.9B) | 5.1 | 11.6 | 16.7 | 6.2 | 22.9 (**×1.3**) | 30.2 | 69.8 / 77.6 / 86.0 |
> | LoRA-Loop (C&T) | – | – | SD3-M (2B)   | 8.2 | 12.4 | 20.6 | 6.2 | 26.8 (**×1.5**) | 47.2 | 70.0 / 77.7 / 86.1 |
> | CaRD | QWen2.5-VL (3B) | 8.4  | SD3-M (2B)   | 8.2 | 12.4 | 29.0 | 6.7 | 35.7 (**×2.0**) | 49.2 | 71.1 / 79.2 / 88.2 |
> | CaRD | QWen2.5-VL (7B) | 13.6 | SD1.5 (0.9B) | 5.1 | 11.6 | 30.3 | 6.7 | 37.0 (**×2.1**) | 34.2 | 71.0 / 79.1 / 88.0 |
> | CaRD | QWen2.5-VL (7B) | 13.6 | SD3-M (2B)   | 8.2 | 12.4 | 34.2 | 6.7 | 40.9 (**×2.3**) | 49.2 | 71.2 / 79.2 / 88.3 |
> |  |  |  |  |  |  |  |  |  |  |  |
> | *5-shot* | | | | | | | | | | |
> | GIFT (C&T) | – | – | SD3-M (2B)   | –   | 12.4 | 12.4 | 6.2 | 18.6 (**×1.0**) | 0.2  | 69.0 / 72.5 / 77.8 |
> | LoRA-Loop (C&T) | – | – | SD3-M (2B)   | 8.2 | 12.4 | 20.6 | 6.2 | 26.8 (**×1.4**) | 47.2 | 69.2 / 72.9 / 78.0 |
> | CaRD | QWen2.5-VL (7B) | 0.5 | SD3-M (2B)   | 8.2 | 12.4 | 21.1 | 6.7 | 27.8 (**×1.5**) | 47.6 | 70.4 / 73.2 / 79.2 |
>
>
> [vLLM] Kwon, Woosuk, et al. "Efficient memory management for large language model serving with pagedattention." Proceedings of the 29th symposium on operating systems principles. 2023. https://docs.vllm.ai/en/latest/
>
> [TensorRT] https://github.com/NVIDIA/TensorRT/tree/main/demo/Diffusion

---

> ### Author Response · Authors · 2025-11-21
> **Author Response 3**
>
> > ### Q2: Whether CaRD gains an unfair advantage because it relies on two large auxiliary models—approximately 9B parameters in total—which are around 60× larger than the CLIP model being trained.
>
> We thank the reviewer for raising the concerns regarding auxiliary model size and fairness. As noted in our response to Q1, CaRD is a generative replay–based method and therefore should be compared primarily against methods of the same category when analyzing cost and performance. In generative replay, an auxiliary generation model is typically used to synthesize images that approximate the data distribution of previous tasks, enabling replay without accessing original real data and thereby avoiding privacy or storage risks. **Compared with non-replay methods that do not use such auxiliary models, generative replay methods (including GIFT, LoRA-Loop, and our CaRD) naturally incur additional offline cost. However, we believe that these non-replay methods can also benefit from the synthetic replay data generated by our approach.**
>
> To ensure a fair comparison **within generative replay methods**, we provide **Table 9 in section 4.4.4 at page 10 (and also reproduced below for convenience)**, which adopts a *matched generation model* setting: within each panel, all methods use exactly the same generator. Under this controlled setup, the following observations emerge:
>
> 1) **When all methods use the same Multi-modal LLM + T2I combination (QWen2.5-VL (7B) + SD3-M (2B)), CaRD consistently achieves the best performance.**
> Under identical generators, GIFT* + Caption and LoRA-Loop* + Caption obtain Avg./Last scores of 77.6 / 86.3 and 78.3 / 86.9, respectively, whereas CaRD reaches 79.2 / 88.3 with substantially improved BWT/AF (–0.61 / 0.64). **This indicates that CaRD’s advantage stems from its caption-guided replay and DistKD strategy rather than from larger auxiliary models.**
>
> 2) **CaRD remains strong even when the generator capacity is reduced.**
> Reducing CaRD’s generators from QWen2.5-VL (7B) to **3B**, or replacing SD3-M (2B) with **SD1.5 (0.9B)**, still yields performance that surpasses other methods using larger generators. This further confirms that CaRD does not rely on particularly large auxiliary models.
>
> Overall, within the generative replay setting, CaRD’s improvements are not the result of an “unfair advantage” in auxiliary model size . Furthermore, Table 8 reports the full parameter sizes, replay costs, and storage overheads of all auxiliary models, making these trade-offs fully transparent.
>
> | Method | Replay Data | Generation Model | FT T2I | Trans. | Avg. | Last | BWT | AF |
> |--------|-------------|------------------|--------|----------|--------|--------|--------|-------|
> | *GIFT* | Class name & template | SD1.5 (0.9B) |  | 69.3 | 77.3 | 86.0 | -- | -- |
> | LoRA-Loop | Class name & template | SD1.5 (0.9B) | ✓ | 69.8 | 77.6 | 86.0 | -- | -- |
> ||
> | *GIFT* (reprod.) | Class name & template | SD3-M (2B) |  | 69.7 | 77.2 | 85.7 | -2.01 | 2.53 |
> | LoRA-Loop* | Class name & template | SD3-M (2B) | ✓ | 70.0 | 77.7 | 86.1 | -1.92 | 2.31 |
> ||
> | GIFT* + Caption | Instance-level Caption | QWen2.5-VL (7B) + SD3-M (2B) |  | 69.7 | 77.6 | 86.3 | -1.59 | 1.67 |
> | LoRA-Loop* + Caption | Instance-level Caption | QWen2.5-VL (7B) + SD3-M (2B) | ✓ | 70.2 | 78.3 | 86.9 | -1.47 | 1.39 |
> ||
> | **CaRD** | Instance-level Caption | QWen2.5-VL (7B) + SD3-M (2B) | ✓ | **71.2** | **79.2** | **88.3** | **-0.61** | **0.64** |
> | **CaRD** | Instance-level Caption | QWen2.5-VL (3B) + SD3-M (2B) | ✓ | 71.1 | 79.2 | 88.2 | -0.65 | 0.71 |
> | **CaRD** | Instance-level Caption | QWen2.5-VL (7B) + SD3-M (0.9B) | ✓ | 71.0 | 79.1 | 88.0 | -0.80 | 0.83 |

---

> > ### Comment · Reviewer_J4Rf · 2025-11-26
> >
> > Thank you for the thorough and detailed response, and for the expanded tables covering replay construction time, storage, and matched-generator comparisons. These additions significantly clarify the computational profile of CaRD and help isolate the contribution of the method from the auxiliary models.
> >
> > I still have one remaining concern regarding *resource symmetry*. Although I agree that CaRD should primarily be compared to other generative-replay methods, most existing approaches in this category operate with a much smaller auxiliary footprint. CaRD’s reliance on a large captioner (3B–7B) still places it at a **substantially different** memory scale.
> >
> > To better contextualize CaRD’s real-world applicability, and to allow a fully fair comparison of *algorithmic gain* under a matched auxiliary budget, I would appreciate seeing an additional setting where CaRD uses a **very small captioner** ($\approx$ 1B or similarly lightweight), paired with the smaller diffusion model (0.9B) you already evaluated. This would basically save some memory on the generator and let you have a captioner "for free".
> >
> > Even if this does not surpass state of the art performance, it would be useful as it would show how much of CaRD’s robustness stems from the core method versus auxiliary model capacity.

---

> > > ### Author Response · Authors · 2025-11-28
> > > **Author Response**
> > >
> > > We are pleased that our earlier response addressed several of your concerns, and we sincerely thank you for the thoughtful suggestion regarding evaluating CaRD under a more matched auxiliary model budget. **Following this suggestion, we have added two new configurations to the table below that using smaller captioners, specifically SmolVLM (0.5B) and InternVL3.5 (1B), paired with the SD1.5 (0.9B) diffusion model.**
> > >
> > > These new results demonstrate that CaRD remains robust even when the captioner is replaced with much smaller models. Although using lighter captioners leads to a modest drop in performance, the results remain comparable to those obtained with larger captioners, while caption-generation time is reduced substantially (for example, QWen2.5-VL 7B to SmolVLM 0.5B decreases from 13.6 minutes to 1.8 minutes). **These observations suggest that CaRD’s improvements primarily stem from its caption-guided replay, rather than from the scale of the auxiliary models.**
> > >
> > > We hope this additional analysis helps address your concern.
> > >
> > > | Method | Multi-Modal LLM (#Params) | Caption Gen. (min) | T2I Model (#Params) | FT T2I (min) | Image Gen. (min) | Replay Total Time (min) | Training (min) | Total Time (min) | Storage Cost (MB) | Trans. / Avg. / Last |
> > > |:--------|:--------------------------:|:-------------------:|:---------------------:|:------------:|:-----------------:|:------------------------:|:---------------:|:------------------:|:--------------------:|:----------------------:|
> > > | *Full-shot* | | | | | | | | | | |
> > > | GIFT (C&T) | – | – | SD1.5 (0.9B) | –   | 11.6 | 11.6 | 6.2 | 17.8 (**×1.0**) | 0.2  | 69.3 / 77.3 / 86.0 |
> > > | LoRA-Loop (C&T) | – | – | SD1.5 (0.9B) | 5.1 | 11.6 | 16.7 | 6.2 | 22.9 (**×1.3**) | 30.2 | 69.8 / 77.6 / 86.0 |
> > > | LoRA-Loop (C&T) | – | – | SD3-M (2B)   | 8.2 | 12.4 | 20.6 | 6.2 | 26.8 (**×1.5**) | 47.2 | 70.0 / 77.7 / 86.1 |
> > > | CaRD | **SmolVLM (0.5B)** | 1.8 | SD1.5 (0.9B) | 5.1 | 11.6 | 18.5 | 6.7 | 25.2 (**×1.4**) | 34.1 | 70.6 / 78.6 / 87.5 |
> > > | CaRD | **InternVL3.5 (1B)** | 3.9 | SD1.5 (0.9B) | 5.1 | 11.6 | 20.6 | 6.7 | 27.3 (**×1.5**) | 34.2 | 70.8 / 78.9 / 87.9 |
> > > | CaRD | QWen2.5-VL (3B) | 8.4  | SD3-M (2B)   | 8.2 | 12.4 | 29.0 | 6.7 | 35.7 (**×2.0**) | 49.2 | 71.1 / 79.2 / 88.2 |
> > > | CaRD | QWen2.5-VL (7B) | 13.6 | SD1.5 (0.9B) | 5.1 | 11.6 | 30.3 | 6.7 | 37.0 (**×2.1**) | 34.2 | 71.0 / 79.1 / 88.0 |
> > > | CaRD | QWen2.5-VL (7B) | 13.6 | SD3-M (2B)   | 8.2 | 12.4 | 34.2 | 6.7 | 40.9 (**×2.3**) | 49.2 | **71.2** / **79.2** / **88.3** |
> > >
> > > [SmoVLM] Marafioti, Andrés, et al. "Smolvlm: Redefining small and efficient multimodal models." arXiv preprint arXiv:2504.05299 (2025). https://huggingface.co/HuggingFaceTB/SmolVLM-500M-Instruct
> > >
> > > [InternVL3.5] Wang, Weiyun, et al. "Internvl3. 5: Advancing open-source multimodal models in versatility, reasoning, and efficiency." arXiv preprint arXiv:2508.18265 (2025). https://huggingface.co/OpenGVLab/InternVL3_5-1B

---

### Author Response · Authors · 2025-12-02
**Summary Comment**

Dear Reviewers, ACs, SACs and PCs,

We sincerely thank all reviewers for their thoughtful and constructive feedback, and we are also deeply grateful to ACs, SACs and PCs for devoting their valuable time and effort to handling our submission. After carefully studying each concern, we conducted additional analyses and experiments, and we summarize the key points and updates below.

---

Firstly, we are pleased to observe that the majority of reviewers recognized the innovativeness, clarity, and strong experimental evaluation of our work. Specifically:

* This paper introduces a **well-motivated and novel** caption-guided replay paradigm (Reviewers LRVX and DtAe), which is compatible with existing frameworks and can be **seamlessly integrated** to enhance replay fidelity (Reviewer DtAe). The visualization of feature drift in Fig. 1b is **insightful** (Reviewer LRVX).

* The proposed method **outperforms all competitors** in VLM-based CL benchmarks (Reviewer J4Rf). Experiments are conducted on **a variety of scenarios** with **extensive and thorough comparisons** (Reviewer KkVw), including MTIL and X-TAIL in full- and few-shot settings (Reviewer LRVX).

* The paper is well written, easy to follow, clearly presented (Reviewers J4Rf and LRVX), and easy to understand with a clear structure and diagram (Reviewer KkVw).

---

Secondly, we provide point-by-point responses to all four reviewers, and we revised the manuscript accordingly **(highlighted in blue)**. Importantly, all reviewers provided positive or encouraging feedback during the rebuttal: **one reviewer explicitly raised the score to 6, and two reviewers stated they would consider raising their score.** Specifically:

* *Reviewer J4Rf*: “these additions **significantly clarify** the computational profile of CaRD and help isolate the contribution of the method from the auxiliary models.” We also provided new experiments with very small captioners, **directly addressing the reviewer’s only remaining concern** regarding “resource symmetry.”
* *Reviewer LRVX*: the supplementary experiments were “**well-designed and targeted**” and “**effectively address and resolve all the questions**,” and they **raised the evaluation score to 6**.
* *Reviewer KkVw*: “the authors’ rebuttal **largely addressed my concerns**,” and would consider **raising the score**.
* *Reviewer DtAe*: the direction is **generally convincing**, task-specific LoRA fine-tuning for T2I models is **indeed an innovative idea**, and the explanation of DistKD differences from PODNet is **clear**. They would reconsider and **raise the score** after the remaining implementation details and additional results on CUB200, which we have provided in the response.

---

Below are the major additions made during the rebuttal:

**1. Complete computational cost (All reviewers)**

We redesigned Table 8 (page 10) to report the full offline replay cost, including caption generation, LoRA fine-tuning, synthetic image generation, and CLIP training. These results clarified that CaRD’s **total computational cost is comparable** to existing generative-replay methods and **introduces no extra training or inference cost** for CLIP.

**2. Comparison under matched generator setting (Reviewer J4Rf and KkVw)**

We added Table 9 (page 10) for fair comparison under matched generator settings, showing that CaRD **consistently outperforms** other generative-replay methods. Additional experiments with **very small captioners** (SmolVLM(0.5B) and InternVL3.5(1B)) show CaRD **maintains robust performance**, demonstrating gains primarily come from caption-guided replay, **not auxiliary model capacity**.

**3. Differentiation between DistKD and prior work (Reviewer LRVX, KkVw and DtAe)**

We clarified that DistKD is motivated by a previously unreported **multi-layer, multi-token feature drift** in VLM-based CL and is designed as a complementary, multi-level distillation formulation **tailored to VLM structures**. We further contrasted DistKD with Wasserstein alignment and PODNet, adding comparisons (Table 5, page 9) and visualizations (Figure 3, page 19).

**4. Fine-grained CIL (Reviewer DtAe)**

We added **intra-dataset class-incremental learning (CIL)** experiments on CUB (following PROOF protocol, Table 16, page 25). CaRD (only with generative-replay) is competitive with real-replay methods and surpasses them when combined with real samples.

**5. Improved presentation (Reviewer LRVX, KkVw and DtAe)**

We reorganized the Figure 1 and refined its terminology (using “class name & template” for consistency). And we added a Replay column to Table 1 and 2, and moved X-TAIL results into the main text.

---

Finally, based on the constructive suggestions from all reviewers, we have carefully revised our manuscript. Their comments have been invaluable in improving its overall quality. We sincerely appreciate the reviewers’ thoughtful feedback, and we are also grateful to the ACs, SACs and PCs for careful handling of our submission.

---

### Meta-Review · Area_Chair_bf7o · 2025-12-29

**Summary:**

This submission proposes CaRD, a caption-guided generative replay pipeline for VLM continual learning that stores instance-level captions and uses a LoRA-adapted text-to-image generator, coupled with a multi-layer distillation objective intended to mitigate feature drift in sequentially fine-tuned CLIP. Multiple reviewers independently flagged that the approach leans heavily on large external models (captioner + diffusion) and an offline multi-stage pipeline, raising fairness and “resource symmetry” questions relative to baselines and to the trained model itself. Reviewers also questioned the novelty of the proposed distillation component given close conceptual proximity to prior distribution/feature distillation and PODNet-style intermediate matching, and raised clarity/positioning issues (e.g., explicitly stating the CL setting, properly contextualizing replay vs non-replay baselines, and tightening related work/presentation). Overall, while the rebuttal adds helpful analyses and improves transparency, the paper still reads closer to an engineering-heavy combination of existing ingredients than a clean methodological advance with a clearly justified cost–benefit trade-off, leading to a weak reject.

**Reviewer Concerns:**

The rebuttal substantially improves transparency around total compute by expanding the cost accounting beyond per-step CLIP training and by separating offline replay construction from training/inference, which addresses the original “missing overhead” critique and reduces ambiguity about what Table 8 measures; it also responds reasonably to fairness concerns by adding matched-generator comparisons and by including smaller-captioner settings, which helps disentangle some gains from sheer auxiliary scale. The authors also clarified several presentation and positioning issues (explicitly defining MTIL/X-TAIL protocols, adding a replay indicator in tables, reorganizing key results, and patching related-work omissions such as LADA and prototype augmentation), and they provided additional intra-dataset CIL evidence on CUB that partially addresses scope questions about fine-grained settings. However, several central concerns remain only partially resolved: even with smaller captioners, the method’s practical appeal still depends on a multi-stage generator/captioner pipeline whose complexity and auxiliary footprint is materially different from many competing CL approaches, and the “offline” framing does not eliminate the real resource barrier for typical users. On the contribution side, the distillation component is better differentiated in writing and via comparisons, but the core idea still appears incremental relative to established intermediate-feature/disttribution alignment practices, and the paper does not fully crystallize what is fundamentally new beyond a sensible—but heavy—system integration. Finally, some fairness/interpretability issues linger around how much improvement comes from stronger generators versus algorithmic design, and while the added tables move in the right direction, the overall story remains somewhat over-claimed relative to the methodological novelty and deployability implied by the framing.

**Reviewer Scores:**

Reviewer LRVX explicitly raised their rating to 6 after finding the rebuttal persuasive and considering the added analyses and targeted experiments sufficient, so I would map their post-discussion stance to a clear accept-leaning score change. Reviewer J4Rf acknowledged that the expanded cost accounting and matched-generator comparisons clarified the computational profile, but they maintained a substantive remaining concern about resource symmetry and auxiliary footprint even after the additional smaller-captioner setting; given that persistence, I expect their score would stay at 4. Reviewer KkVw stated that the rebuttal largely addressed the listed issues yet still framed the practical value of using two auxiliary models as debatable; in line with that position and absent a clear endorsement, I expect their score would also remain at 4. Reviewer DtAe became generally convinced by the clarified time breakdown, the more explicit DistKD-vs-PODNet differentiation, and the strengthened CUB200 evidence with clear real/synthetic replay accounting; I would therefore anticipate a modest increase, contingent on the final paper maintaining these clarifications cleanly in the main text and tables.

---

### Decision · Program_Chairs · 2026-01-26

Reject